



# Dynamics and composition of the Asian summer monsoon anticyclone

Klaus-D. Gottschaldt[1], Hans Schlager[1], Robert Baumann[1], Duy S. Cai[1], Veronika Eyring[1], Phoebe Graf[1], Volker Grewe[1,2], Patrick Jöckel[1], Tina Jurkat[1], Christiane Voigt[1,3], Andreas Zahn[4] and Helmut Ziereis[1]

[1]Deutsches Zentrum für Luft- und Raumfahrt (DLR), Institut für Physik der Atmosphäre, Oberpfaffenhofen, Germany
[2]Delft University of Technology, Aerospace Engineering, Delft, The Netherlands
[3]Johannes Gutenberg-Universität, Institut für Physik der Atmosphäre, Mainz, Germany
[4]Karlsruher Institut für Technologie (KIT), Institut für Meteorologie und Klimaforschung, Karlsruhe, Germany

*Correspondence to*: Klaus-D. Gottschaldt (klaus-dirk.gottschaldt@dlr.de)

**Abstract.** This study places HALO research aircraft observations in the upper-tropospheric Asian summer monsoon anticyclone (ASMA) obtained during the Earth System Model Validation (ESMVal) campaign in September 2012 into the context of regional, intra-annual variability by hindcasts with the ECHAM/MESSy Atmospheric Chemistry (EMAC) model. The simulations demonstrate that tropospheric trace gas profiles in the monsoon season are distinct from the rest of the year.

Air uplifted from the lower troposphere to the tropopause layer dominates the eastern part of the ASMA's interior, while the western part is characterised by subsidence down to the mid-troposphere. Soluble compounds are being washed out when uplifted by convection in the eastern part, where lightning simultaneously replenishes reactive nitrogen in the upper troposphere. Net photochemical ozone production is significantly enhanced in the ASMA, contrasted by an ozone depleting regime in the mid-troposphere and more neutral conditions in autumn and winter.

An analysis of multiple monsoon seasons in the simulation shows that stratospherically influenced tropopause layer air is regularly entrained at the eastern ASMA flank, and then transported in the southern fringe around the interior region. Observed and simulated tracer-tracer relations reflect photochemical $O_3$ production, as well as in-mixing from the lower troposphere and the tropopause layer. The simulation additionally shows entrainment of clean air from the equatorial region by northerly winds at the western ASMA flank. Although the in situ measurements were performed towards the end of

summer, the main ingredients needed for their interpretation are present throughout the monsoon season.

A transition between two dynamical modes of the ASMA took place during the HALO ESMVal campaign. Transport barriers of the original anticyclone are overcome effectively when it splits up. Air from the fringe is stirred into the interiors of the new anticyclones and vice versa. Instabilities of this and other types occur quite frequently. Our study emphasises their paramountcy for the trace gas composition of the ASMA and its outflow into regions around the world.



# 1 Introduction

The Asian monsoon system is one of the largest and dominating atmospheric features on Earth. Episodic deep convection (Hoskins and Rodwell, 1995) related to the northwards shifted inter-tropical convergence zone (Lawrence and Lelieveld, 2010), elevated surface heating over the Tibetan plateau (Flohn, 1960; Fu et al., 2006), and orographic uplifting at the

southern/southwestern slopes of the Himalayas (Li et al., 2005; Liu et al., 2009b) contribute to an overall ascending air current. This drives an anticyclonic circulation, centred at 200 to 100 hPa (Dunkerton, 1995; Randel and Park, 2006; Garny and Randel, 2015).

Location, shape and strength of the ASMA strongly vary on intra-seasonal, inter-annual and longer timescales (Dunkerton, 1995; Lin et al., 2008; Kunze et al., 2010; Pokhrel et al., 2012), which is subject to ongoing discussion (Pan et al., 2016;

Nützel et al., 2016). An elliptical vortex is intrinsically unstable (Hsu and Plumb, 2001; Popovic and Plumb, 2001), thus prone to splitting up and mostly westward eddy shedding. Variable forcing by convection (Randel and Park, 2006; Garny and Randel, 2013), sub-seasonal oscillations (Lin et al., 2008; Goswami, 2012), the interaction with Rossby waves or mid-latitude synoptic disturbances (Dethof et al., 1999) add further complexity. In particular, the Rossby wave response to convection in the ASM region results in large-scale subsidence over the Arabian Peninsula (Rodwell and Hoskins, 1996),

making it one of the warmest and driest regions on Earth. The associated heat low supports an anticyclone (Lelieveld et al., 2009), which intermittently merges with the ASMA. If both anticyclones are separated, the western and eastern parts will be denoted "Iranian" and "Tibetan", respectively. "ASMA" will be used for the Tibetan anticyclone, or the whole system when merged. Based on reanalysis data of the National Centers for Environmental Prediction – NCEP1 (Kalnay et al., 1996) it was suggested that the core of the anticyclone oscillates between Iranian and Tibetan mode on a quasi-biweekly timescale (Zhang

et al., 2002). However, amongst 6 re-analyses such a bimodality is only found in NCEP1 and to a lesser degree in its successor (Nützel et al., 2016).

The interplay of the above dynamical ingredients makes the Asian summer monsoon a switch yard and mixing vessel for air masses of different origin and with different composition. Monsoon air is received by regions around the globe (Rauthe-Schöch et al., 2016), and was for instance shown to affect the tropospheric chemical composition in the Mediterranean

(Lelieveld et al., 2001; Lelieveld et al., 2002; Scheeren et al., 2003). A mid-tropospheric (400-500 hPa) summertime $O_3$ maximum over the eastern Mediterranean / Middle East region (Li et al., 2001; Lelieveld et al., 2009; Schuck et al., 2010; Akritidis et al., 2016) is enhanced by Asian monsoon outflow  (Liu et al., 2009b; Richards et al., 2013; Barret et al., 2016), but it is not clear if $O_3$ in the ASMA plume is generally enhanced or depleted (Lawrence and Lelieveld, 2010). Furthermore, the UT anticyclone is important for the exchange of air masses between the stratosphere and the troposphere. It episodically

drags stratospherically influenced air from the tropopause layer (TL) into the troposphere . Eddy shedding is also a means to transport tropospheric ASMA air into the stratosphere (Ungermann et al., 2016).

In the following "TL" refers to the mixing zone at the tropopause, where cross-tropopause exchange of air masses on average creates a gradient between stratospheric to tropospheric trace gas signatures. The TL is also denoted ExTL in the



extratropics and TTL in the tropics, reflecting that the dominating physical processes change at about the 30° circles of latitude. There are no rigid boundaries, but rather stratospheric influence decreases towards the troposphere over a range of several kilometres (Gettelman et al., 2011). In contrast, "upper troposphere" (UT) is used here to describe the altitude region that is dominated by the ASMA. Despite its importance for redistributing trace gases between boundary layer, troposphere

and lower stratosphere, the highly variable composition of the ASMA and the processes behind it are not well understood yet (Randel et al., 2016).

In situ measurements were conducted in the ASMA during the ESMVal field experiment with the High Altitude and LOng Range (HALO) research aircraft in September 2012. Gottschaldt et al. (2017) identified key processes for the interpretation of the in situ data from selected flight segments and we refer to that study as "accompanying paper" in the following. Those

processes include the entrainment of stratospherically influenced air at the eastern flank and its transport in the fringe around the interior of the ASMA, intermittent mixing of uplifted lower tropospheric with UT air, net photochemical $O_3$ production, and the splitting-up of the anticyclone into a Tibetan and an Iranian part. In the present study the specific situation observed during the HALO ESMVal campaign is put into a regional, seasonal and multi-annual perspective, which is provided by global chemistry climate simulations with the EMAC model. We examine, if the above processes are important beyond

individual flight segments and for the ASMA composition in general. This touches several key aspects of the UT monsoon system: dynamical and chemical coupling with convection, composition/reactive chemistry in the monsoon region, the relative importance of different reactive nitrogen sources, mixing of higher-latitude lower-stratospheric air into the tropical TL by the ASMA – all of which are only poorly understood (Randel et al., 2016).

A recent paper of Pan et al. (2016) discussed CO distributions in the monsoon region in the context of daily ASMA

dynamics. Another paper (Barret et al., 2016) focused on monthly budgets of CO and $O_3$, confined within the ASMA. Building on the ASMA observations during the HALO-ESMVal campaign, our study complements these papers by considering additional tracers on a 10-hourly scale to characterise key processes relevant for the $O_3$ distribution in the monsoon region. Section 2 briefly summarises the data used here, i.e. the in situ measurement techniques for selected tracers during the HALO ESMVal campaign, and the global chemistry climate simulations and trajectory calculations used for the

interpretation of the observations. Those methodological aspects are described in further detail in the accompanying paper. From all the observed and simulated tracers available, a subset is chosen for the present study. We discuss the intra-annual variability of these tracers in the ASMA region in section 3 by analysing the year 2012 of an EMAC simulation. In section 4 observed tracer-tracer relations are put in the context of simulated ones. The latter are not limited to the flight track of the HALO ESMVal campaign, but provide a wider view on the prevailing trace gas relations in the Asian monsoon region in

September 2012. Section 5 is dedicated to the interplay of the processes that contributed to the observed trace gas signatures and their relevance beyond the specific situation observed during the HALO ESMVal campaign. By analysing multiple monsoon seasons in the EMAC simulation we show that those processes occur frequently, discuss their implications for composition and transport in the ASMA, and note remaining open questions.




**2 Data**

We focus on the analyses of $O_3$, CO, hydrogen chloride (HCl) and reactive nitrogen (NO, $NO_x$, $NO_y$), as those tracers reflect the processes most relevant for the interpretation of in situ measurements in the ASMA during the ESMVal flight from Male (Maldives) to Larnaca (Cyprus) on 18 September 2012. All in situ measurements used here are based on a data set with 10 s

time resolution, which is available from the HALO database (https://halo-db.pa.op.dlr.de). The corresponding measurement techniques are described in the accompanying paper, and in more detail in publications about the individual instruments: CO (Hoor et al., 2004; Schiller et al., 2008; Müller et al., 2016), HCl (Jurkat et al., 2014; Voigt et al., 2014), NO/$NO_y$ (Ziereis et al., 2000), $O_3$ (Zahn et al., 2012). Among those tracers, only HCl mixing ratios were at the instrument's detection limit during the considered flight (Jurkat et al., 2016).

The transport pathways of air parcels before being encountered by HALO were calculated with the Lagrangian HYSPLIT model (Draxler and Hess, 1998; Draxler and Rolph, 2015). That analysis is described in the accompanying paper. It led to a pragmatic classification of the observations into 7 periods of interest (POI), which is also adopted here. The periods of interest number 3, 5 and 6 ("ASMA filament" in the following) were shown to be part of an UT air filament that was entrained by lower-/mid-tropospheric air at the eastern ASMA flank. This filament is part of the ASMA circulation and

associated with the entrainment of air from the TL. POI1 and POI7 cover ascent and descent, respectively. During POI2 HALO was flying in the UT south of the ASMA, and dived into the lower troposphere during POI4.

All simulation data of this paper stem from global chemistry climate simulations with the EMAC model (Jöckel et al., 2010), performed within the ESCiMo (Earth System Chemistry integrated Modelling) project (Jöckel et al., 2016) and the DLR-internal ESMVal project. Our reference simulation has been described and generally evaluated as RC1SD-base-10a by

Jöckel et al. (2016). Its set-up is designed for best possible comparability to observations by nudging of the dynamics to reanalysis data and covers the period 1980 – 2013 (excluding spin-up). Convection is not resolved in the simulation, but its effects are captured by a parameterisation in EMAC. We have shown in the accompanying paper that this simulation reproduces the measured trace gas mixing ratios along the HALO flight track of 18 September 2012 reasonably well. In the following we analyse the results of the reference simulation for the ASMA region (Fig. 1) on three timescales: (i) September

2012 to elucidate the synoptic situation during the flight; (ii) the entire year 2012 to discuss the peculiarities of the monsoon season with respect to other seasons; (iii) 5 consecutive years (2010 – 2014), either entirely or the monsoon months July to September. The goal of the longer analyses (ii, iii) is to put the in situ observations into an intra- and inter-annual context. Additional EMAC simulations were performed in quasi chemistry transport model mode (Deckert et al., 2011; Gottschaldt et al., 2013) to test the impact of lightning $NO_x$ ($LNO_x$). Those are described in Appendix A and referred to always by their

acronyms. For brevity, "simulation" without further specification refers to RC1SD-base-10a in the following.





### 3 Simulated intra-annual variability of trace gas dynamics in the monsoon region

A sudden enhancement of measured $O_3$ when HALO entered the ASMA from the south triggered the accompanying paper and this study, since those measurements contrast the presumption of decreased $O_3$ in the ASMA. The other tracers considered here shall help to understand the corresponding $O_3$ dynamics and photochemistry. In this section we discuss

simulated trace gas profiles in the ASMA region and their evolution throughout the year. Some features and processes that distinguish the monsoon season are highlighted.

Figures 2 and 3 show the simulated evolution of trace gas profiles relative to the EMAC tropopause throughout 2012, laterally averaged separately for the western (Iranian) and eastern (Tibetan) ASMA region (Fig. 1), respectively. The regions' delimitations were chosen by eye, considering the following: (i) For putting the measurements into perspective, the

regions shall capture the synoptic situation during the HALO ESMVal campaign (see the accompanying paper for details); (ii) Both parts shall be equally sized; (iii) The variability of the ASMA's location and extent shall be covered. The chosen meridional range of 15°N to 35°N covers the simulated ASMA ridgeline for most of the monsoon season (Fig. 10b). We note that the EMAC tropopause is diagnosed by a potential vorticity of 3.5 PVU in the extratropics and by the WMO definition between 30°N and 30°S (Jöckel et al., 2006). The extratropics are dominated by baroclinic wave activity and

downward stratospheric circulation, the tropics by radiative-convective balance and upward stratospheric circulation (Gettelman et al., 2011). The zonal ranges are 30°E to 65°E and 65°E to 100°E for Iranian and Tibetan regions, respectively. For comparison, Yan et al. (2011) classified anticyclonic centres between 50°E and 67.5°E as Iranian mode, and between 80°E and 92.5°E as Tibetan mode.

We decided not to adapt the regions dynamically to the actual ASMA, because the boundary definitions we are aware of

(Ploeger et al., 2015; Barret et al., 2016; Pan et al., 2016) emphasise the concept of a closed ASMA volume or transport barriers on monthly or seasonal timescales. However, the ASMA boundaries are not always well defined, particularly during transitions between different dynamical modes. Our pre-fixed regions allow an unbiased view on the effects of complex, 10-hourly dynamics. This comes at the price that features from outside the ASMA might contribute to the analyses occasionally, but we are confident that monsoon-related features dominate the lateral averages. Our approach detects differences between

Iranian and Tibetan parts, because the corresponding circulation is tied to geographical features of the respective regions. Enhanced CO is considered to be a chemical characteristic of the ASMA (Pan et al., 2016), and increased GPH is a dynamical proxy (Barret et al., 2016). Simulated seasonal mean distributions of CO and GPH at 168 hPa are shown in Fig. 1 for comparison, indicating that our regions well capture the ASMA of 2012.

Corresponding plots of the seasonal mean distribution at 168 hPa in the ASMA region of the other tracers considered here

are available in the supplementary material (Figs. S1 – S3). In the following all simulated profiles are given in vertical coordinates relative to the tropopause, as pressure (Figs. 2, 3) or distance (Fig. 6, supplementary material, Figs. S4 – S8, S16).



### 3.1 Ozone

Enhanced $O_3$ mixing ratios in the TL exhibit a strong vertical gradient, which reflects the degree of mixing between $O_3$-poor UT air and $O_3$-rich air from the lowermost stratosphere. Due to its long chemical lifetime, $O_3$ fluctuations in the TL are generally determined by transport on timescales of days. It takes 1 or 2 weeks in the lowermost stratosphere to smooth out
short timescale fluctuations and adopt the mean signature corresponding to a certain vertical distance to the tropopause (Sprung and Zahn, 2010). However, this might not hold under the particular photochemical UT conditions of the ASMA.
$O_3$ is subject to complex photochemical production and loss processes, which in the UT are masked even by small stratospheric entrainment. Steep vertical gradients across the tropopause dominate $O_3$ profiles in the monsoon regions (Figs. 2ef), but the profiles also show temporal fluctuations of various timescales. Note that our lateral averaging regions are rather
large and smaller-scale structures get smoothed out, e.g. when an $O_3$-poor interior is combined with an $O_3$-rich fringe.
There is increased influx from the stratosphere in spring, enhancing $O_3$ in the UT. This is in accordance with the study of Cristofanelli et al. (2010), but in contrast to their study there are non-negligible $O_3$ enhancements connected to the stratosphere during the monsoon season (Fig. 2f, circled). In this respect our simulation is however consistent with trace gas budget considerations for the ASMA (Barret et al., 2016) and the TTL (Konopka et al., 2010). Entrainment from the TL
rather than deep from the stratosphere could reconcile the different findings. In the accompanying paper TL entrainment is considered to be the source of stratospherically influenced trace gas signatures in the HALO ESMVal ASMA observations.
Enhanced $O_3$ prints through in the averaged profiles of the eastern ASMA part only from the tropopause to about 200 hPa below the tropopause, while the mid-troposphere in the Tibetan part is dominated by particularly $O_3$ poor air during the monsoon season (Fig. 2f). The latter is consistent with the findings of Safieddine et al. (2016). $O_3$ depletion in the mid-
troposphere of the eastern part is contrasted by enhanced $O_3$ in the mid-troposphere during the monsoon season in the western part (Fig. 2e, circled), marking the well known summertime $O_3$ maximum there.

### 3.2 Carbon monoxide

Enhanced CO is produced in combustion and thus a good tracer of boundary layer pollution, which is spatially and temporally rather heterogeneous. In the troposphere it is an $O_3$ precursor. CO is also a product of the oxidation of methane
($CH_4$) and higher hydrocarbons with the hydroxyl radical (OH). It has its predominant sink in a reaction with OH. Without tropospheric entrainment the sink dominates under stratospheric conditions, and CO mixing ratios decrease by about an order of magnitude across the tropopause (Hoor et al., 2002). This is reflected in the evolution of CO profiles in the ASMA region in 2012 (Figs. 2cd). CO-poor air dominates in the UT during spring, consistent with the stratospheric influx indicated by $O_3$.
CO-rich air rises throughout the troposphere of the eastern part during the monsoon season (Fig. 2d, circled). On the western side there is a conspicuous CO depleted zone 200 to 300 hPa below the tropopause during the monsoon season (Fig. 2c,



circled), while CO is episodically enhanced in the UT. Uplifted air with enhanced CO mixing ratios hardly reaches higher than 450 hPa below the tropopause in summer.

This difference between CO profiles in the Tibetan and the Iranian parts is consistent to the findings of Pan et al. (2016). Occasional horizontal transport in the UT from the eastern to the western part of the ASMA is an explanation for spatio-
temporal evolution of CO mixing ratios, indicating that trace gas signatures in the Iranian part are dominated by the UT outflow of the Tibetan part of the ASMA.

### 3.3 Hydrochloric acid

As tracer for stratospheric air we use HCl. There are no significant HCl sources in the UT, apart from stratospheric entrainment (Marcy et al., 2004). Marine boundary layer sources for the ASMA are small (Randel et al., 2010; Bergman et
al., 2013), wet scavenging in clouds effectively prevents convective transport of HCl to the UT, and no injections of HCl from volcanic activity affected the ESMVal flight from Male to Larnaca. Like $O_3$, HCl has a photochemical UT lifetime of the order of weeks (Marcy et al., 2004). This makes HCl a viable tracer of stratospheric $O_3$ entrainments, until it is selectively removed by wet scavenging. Consequently the simulated HCl profiles (Figs. 2ab) show a strong anti-correlation with CO in the UT, with increased HCl in times of stratospheric influx (e.g. Fig. 2b, blue circle), and decreased HCl in the
monsoon season. Stratosphere-to-troposphere exchange is pronounced during spring, consistent with the seasonality of the Brewer-Dobson circulation (Holton et al., 1995).

HCl is also emitted by the sea. HCl plumes in the Iranian part rise to about 400 hPa below the tropopause in summer (Fig. 2a, circled), just like CO. Predominantly dry conditions in the western part prevent HCl from being washed out. However, CO and HCl are temporally anti-correlated in the mid-troposphere. We attribute this to alternating marine and continental
origins in the uplifted air (Figs. 2ac).

Some HCl is also slowly descending from the tropopause into the mid-troposphere, as indicated by tilted patterns of enhanced HCl, which start at the tropopause and propagate downward (marked by an arrow in Fig. 2a). Similar tell-tale signs of descent also print through in other species in the Iranian part during summer.

There is almost no HCl in the Tibetan part throughout the monsoon season (Fig. 2b, black circle), except for the UT.
Convection and thunderstorms are galore during the monsoon season in South Asia (see Fig. 3h and the supplementary material of the accompanying paper). Washing out does not affect CO, but effectively prevents transport of HCl to higher altitudes. Thus enhanced HCl in the UT indicates stratospheric influence. Predominantly continental origins also contribute to an increased CO/HCl ratio in the rising plumes of the eastern part.

### 3.4 Reactive nitrogen

Mixing ratios and the distribution of total reactive nitrogen ($NO_y$; comprising NO, $NO_2$, $HNO_3$, PAN, HONO, $N_2O_5$, $HO_2NO_2$, $NO_3$ as the most abundant species) are controlled by a variety of natural and anthropogenic sources, such as lightning, stratospheric input, soil microbiology, biomass burning, air traffic emissions and other fossil fuel combustion. The



apportionment of NO$_y$ sources varies between different parts of the atmosphere. Nitrogen oxides are key parameters in atmospheric chemistry, partly controlling the ozone production in the troposphere and lower stratosphere. In the UTLS, enhanced NO$_y$ originates both from tropospheric and from stratospheric sources. In the lower troposphere odd nitrogen species are co-emitted with carbon monoxide in combustion processes, resulting in a strong correlation between both

species. In the stratosphere HNO$_3$ is the main component of the NO$_y$ family. It is mainly produced from N$_2$O photo-oxidation.

Simulated NO$_y$ profiles in the ASMA region distinctively differ between the summer monsoon season and the rest of the year (Figs. 3cd): during summer there is more NO$_y$ in the UTLS and mid-troposphere in both, the Tibetan and Iranian regions (Figs. 3cd, S5cd, S6). Episodes of enhanced NO$_y$ (~1.5 nmol/mol) in the UT are frequent in the Tibetan part during

summer, and alternate with periods of decreased NO$_y$ (~1.0 nmol/mol). However, the altitude region just above the tropopause is hardly affected by this UT variability and maintains an average mixing ratio of ~1.2 nmol/mol NO$_y$ (Figs. 3cd; for a better resolved UTLS see also the supplementary material, Fig. S5). NO$_y$ mixing ratios generally increase with altitude in the lower stratosphere, but reach 1.6 nmol/mol only at about 15 hPa above the tropopause. E-shaped NO$_y$-profiles dominate the Tibetan part, with maxima in the lower troposphere, in the UT and in the lower stratosphere (see

supplementary material for an example, Figs. S5cd, S6g, S7b). Less NO$_y$ is simulated in many profiles for the mid-troposphere and just above the tropopause transport barrier (see also the supplementary material, Figs. S5, S7). E-shaped NO$_y$–profiles were also reported by the NOXAR measurement campaign in the northern mid-latitudes and corresponding modelling studies (Grewe et al., 2001). The E-shape in northern mid-latitudes was in part attributed to aviation NO$_x$ emissions (Rogers et al., 2002), but aviation effects are much smaller in the tropics (Gottschaldt et al., 2013). Instead of

aviation emissions, in situ production of lightning NO$_x$ in the prevalent thunderstorms of the monsoon season increases NO$_y$ in the UT over South Asia (Fig. 3h, see also Appendix A). Thus a possible explanation for the E-shaped NO$_y$-profiles in the eastern ASMA part during the monsoon season is as follows: NO$_y$ from boundary layer sources' pollution is uplifted, and solvable NO$_y$-components (e.g. HNO$_3$, N$_2$O$_5$) become increasingly washed out (Fig. 3d). At about 400 hPa below the tropopause only non-solvable components (e.g. NO$_x$) are left. Episodes of increased NO$_y$ in the UT are well correlated with

increased lightning NO$_x$ emissions (Figs. 3d, 3h). NO$_y$ mixing ratios however increase with altitude above the tropopause, due to increased photochemical production of HNO$_3$ in the stratosphere. With little in situ production and not much transport from above or below, NO$_y$ mixing ratios in the region between the tropopause and 15 hPa above the tropopause are often smaller than in the adjacent altitudes.

Profiles in the western, i.e. Iranian ASMA part (Fig. 3c) have a different history of origins, and with just one minimum in the

mid-troposphere are mostly C-shaped (supplementary material, Figs. S5, S7). During summer the Arabian Peninsula is dry. Convection (as indicated by lightning NO$_x$ emissions in Fig. 3g) is mainly localised near the Bab al-Mandab Strait (Fig. A1), i.e. at the edge of the region we defined for calculating profiles of the Iranian part of the ASMA. Washing out is negligible throughout most of the Iranian region (Fig. A1, see also satellite pictures in the supplementary material of the accompanying paper), and therefore NO$_y$ can rise to about 400 hPa below the tropopause (circled in Fig. 3c). Downward transport (as





indicated by anti-clockwise tilted signals, one example marked by an arrow in Fig. 3c) dominates above that altitude, preventing further uplift. With little in situ production of lightning $NO_x$ over the Arabian Peninsula in summer (Figs. 3g and A1), UT $NO_y$ in the Iranian part is dominated by the outflow of the Tibetan part.

$NO_x$ ($NO + NO_2$) is part of $NO_y$, but characterises young emissions. There is less $NO_x$ in the mid-troposphere than in the UT

above and in the lower troposphere below (Figs. 2ab, S5ab). This indicates that that relatively little $NO_x$ rises to the UT from the lower troposphere, in both parts and throughout the year. The conversion of non-solvable $NO_x$ into solvable $NO_y$ components facilitates the subsequent loss by washing out. Both, $NO_x$ and $NO_y$ increase with altitude above the tropopause, and recycling of stratospheric $NO_y$ additionally increases $NO_x$ in the troposphere. Therefore stratospheric influx contributes to increased $NO_x$ mixing ratios in the UT (blue circles in Figs. 2b and 3b). Note that the timing of most enhanced $LNO_x$

emissions during spring in Fig. 3h does not match the timing of most enhanced $NO_x$ in the circled period in Fig. 3b. Enhanced UT $NO_x$ during the monsoon (Figs. 3ab, black circles) is rather due to lightning $NO_x$ emissions in the Tibetan part than to stratospheric entrainments. As a combination of the different effects affecting $NO_x$ and $NO_y$, the $NO_x/NO_y$ ratio maintains a broad maximum in the TL throughout the year (Figs. 3ef). The monsoon season is characterised by particularly little fluctuations of $NO_x/NO_y$ (Figs. 3ef, circles). During the monsoon, the $NO_x/NO_y$ ratio in the UTLS is higher in the

western than in the eastern ASMA part. This indicates preferential export of high-$NO_x$ air from the Tibetan part, or is an artefact of the possible dominance of a single source of $LNO_x$ in the Iranian averaging region (Fig. A1). Only NO was measured during the HALO ESMVal campaign, but daytime NO is a good proxy for $NO_x$.

### 3.5 Photochemical ozone production

The net photochemical $O_3$ production rate (Figs. 2gh) is derived from the difference of EMAC simulated diagnostic tracers

ProdO3 and LossO3 (Jöckel et al., 2016). Here we take into account effective ozone production and loss terms following (Crutzen and Schmailzl, 1983) and extended by Grewe et al. (2017, see their supplement). We note that the origin of high-$O_3$ biases in the simulation (Jöckel et al., 2016) is not resolved yet. Uncertainties in the chemical mechanism (Gottschaldt et al., 2013) also impose uncertainties onto the net photochemical $O_3$ production rate. In contrast to the other tracers, there is no equivalent tracer in the HALO ESMVal measurements that could be used for independent evaluation of the photochemical

$O_3$ production.

More $O_3$ is produced than destroyed in the ASMA, signalled by a pronounced maximum of net $O_3$ production in the UT during the monsoon season (Figs. 2h, circled). This is accompanied by net ozone destruction during the monsoon season 300 hPa below the tropopause and lower. At the tropopause and slightly above there is a local minimum of net $O_3$ production, followed by increased net $O_3$ production in the stratosphere.

$O_3$ photochemistry is dominated by photolytic cycles in the stratosphere, and catalytic cycles in the troposphere. Net $O_3$ production by the latter non-linearly depends on the availability of precursors like $NO_x$, CO and hydrocarbons. For instance, net $O_3$ production is at maximum at roughly 0.3 nmol/mol $NO_x$, but the optimum $NO_x$ mixing ratio depends on several other





parameters (Grooß et al., 1998; Jaeglé et al., 1999). We consider CO (Fig. 2d) not only as an ozone precursor in itself, but also as a proxy for co-emitted hydrocarbons (volatile organic compounds, VOCs).

The simulated profiles of Fig. 2h show that enhanced net $O_3$ production in the ASMA has two prerequisites: enhanced UT $NO_x$ meeting an enhanced supply of uplifted other precursors. We note that net $O_3$ production is enhanced during the spring

periods of enhanced lightning NOx (circled in Fig. 3h), but still about 3 nmol mol$^{-1}$ day$^{-1}$ smaller than during summer (see also supplementary material, Figs. S8cd): there is less CO in the UT during spring (Fig. 2d) and lightning $NO_x$ is available only locally (Fig. A1: compare April vs. August). The local minimum of net $O_3$ production at the tropopause is due to missing uplifted $O_3$ precursors, while the decreasing net $O_3$ production 200 hPa below the tropopause and lower is due to missing $NO_x$. Note that in contrast to these conclusions from laterally averaged profiles, $O_3$ production in any specific air

mass might still be limited by either $NO_x$ or CO.

Both, $NO_x$ and other precursors are more abundant in the Tibetan part UT, resulting in higher photochemical $O_3$ production than in the Iranian part (Figs. 2gh; see also supplementary material, Fig. S8 for time averaged profiles). $O_3$ depleting conditions prevail in the mid-troposphere over the Arabian Peninsula throughout the summer (Fig. 2g, circled). Thus increased $O_3$ there (Fig. 2e) must be due to transport. Increased net photochemical $O_3$ production certainly contributes to

relatively high $O_3$ mixing ratios in the ASMA, despite the influx of $O_3$-poor air linked to the CO-rich updraughts. Entrainment of $O_3$-rich air from the TL additionally enhances $O_3$ in the ASMA.

## 4 Tracer-tracer relations in September 2012

The distribution of points in a tracer-tracer diagram (i.e. mixing ratios of two species encountered simultaneously) provides hints on the origin and evolution of air masses. Here we focus on CO versus $O_3$ (Fig. 4) and HCl versus $O_3$ (Fig. 5), while

$NO_x$ versus $O_3$ and $NO_x$ versus $NO_y$ are shown in the supplementary material (Figs. S9, S10). A short primer for the interpretation of such diagrams is provided in Appendix B.

In order to place the observed tracer-tracer relations into context, we plot the measured samples together with grid-cell samples from the EMAC simulation. Because simulation output along the flight track is too sparse for a meaningful comparison (e.g. POI3: about 180 observations correspond to only 3 simulated samples). Therefore 5000 simulated samples

per panel are chosen randomly, from the entire month of September 2012 and from throughout the ASMA region, as shown in Fig. 1 (Tibetan plus Iranian parts). Plotting all corresponding samples form the EMAC simulation would impair the visibility of clustering. Two different vertical ranges are chosen. The range from 50 hPa above to 100 hPa below the actual EMAC tropopause (Figs. 4a, 5a, S9a, S10a) provides a zoom-out view of possible tropospheric and stratospheric tracer mixing ratios and tracer-tracer relations for the time of year and region of the measurements. Zooming-in to the altitude

range of measurements, we choose tropospheric tracers from the pressure altitude range 200 hPa to 100 hPa (Figs. 4b, 5b, S9b, S10b).



The observations from the beginning of POI2 to the end of POI6 are shown in (Figs. 4c, 5c, S9c, S10c), where colouring corresponds to measurement time. Similar colours indicate spatial and temporal proximity, a prerequisite for mixing lines.

### 4.1 Ranges covered by observed and simulated tracer-tracer distributions

POI2 is marked by dark blue dots in Figs. 4c, 5c, S9c, S10c, and is clearly distinct from the measurements in the ASMA filament (orange boxes). The ranges covered by the measurements are also given in the corresponding panels with simulated data, but are adjusted for model biases there. Those biases were estimated according to comparisons by eye, of measured versus simulated trace gas mixing ratios along the flight track in the ASMA filament (see accompanying paper). All measured ranges fit into the simulated monthly averages for September 2012 in the ASMA region, so the simulation captures this aspect well and the measurements are unlikely to represent an exceptional situation. We also note that all measurements clearly fall into the tropospheric regions of the respective simulated tracer-tracer spaces. This is no surprise: all HALO ESMVal measurements considered here were taken well within the troposphere, and tropospheric trace gas signatures are expected to dominate the distribution of points in the tracer-tracer diagrams.

### 4.2 In situ photochemistry, tropospheric and TL contributions

CO versus $O_3$ (Fig. 4): $O_3$ and CO display opposite gradients across the tropopause, and globally have lifetimes of several months in the UT (IPCC, 2013). Thus mixing lines in a CO versus $O_3$ scatter plot are generally suited to identify stirring and mixing processes in the UT that occur on timescales of days to weeks, including cross-tropopause mixing (Fischer et al., 2000). The well known L-shape (Hoor et al., 2002; Pan et al., 2004; Müller et al., 2016) is reproduced by the simulation in the CO vs. $O_3$ diagram for the UTLS (Fig. 4a), consisting of a CO-poor & $O_3$-rich stratospheric branch, connected by UTLS mixing lines to a CO-rich & $O_3$-poor tropospheric branch.

However, the above studies (Hoor et al., 2002; Pan et al., 2004; Müller et al., 2016) focused on the extratropics. The ASMA is mostly situated in the tropics, where trace gas mixing ratios are controlled by different processes (Gettelman et al., 2011). The ASMA in particular constitutes a special atmospheric situation, because a continuous resupply of rapidly uplifted lower tropospheric air impedes UT photochemical equilibrium there. $O_3$ is photochemically produced in the ASMA at a net rate of almost 4 nmol/mol/day (Fig. 2h). Only 2 weeks are needed to increase $O_3$ mixing ratios by 50 nmol/mol, i.e. to produce the $O_3$ enhancement observed at the beginning of POI3. This is not much longer than the advection timescales (~10 days) discussed in the context of the HALO ESMVal campaign. Thus photochemical production needs to be considered as an alternative to stratospheric in-mixing for explaining enhanced $O_3$ in the ASMA. Photochemical ageing increases $O_3$ and depletes CO here.

Mixing lines with negative slopes in CO vs. $O_3$ space dominate the UT observations (black dotted in Fig. 4c). Such mixing lines in the troposphere could result from one or a combination of the following: (i) mixing between stratospherically and tropospherically influenced air masses; (ii) mixing between photochemically aged and freshly uplifted lower tropospheric



air; (iii) an $O_3$ depleting photochemical regime (Baker et al., 2011). While the latter is unlikely in the ASMA (Fig. 2h), we need to consider additional tracers to disentangle stratospheric influence and photochemical ageing.

HCl versus $O_3$ (Fig. 5): As discussed in the context of Fig. 2, HCl is a proxy for stratospheric entrainment and CO marks tropospheric influence. Consider the hypothetical case of constant HCl (indicated by schematic line L1 and parallels in Figs.

4b, 5b): increasing $O_3$ corresponds to increasing CO then. The trace gas gradients along that hypothetical line reflect a gradient in net $O_3$ production rather than differences with respect to stratospheric influence between two reservoirs. Now consider the opposite case, i.e. constant CO (hypothetical line L2 in Figs. 4b, 5b): increasing O3 corresponds to increasing HCl, indicating a gradient of stratospheric influence. CO mixing ratios decrease for increasing HCl in the special case of constant $O_3$ and different HCl mixing ratios (hypothetical line L3 in Figs. 4b, 5b). This indicates mixing between a

tropospheric and a stratospheric reservoir, where two opposite effects lead to almost constant $O_3$ mixing ratios: increased net $O_3$ production in air with decreased HCl, versus both increased $O_3$ and HCl in the more stratospheric components. In intermediate cases the trace gas gradients in the tracer-tracer plots reflect a combination of gradients in in-mixing as well as in situ photochemistry. Spatial gradients of photochemical $O_3$ production dominate over gradients of stratospheric influence (i.e. in-mixing from the TL or stratosphere) within the sampled air mass, if increasing $O_3$ correlates with increasing CO and

decreasing HCl (hypothetical line L4 in Figs. 4b, 5b). In contrast, gradients of stratospheric or TL in-mixing dominate, if increasing $O_3$ correlates with increasing HCl and decreasing CO (hypothetical line L5 in Figs. 4b, 5b).

The measurements (Figs. 4c, 5c) mostly –but not exclusively- show the latter case: neighbouring points form negatively sloped lines in CO vs. $O_3$ space (black dotted in Fig. 4c), corresponding to horizontal to positively sloped lines in HCl vs. $O_3$ space (black dotted in Fig. 5c). Thus observed trace gas gradients are mostly due to gradients of stratospheric influence on

some well mixed UT background. This could either be entrainment of tropospheric air into a more stratospheric background, or entrainment of TL air into a more tropospheric background. There are also a few almost vertical mixing lines in Fig. 5c, mostly at the lower end of observed HCl mixing ratios. We interpret those vertical components as a result of mixing between young and aged lower tropospheric air, because HCl mixing ratios in convectively uplifted air masses are decreased. Systematic HCl gradients –like across the tropopause- are not expected in convectively uplifted air. $O_3$ variability in such air

masses is at least partly due to different amounts of in situ produced $O_3$. However, mixing between aged and young tropospheric air alone cannot explain the observations.

We further note that mixing lines in Fig. 5c cover similar ranges of HCl, but are separated by different levels of $O_3$. The corresponding background air had seen similar amounts of stratospheric influence, but different $O_3$ production. As long as all points of an individual mixing line are subject to similar $O_3$ production, the entire line will be shifted to different $O_3$ levels.

The $O_3$ ranges covered by individual mixing lines are similar to the offsets between different lines. Individual mixing lines in the measurements cover timescales of about 20 minutes (Fig. 4c), corresponding to 300 km at typical HALO speeds. The flight track in the ASMA filament altogether covers more than 3000 km and multiple mixing lines were found on that scale. Summarizing, our observations of $O_3$, HCl and CO in an ASMA filament show that: (i) Both, photochemical production and TL/stratospheric in-mixing contribute to increased $O_3$ in the observed ASMA filament; (ii) small-scale gradients of



stratospheric influence are superimposed on background regions that are rather homogeneous on small scales (hundreds of kilometres), but differ in their amounts of photochemically produced $O_3$ on larger scales (thousands of kilometres).

## 5 Discussion of the interplay of dynamics and composition in the ASMA beyond the HALO ESMVal campaign

In this section the observed trace gas signatures are related to simulated photochemical, transport and mixing properties of the ASMA for timescales beyond the HALO ESMVal campaign by analysing selected processes in the EMAC simulation for 5 consecutive monsoon seasons (2010 – 2014). We note upfront that the intra-annual variability of trace gas dynamics in the Tibetan and Iranian ASMA regions as discussed in detail for the year 2012 in section 3 is largely similar in the other considered years (Fig. 6; supplementary material, Fig. S16). Building on that, we discuss additional aspects of the following processes: (1) Entrainment of $O_3$-rich TL air at the northern or eastern edge of the anticyclone; (2) Photochemical in situ $O_3$ production that also contributed to the increased $O_3$ observed during our campaign; (3) Radial stratification in the ASMA circulation; (4) Stirring related to dynamical instabilities of the anticyclone. Finally we consider the interplay of those processes and the implications for trace gas distributions in the ASMA and its outflow. The term "interplay" is thereby used in a neutral sense regarding the direction of feedbacks between different processes: It subsumes mostly one-way interactions (e.g. emissions affecting $O_3$ production, dynamics affecting trace gas distributions) here.

### 5.1 Entrainment of lower tropospheric and TL air

The uplift of lower tropospheric air to the UT is a well known characteristic of the ASMA (see section 1 and references therein). We consider enhanced CO in the ASMA as a marker for air of lower tropospheric origin. Simulated CO profiles in the Tibetan region show episodes of such uplift in every monsoon season 2010 – 2014 (Fig. 6b), and particularly throughout the monsoon season of 2012 (Fig. 2d). This is consistent with the HALO ESMVal measurements, since the trace gas gradients observed in the ASMA can be explained by mixing between lower tropospheric air and stratospherically influenced air (section 4 and accompanying paper).

Here we focus on the less well known entrainment of stratospheric or TL air into the free troposphere, which is supported by the unique thermodynamic conditions over the Tibetan plateau in summer. The heating of the plateau mainly initiates and maintains the ASMA. Thereby PV surfaces are lifted to form domes above the Tibetan plateau, while potential temperature surfaces form wells at the same time. Furthermore, the transition from the extratropics to the tropics occurs at the same latitude range and is accompanied by a steeply inclined tropopause (Fig. 7). The dominating physical processes differ between the extratropics and the tropics (Gettelman et al., 2011), resulting in a generally higher tropopause in the tropics. An isocontour of potential vorticity (PV) is commonly used to diagnose the tropopause in the extratropics, e.g. at 3.5 PVU in EMAC. Flows in the atmosphere, as a first-order approximation, tend to follow isentropic surfaces. Between 20°N and 30°N, isentropes from the extratropical lower stratosphere intersect the tropopause, some almost perpendicular (Fig. 7). The prevailing northerly winds (Kunze et al., 2010) of the eastern ASMA flank tend to transport high-PV (stratospheric or TL)



air along the isentropic surfaces into the troposphere (Ren et al., 2014; Kunz et al., 2015). This effect was also detected by Konopka et al. (2010) as enhanced horizontal transport of $O_3$-rich air from the extratropics into the TTL. Such transport from the TL or even the extratropical stratosphere into the free tropical troposphere may not leave a tell-tale signature of increased potential temperature (Tpot) in the corresponding air masses in the tropics. This includes the 350 K – 370 K isentropes that

were encountered during the HALO ESMVal campaign in the tropics (Fig. 7; supplementary material, Figs. S11, S12). Additional stratospheric or TL contributions to the outer ASMA fringe other than at the eastern flank are also plausible. The eastern Mediterranean and Central Asian region is a global hot spot of tropopause folding activity (Tyrlis et al., 2014), which is related to ASMA dynamics and generates enhanced $O_3$ levels through stratosphere-troposphere exchange (Akritidis et al., 2016). If the ASMA circulation encompasses that tropopause folding hotspot, it may pick up stratospheric entrainments (see

also supplementary material, Fig. S13). Lawrence and Lelieveld (2010) already suggested that $O_3$ rich air masses might be swept around the ASMA.

A stratospheric influence is manifested in our measurements by increased HCl mixing ratios in combination with other tracers (see section 4.2). Furthermore the detailed analyses of POI3 in the accompanying paper show entrainment of TL air into the ASMA circulation at the eastern flank of the UT anticyclone. No indication for other stratospheric contributions was

found. The stratospheric influence in the observed ASMA filament is consistent with recent TL entrainment at the eastern ASMA flank, but also with earlier entrainments of TL or stratospheric air. In the latter case stratospherically influenced air was already circling in the ASMA fringe before arriving at the eastern flank.

Did the HALO ESMVal campaign encounter an exceptional situation, or does TL entrainment at the eastern ASMA flank occur more often? We use HCl as a tracer for TL entrainment in the altitude range of the measurements in the simulation.

Indeed situations similar to the one seen in September 2012 occur quite regular in the monsoon season. Figure 8 shows a collection of snapshots of the corresponding simulated synoptic distributions of HCl: one HALO ESMVal campaign-like situation was selected from each month of the monsoon seasons (July – September) of the years 2010 – 2014. In each panel (Fig. 8) the south-eastern ASMA flank is marked by a filament of enhanced HCl, which often protrudes from a tropopause trough at the eastern flank. HCl mixing ratios steeply decrease towards the ASMA interior in all snapshots, which can be

explained by HCl-poor upwellings. Those lower tropospheric upwellings are the main driver of the ASMA circulation and mostly located in the eastern part of the Tibetan anticyclone (Nützel et al., 2016).

A more continuous exploration of the frequency of TL entrainment in the simulation is given in Fig. 10. Figure 10b shows the zonal wind fraction ($u/\sqrt{u^2 + v^2}$) along a meridional (N-S) transect throughout the monsoon seasons of 2010 to 2014. The UT transect is located at 90°E, i.e. approximately through the centre of the Tibetan anticyclone. This is a proxy for the

meridional location of the ASMA. The pattern corresponds to an anticyclone, with its centre (zero = white = "ridgeline") moving north and south around 30°N. Enhanced tropospheric HCl mixing ratios south of the ridgeline serve as an indicator of TL entrainment (Fig. 10c). Caution with this interpretation is only needed at times when the ASMA is shifted to the West (compare Fig. 10a), because meridional transects in Fig. 10c may then be too far within the eastern ASMA flank. Episodes



of increased HCl in the southern or eastern ASMA flank cover at least half of the time axis, showing that entrainment of TL air into the ASMA circulation is quite a common process.

### 5.2 Photochemical $O_3$ production and ageing

A net $O_3$ producing photochemical regime prevails in the ASMA throughout the monsoon season (Figs. 2gh), and thus
observed $O_3$ mixing ratios are partly due to in situ production (see section 4.2). $O_3$ production may be limited by the availability of precursors, namely $NO_x$ and CO (and co-emitted, other volatile organic compounds). Furthermore, net $O_3$ production non-linearly depends on $NO_x$ mixing ratios, being at maximum at about 300 pmol/mol (see section 3.5). $NO_x$ and CO distribution show opposite vertical gradients in the ASMA: $NO_x$ mostly decreases from the stratosphere towards the UT, CO decreases from the UT towards the tropopause (Figs. 2d, 3b). Lightning produces $NO_x$ in the ASMA (Fig. 3gh,
Appendix A) and some $NO_x$ of stratospheric origin gets entrained from the TL. CO and co-emitted precursors are uplifted from the lower troposphere. The net $O_3$ production is at maximum in the altitude range, where uplifted young air mixes with $NO_x$-rich UT air. It was shown in the accompanying paper that the simulated net $O_3$ production at 168 hPa is rather limited by the availability of CO than by too much or too less $NO_x$. However, $NO_x$ mixing ratios of 300 pmol/mol are abundant in the ASMA (Figs. 3ab), facilitating increased $O_3$ production.
Confinement in the ASMA circulation allows the mixed air to age, i.e. to produce $O_3$. CO is being depleted in the process of ageing, and $NO_x$ is transferred to $NO_y$. The other source of aged air is entrainment from the TL, which is however enriched in HCl. There is no appreciable HCl in uplifted air, and none is produced during ageing in the UT. The distinction between aged uplifted and entrained TL air is not always clear, since HCl in the UT is being depleted through oxidation by OH (see Appendix A).
We attribute some larger-scale trace gas gradients in the measurements to different photochemical $O_3$ production (section 4). There are positive correlations between NO & $O_3$ and NO & $NO_y$ (Fig. S9c and Fig. S10c), but not between CO & $O_3$ (Fig. 4c). This combination indicates that the measurements show a $NO_x$-limited photochemical regime.

  $O_3$ is produced in the ASMA at a net rate of about 4 nmol mol$^{-1}$ day$^{-1}$ (Fig. 2h). Simulated $O_3$ mixing ratios in the UT of the ASMA region vary by about 120 nmol/mol (Fig. 4b). It would take 30 days to cover that range by photochemical $O_3$
production alone. The observed values in the ASMA filament cover a range of about 48 nmol/mol (Fig. 4c), corresponding to 12 days of photochemical $O_3$ production. This is about the time needed to circle the ASMA (see accompanying paper). The above $O_3$ variability of course includes different amounts of $O_3$ from the TL, and $O_3$ productivity varies too. Ignoring these uncertainties, it takes one or two rotations of the ASMA to transform $O_3$-depleted, freshly uplifted air into aged, $O_3$-enhanced air. $O_3$-depletion in the ASMA relative to the regional average can only be maintained by frequent replenishment
of young air.





### 5.3 Radial stratification and patchy trace gas distributions

Deep convection from the lower troposphere discharges more towards the ASMA interior, as shown by studies that report relatively young air there (Li et al., 2005; Randel and Park, 2006; Park et al., 2008; Liu et al., 2009b; Kunze et al., 2010; Liu et al., 2011) and also by our simulation (Fig. 8). In contrast, trace gas signatures in a belt of outer streamlines are dominated by a combination of photochemically aged lower tropospheric air and entrainments of UT air surrounding the ASMA. In this schematic of an undisturbed anticyclone, interior trace gas signatures are generally characterised by lower $O_3$ mixing ratios than fringe signatures then.

Such radial zoning in the ASMA is an expression of almost closed circulation, and was observed in CARIBIC (http://www.caribic-atmospheric.com) in situ data of flights between Chennai, India and Frankfurt, Germany (Baker et al., 2011; Rauthe-Schöch et al., 2016). CARIBIC data show increased $O_3$ mixing ratios in the northern part of the ASMA, and decreased levels towards the southern end of the flights. This was tentatively attributed to differences in the time available for $O_3$ production there (Lawrence and Lelieveld, 2010), which is supported by corresponding differences in photochemical age (Baker et al., 2011; Rauthe-Schöch et al., 2016). Those observations are consistent with the above scenario of preferential replenishment of young air in the ASMA's interior, while the outermost streamlines are less affected by upwellings and photochemical ageing may proceed for longer. Furthermore, TL entrainment affects the ASMA fringe, enhancing the aged-air (increased $O_3$) signatures. It was even pointed out by Baker et al. (2011) that the aged-air signatures can also be explained by stratospheric contributions, which they could only rule out along the CARIBIC flight tracks through the north-western ASMA flank.

The general on-off nature of TL entrainment, upwellings from the lower troposphere and lightning leads to patches of air with different trace gas signatures in the UT ASMA circulation. Each of these patches might again receive contributions from any of the above sources. In principle all sorts of combinations are possible, generating heterogeneity. In contrast, mixing and photochemical ageing are homogenizing effects. In combination with closed streamlines the preferential positions of the different sources should print through as radial stratification in the ASMA. However, neither the HALO ESMVal measurements, nor the corresponding simulated snapshots or monthly mean values in the accompanying paper show a clear stratification. The idealised picture that the ASMA circulation is dominated by stationary, closed streamlines is certainly not realistic - at least not on the timescales of the homogenizing effects.

### 5.4 Splitting-up and stirring

Transient streamlines, particularly eddy shedding or splitting of the ASMA, effectively overcome radial transport barriers. Whether stratified or patchy – any trace gas distribution in the ASMA might be subject to effective stirring then. There is an ongoing discussion about different dynamical modes of the ASMA (Nützel et al., 2016; Pan et al., 2016). The study of Pan et al. (2016) distinguishes 4 phases: Tibetan plateau phase, Iranian plateau phase, longitudinally elongated phase, double centre phase.





A splitting-up event occurred just during the HALO ESMVal campaign, as consistently indicated by backward-trajectories, structure of the probed filament, and the simulated synoptic situation (see Fig. 9, Fig. 10, supplementary material and accompanying paper). It corresponds to the transition from an longitudinally elongated phase to a double centre phase in the nomenclature of Pan et al. (2016). We use the terms Iranian or Tibetan part/eddy/anticyclone to describe the splitting of one

big into two smaller anticyclones. A sequence of simulated streamlines and tracer distributions illustrates that event (Fig. 9). Although patchy, CO mainly marks the ASMA interior and HCl serves as a proxy to track the ASMA fringe. The sequence starts with an elongated anticyclone on 16 September 2012 (Figs. 9ab). Then a tropopause trough (marked as "$T_1$" in Fig. 9) evolves from the west along the northern ASMA flank. The elongated and thus inherently unstable anticyclone succumbs to the perturbation and splits up. A part of the initially compact increased CO interior region ("1") is entrained by the outer

streamlines of the Iranian part ("$1_f$"), while the rest of the patch is diverted into the interior of the Tibetan anticyclone ("$1_i$"). A patch of increased HCl ("2" in Fig. 9b) dominates the northern interior part of the intact ASMA at start, which is partly diverted into the interior of the Iranian ("$2_i$") and partly into the fringe and interior of the Tibetan part ("$2_{fi}$"). The majority of an HCl filament ("3") that marks TL entrainment is diverted to the Tibetan part. We also note entrainment of tropospheric air by southerly winds at the western flank ("4" in Fig. 9h).

The above example shows that fringing parts of an elongated anticyclone may become part of the interiors of both resulting anticyclones after the splitting. Interior parts may be diverted into the fringes. Even if not all possible cases are covered by the example, it is easily conceivable that fringing parts may also stay in the fringes and interior parts in the interiors. For a given location and timing of eddy shedding or splitting, the final trace gas distribution simply depends on the initial distribution of different patches. The redistribution of different parts of the anticyclone guarantees a high variability for

outflow and interior, whenever the closed circulation breaks down. How often does this happen?

Figure 10a gives an impression of the variability of the ASMA during 5 consecutive monsoon seasons at 168 hPa. The meridional wind fraction is a proxy for the zonal (E-W) location of the anticyclone(s). It is calculated as $v/\sqrt{u^2 + v^2}$, with meridional velocity $v$, and zonal velocity $u$. The pattern at the given altitude is consistent with one dominating anticyclone, centred at about 90°E. While the northerly winds east of 90°E are relatively persistent, episodes of entirely southerly winds

in the western part of the one-piece ASMA alternate with episodes of smaller, secondary anticyclones. Smaller anticyclones also occur regularly east of the Tibetan anticyclone, corresponding to eddy shedding to the East. The splitting event that occurred during the HALO ESMVal campaign is clearly visible in Fig. 10a, too. Such instabilities occur approximately twice a month. This coincidentally corresponds to the timescale needed to photochemically erase $O_3$-depleted signatures in young air masses, which have ended up in parts of the ASMA that do not receive additional entrainments.

Open questions remain with regards to the controls of ASMA instabilities and the stirring associated with other transitions between the different dynamical modes. Consistent to the discussion in section 1 we just note that there is an independently sustained, mid-tropospheric anticyclone over the Arabian peninsula (supplementary material, Fig. S14). Whether or not





coupling between the mid-tropospheric and UT systems plays a role for triggering ASMA splitting is not further analysed here.

**5.5 ASMA outflow**

It was shown in the previous sections that entrainment of TL air and dynamical instabilities of the ASMA occur quite
frequently in the simulation. Air is uplifted from the lower troposphere in the eastern part of the ASMA (indicated by CO: Fig. 2d) and a net $O_3$ producing photochemical regime prevails (Figs. 2gh), both throughout the monsoon season. Thus the main ingredients needed for the interpretation of our measurements are relevant beyond the HALO ESMVal campaign case. First we discuss some implications for the question of Lawrence and Lelieveld (2010), whether $O_3$ in the monsoon outflow is enhanced or depleted. This is relevant for understanding the observed mid-tropospheric summer $O_3$ maximum over the
Arabian Peninsula, i.e. one of the questions raised in section 1. ASMA contributions and transport pathways for the Arabian $O_3$ maximum have already been established (Liu et al., 2011; Richards et al., 2013; Garny and Randel, 2015) and are also seen in our simulation (section 3). Preferential export of fringe air with enhanced $O_3$ to the Arabian Peninsula is plausible because the secondary anticyclones or shed eddies tend to be smaller than the Tibetan anticyclone (Fig. 10a). The enhancement of fringe over interior air in the smaller eddies of ASMA outflow is a simple geometrical effect: the ratio of
circumference (fringe) to area (interior) is inversely proportional to the size (radius) of the vortex. Even the export of younger, precursor-rich air (Liu et al., 2009a; Richards et al., 2013) from the ASMA interior would contribute to increased $O_3$ within a week due to photochemical net $O_3$ production. Both scenarios are consistent with a climatologically decreased $O_3$ interior in the UT ASMA and at the same time increased $O_3$ over the Arabian peninsula. However, at least in the simulation transport of $O_3$ dominates over in situ production in the mid-troposphere of the Iranian part. Enhanced $O_3$ in the
ASMA outflow would follow even more straightforward from climatologically enhanced $O_3$ throughout the ASMA.
We note that the entrainment of $O_3$-poor air by northerly winds at the western ASMA flank ("4" in Fig. 9; supplementary material, Fig. S15) counteracts the $O_3$-enrichment in the western ASMA region. However, there is an asymmetry: processes increasing $O_3$ dominate in the eastern part of the ASMA, which dynamically dominates the western part in the UT.
Dynamical instabilities of the ASMA also have contributed to the enhanced $O_3$ ASMA interior simulated for mid-September
2012. Larger fractions of the $O_3$-rich fringe were entrained during splitting up events (supplementary material, Fig. S15). The sequence of snapshots (Fig. S15) covers almost half a monsoon season and episodic $O_3$-poor upwellings over the Tibetan plateau are smaller and shorter lived than $O_3$-rich regions at 168 hPa. This is consistent with the corresponding monthly mean distributions, which show increased $O_3$ in the ASMA (accompanying paper).
However, longer periods of stability seem to strengthen lower tropospheric signatures in the ASMA interior. It was found
that the residence time in the ASMA is crucial for the probability to enter the stratosphere (Garny and Randel, 2015). So this interrelation could lead to preferential transport of certain trace gas signatures from the ASMA into the stratosphere, but is not further analysed here.





## 6 Summary

This study complements a detailed analysis of in situ trace gas measurements in the ASMA, obtained during the ESMVal campaign with the research aircraft HALO in September 2012 (Gottschaldt et al., 2017). The measurements are put in the context of the EMAC simulated annual evolution of trace gas profiles in the ASMA region and simulated tracer-tracer

relations. This led to the following qualitative understanding of the interplay of processes that determine the trace gas distributions in the ASMA and its outflow (Fig. 11):

Air from the steeply inclined TL is entrained by outer ASMA streamlines at the eastern and possibly northern ASMA flank, defining a fringing zone. Tropopause troughs facilitate the entrainment.

The TL is characterised by a gradient between stratospheric and tropospheric trace gas signatures. Stratospherically

enhanced tracers like HCl and $O_3$ print through in the entrained air. Thus the fringe is not just a transport barrier, separating the ASMA interior from the respectively surrounding UT. It has a distinct genesis, resulting in air masses with distinct trace gas signatures that may be transported relatively unperturbed over long distances. Deep convection and the upwellings in conduit over the Tibetan plateau (Bergman et al., 2013) inject lower tropospheric air mainly into the Tibetan part of the ASMA. Enhanced CO is an indicator for this process. Convection is accompanied by in situ production of lightning $NO_x$,

mainly determining mixing ratios of this $O_3$ precursor in the ASMA.

Uplifted air preferentially feeds the ASMA interior, as also indicated by studies that report younger air there (Randel and Park, 2006; Park et al., 2007; Park et al., 2008; Kunze et al., 2010). In the idealised case of one intact anticyclone (Fig. 11a) the interior would then be dominated by photochemical ageing of those $O_3$-poor injections. Net $O_3$ production dominates in the ASMA, and is particularly enhanced where lower tropospheric $O_3$ precursors (VOCs) meet UT precursors ($NO_x$). The

preferential positions of convective versus TL entrainments facilitate radial stratification in the ASMA. The intermittent nature of the entrainments, combined with the varying position of the anticyclone lead to patches of air that have different origins and are in different stages of ageing. Mixing and ageing act homogenizing, but each of these patches might again receive fresh entrainments from the TL or by convection plus lightning.

Eddy shedding, splitting of the ASMA into an Iranian and a Tibetan part, or transitions between other (Pan et al., 2016)

dynamical modes effectively overcome radial transport barriers (Fig. 11b; see also Fig. 9 and supplementary material, Fig. S15). Whether stratified or patchy – any trace gas distribution in the ASMA is subject to effective stirring then. Fringe air can be diverted into the interiors of both anticyclones, and likewise interior air is redistributed throughout the UT in the monsoon region. Remnants of earlier such events gradually lose memory of their origins, leading to a mixed "background" (grey in Fig. 11b). It remains an open question, if different dynamical modes of the ASMA are preferentially related to

particular trace gas distributions.

By analysing 5 consecutive monsoon seasons in the EMAC simulation we found that the processes that led to the curious combination of both enhanced lower tropospheric and TL tracers in the ASMA filaments encountered by the HALO ESMVal campaign are not exceptional: entrainment of TL air and dynamical instabilities of the ASMA occur quite




frequently. Convection and thunderstorms are common throughout the monsoon season, accompanied by a net O₃ producing photochemical regime. The alternating interplay of those processes results in highly variable, patchy trace gas distributions in the ASMA. Processes that increase O₃ and O₃ precursors dominate in the Tibetan part of the ASMA. The Iranian part is dynamically dominated by the Tibetan part in the UT and this asymmetry tends to increase O₃ in the tropospheric ASMA

outflow, e.g. over the Arabian Peninsula.

**Appendix A: Lightning NOₓ in the ASMA**

As noted already in the main text, LNOₓ emissions play a major role in UT photochemistry of the ASMA region. At the same time those emissions are subject to considerable uncertainties (Schumann and Huntrieser, 2007). In our EMAC

simulations, LNOₓ is released based on a parameterisation that links flash frequency to updraught velocity in -also parameterised- convection (Grewe et al., 2001). For simulation RC1SD-base-10a the LNOₓ emission rate profiles in the eastern and western ASMA regions for 2012 are shown in Figs. 3gh. This is supplemented by Fig. A1, which shows the corresponding lateral distribution of monthly mean LNOx emission rates at the altitude of the HALO ESMVal measurements. We note that there are strong, but localised emissions in the Iranian and Tibetan parts in spring (Fig. A1:

Apr-May). LNOₓ emissions are distributed throughout the Tibetan region in summer (Fig. A1: Jul-Sep). The simulated spatio-temporal emission patterns are similar for 2013 and 2014 (not shown).

In order to link the LNOx emissions to the NOx burden in the ASMA region, a suite of EMAC sensitivity simulations with modified emission factors was conducted (Figs. A2, A3). All EMAC analyses in the main text are based on simulation RC1SD-base-10a (Jöckel et al., 2016), which is given in Fig. A2 just for comparison. The other simulations discussed in the

context of Figs. A2 and A3 here are derived from EMAC simulation RC1SD-base-10, which differs in road traffic emissions and optical properties of stratospheric aerosol (Jöckel et al., 2016) from RC1SD-base-10a. Total LNOₓ emissions in RC1SD-base-10 are 4.6 Tg(N) in 2012 (Jöckel et al., 2016), which is in the realistic range of 2 – 8 Tg(N) yr⁻¹ (Schumann and Huntrieser, 2007). RC1SD-base-10 and our base simulation for the LNOx sensitivity analysis (b01) are both operated in chemistry-climate model (CCM) mode, i.e. interactive chemistry with feedback on dynamics. Simulation b01 differs only in

the usage of daily (Kaiser et al., 2012) instead of monthly biomass burning emissions and 5 h instead of 10 h output intervals. Feedbacks from chemistry on dynamics in all quasi chemistry-transport model mode (QCTM) (Deckert et al., 2011) simulations are based on identical trace gas climatologies from b01. The same dynamics incl. convection is simulated in all QCTM simulations. Differences between a QCTM reference simulation (q01) and sensitivity simulations (s*) are thus exclusively due to chemical perturbations. All QCTM simulations cover June – September 2012, but the first 3 months were

discarded for spin-up.

Figure A2 shows that RC1SD-base-10a captures observed O₃, CO, NO and NOᵧ along the HALO ESMVal flight path slightly better than b01 and q01. We are yet confident that the overall agreement is good enough for the analysis of chemical



perturbations. For the QCTM sensitivity analyses it is more important to note that differences between b01 and q01 are negligible.

Figure A3k shows that halving $LNO_x$ emission factors results in almost halved $NO_x$ in the uppermost troposphere. Doubling of $LNO_x$ emissions leads to almost doubled $NO_x$ just below the tropopause (Fig. A3m). The biggest relative sensitivity in Fig. A3km almost coincides with the altitude range of the largest $NO_x$ mixing ratios just below the tropopause (Fig. A3j). Thus, in our simulations $LNO_x$ clearly dominates the $NO_x$ budget from the tropopause to 100 hPa below it. The impact of $LNO_x$ fades out at lower altitude, and almost vanishes at 400 hPa below the tropopause. This is consistent with the profiles of $LNO_x$ emissions in September 2012, which mainly occur in the Tibetan part of the ASMA (Fig. 3h).

Modifications of $NO_x$ print through on other $O_3$ precursors mainly via changes to the atmospheric oxidizing capacity (OH: Figs. A3ghi). In response to halved $LNO_x$, OH decreases 200 hPa below the tropopause and lower, and increases above (Fig. A3h). The effects are reversed for doubled $LNO_x$ (Fig. A3i). The largest relative effects coincide with largest absolute OH mixing ratios.

CO decreases throughout the shown altitude range for halved $LNO_x$ (Fig. A3e). Without major production terms in the UT, modifications to CO mixing ratios are dominated by the loss reaction $CO + OH \rightarrow H + CO_2$. The rate coefficient of this reaction is proportional to pressure, and otherwise depends only on constants (see supplement to Jöckel et al. (2016)). Laterally averaged CO mixing ratios vary little from 50 to 400 hPa below the tropopause (Fig. A3d), but are affected by decreased and increased OH (Figs. A3fg). Decreased OH in the lower half of the domain dominates the overall CO response. CO rises through this region with higher pressures before reaching the UT in the Tibetan part of the ASMA (Fig. 2), which obviously outweighs the CO response to increased OH in the UT of Tibetan and Iranian part combined. Increased OH 200 – 500 hPa below the tropopause consequently leads to an overall decrease of CO in response to doubled $LNO_x$ (Fig. A3f). The $O_3$ precursors $NO_x$ and CO display opposite trends in response to $\Delta LNO_x$.

Curiously, HCl shows the opposite response to modified $LNO_x$ (Figs. A3abc). There is no chemical production of HCl in the UT, and the only loss term in the simulations is $HCl + OH \rightarrow Cl + H_2O$. The rate coefficient of this reaction is 1.7E-12 * EXP(-230/Temperature), see supplement to Jöckel et al. (2016). However, the tropospheric response of HCl to $\Delta OH$ is dominated rather by the vertical profile of HCl mixing ratios than by lower temperatures towards the tropopause. Almost all HCl in the UT is of stratospheric origin, and HCl mixing ratios steeply increase across the tropopause. Thus the UT response of HCl is dominated by $\Delta OH$ near the tropopause: increased OH for halved $LNO_x$ increases HCl losses, and vice versa for doubled $LNO_x$.

The response of UT net $O_3$ production to $\Delta LNO_x$ (Figs. A3qrs) has mostly the same sign as $\Delta NO_x$. As noted already in the context of Fig. 2, opposite gradients of $O_3$ precursors $NO_x$ and CO in the UT lead to a broad altitude range of enhanced net $O_3$ production in the ASMA, centred about 100 hPa below the tropopause. $O_3$ production is limited by $NO_x$ in lower altitudes and by CO (and other volatile organic compounds) towards the tropopause. $NO_x$ and CO display opposite trends in response to $\Delta LNO_x$, but relative changes to $NO_x$ are larger and dominate the overall response of net $O_3$ production. We note however, that the largest increase of $NO_x$ at the end of September (circled in Fig. A3m) decreases net $O_3$ production to zero or even net





loss (circled in Fig. A3s). The opposite effect is not seen for halved $LNO_x$ (Fig. A3r). Thus it is most likely due to the non-linearity of net $O_3$ production, which decreases above a certain $NO_x$ concentration threshold that also depends on other parameters (Grooß et al., 1998).

$O_3$ mixing ratios respond to $\Delta LNO_x$ essentially like net $O_3$ production in the UT (Figs. A3nop). The altitude of maximum

relative $\Delta O_3$ is slightly lower than the altitude of maximum absolute changes to net $O_3$ production. We attribute this effect to upwards increasing absolute $O_3$ mixing ratios.

**Appendix B: Primer for tracer-tracer diagrams**

Sampling of two different air masses that are in the process of mixing is indicated by a mixing line in the tracer-tracer diagram. The slope of the mixing line provides additional clues about the origin of the original air parcels ("end-members").

If the ratios of the end members remain constant over time, the slope of the mixing line is conserved, as long as the mixing process continues. If the mixing processes stops, the mixing lines converge to a single point in the tracer-tracer diagram. If the reservoir of one end-member is bigger than the other, points in the tracer-tracer diagram will be close to the dominating end-member. However, the relative size of the reservoirs does not affect the slope of mixing lines, thus allowing detection of even small entrainments. Slopes change in case of mixing ratio changes over time (e.g. via in situ production or loss) of one

or both reservoirs. Different effects may lead to similar tracer-tracer relations, resulting in ambiguity when trying to reconstruct end-members or disentangling mixing and chemical effects. Furthermore, mixing lines in general exist in a multi-dimensional tracer space, and thus lines in a tracer-tracer plot need to be considered as projections onto 2d space. They might also be the result of mixing between more than two reservoirs. Additional dimensions (tracers) need to be considered to reduce ambiguities.

**Data availability**

The simulation results analysed here are archived at the German Climate Computing Center (DKRZ) and are available on request. It is planned to move them to the Climate and Environmental Retrieval and Archive (CERA) database at the German Climate Computing Centre (DKRZ; http://cera-www.dkrz.de/WDCC/ui/Index.jsp). The corresponding digital object identifiers (doi) will be published on the MESSy consortium web page (http://www.messy-interface.org).

The observational data of the HALO ESMVal flight used here are available from the HALO database: https://halo-db.pa.op.dlr.de/dataset/830. Registration is needed to access the data (https://halo-db.pa.op.dlr.de/account/register).





**Competing interests**

The authors declare that they have no conflict of interest.

**Author contributions**

K. Gottschaldt analysed the EMAC and final in situ data, conducted the Lagrangian calculations, produced the plots and
drafted the paper. H. Schlager conceived the study, led the HALO ESMVal campaign and interpreted EMAC and in situ
data. R. Baumann wrote and helped with the code that facilitated the HYSPLIT calculations. P. Jöckel led the ESCiMo
project, coordinated the preparation of and conducted the EMAC simulations. D. S. Cai and P. Graf prepared a significant
part of the boundary conditions, and V. Grewe was responsible for the ProdO3 and LossO3 diagnostics in the ESCiMo
simulations. V. Eyring conceived and led the ESMVal project. T. Jurkat, C. Voigt, A. Zahn, and H. Ziereis supplied in situ
measurements. All authors contributed to the text.

**Acknowledgements**

The authors gratefully thank H. Garny and P. Hoor for valuable comments on the manuscript, A. Baker, B. Barret, B. Brötz,
F. Frank, H. Huntrieser, R. Müller, M. Nützel, L. Pan, G. Stiller, and B. Vogel for helpful discussions.
We thank the German Science Foundation DFG for funding within HALO-SPP 1294 under contracts JU 3059/1-1, SCHL
1857/2-2, SCHL 1857/4-1, VO 1504/2-1 and VO 1504/4-1. The HALO ESMVal aircraft campaign was funded by the DLR-
project ESMVal. KG and HS appreciate support by the EU project StratoClim (grant no. 603557) and BMBF project Spitfire
(grant no. 01LG1205B). CV and TJ thank financing by the Helmholtz Association under contract no. VH-NG-309 and under
contract No. W2/W3-60. In addition we thank the flight department of DLR for their great support during the campaign. P.
Hoor and S. Müller contributed to the CO measurements and S. Kaufmann supervised the HCl measurements during the
flight.
The EMAC model simulations were performed at the German Climate Computing Centre (DKRZ) through support from the
Bundesministerium für Bildung und Forschung (BMBF). DKRZ and its scientific steering committee are gratefully
acknowledged for providing the HPC and data archiving resources for the projects 853 (ESCiMo - Earth System Chemistry
integrated Modelling) and 854 (ESMVal).
We used the NCAR Command Language (NCL) for data analysis and to create some of the figures of this study. NCL is
developed by UCAR/NCAR/CISL/TDD and available on-line: http://dx.doi.org/10.5065/D6WD3XH5.
The article processing charges for this open-access publication were covered by a Research Centre of the Helmholtz
Association.





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





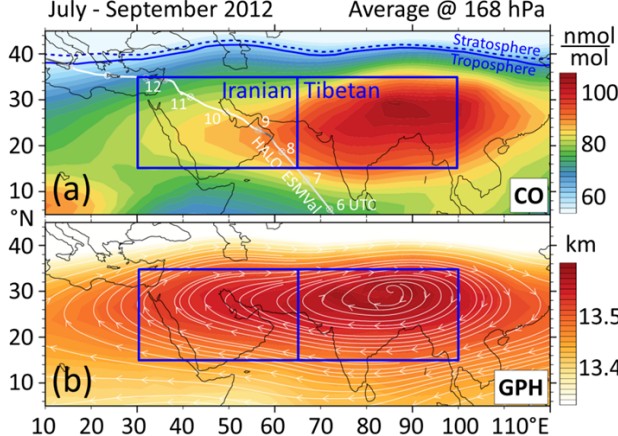

**Figure 1. CO mixing ratios and geopotential height (GPH) as simulated by EMAC at 168 hPa, averaged for the monsoon months of 2012. Enhanced CO is considered to be a chemical characteristic of the ASMA (Pan et al., 2016), and increased GPH is a dynamical proxy (Barret et al., 2016). The Iranian and Tibetan domains correspond to the regions used for lateral averaging throughout the paper (e.g. Figs. 2 - 5, S4 – S10). The Iranian region was traversed by the HALO ESMVal campaign during a flight from Male (Maldives) to Larnaca (Cyprus) on 18 September 2012. HALO was flying in the upper troposphere where the flight track is coloured white and dived to the lower troposphere where it is grey. Beads show the HALO positions at full UTC hours. Panel a additionally shows the intersection between the tropopause and the 168 hPa pressure level, panel b additionally shows streamlines.**





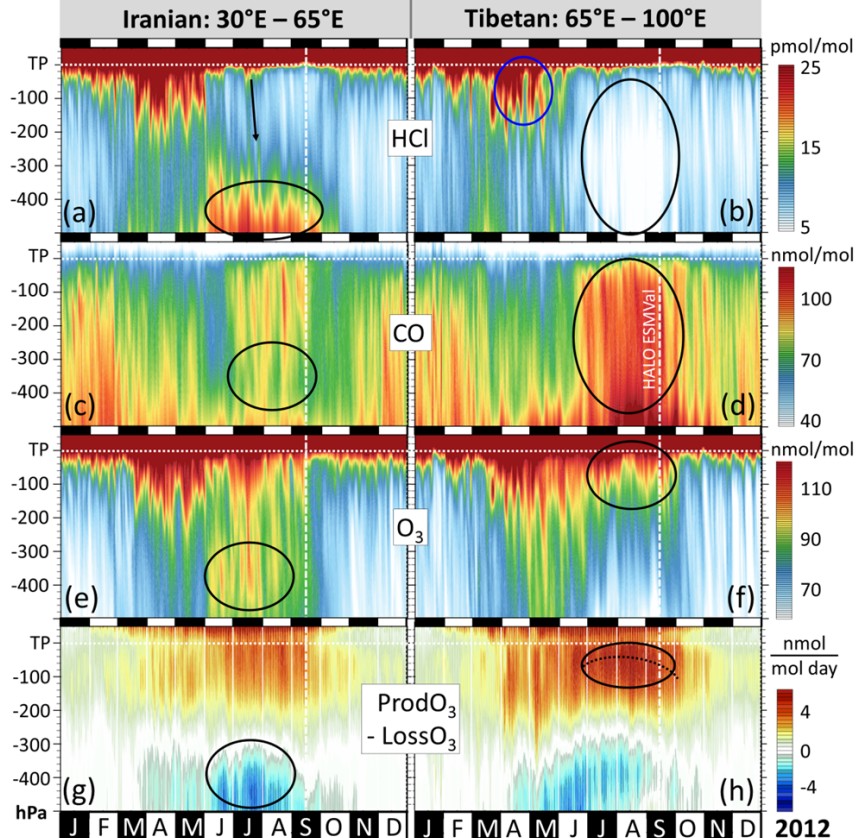

**Figure 2.** Evolution of simulated trace gas profiles and related diagnostics throughout 2012 in the UTLS and middle troposphere of the ASMA region. The time of the HALO ESMVal measurements is indicated by a dashed line. Vertical coordinates are given as pressure distance to the tropopause ("TP"), whose altitude depends on time and location. All values are grid-cell dry air mass weighted averages from 15°N to 35°N, respectively for the western (30° - 65°E) and eastern (65° - 100°E) parts of the ASMA (see Fig. 1). See text for details of the calculation of O$_3$ net photochemical production, and for the discussion of features marked by circles and arrows.





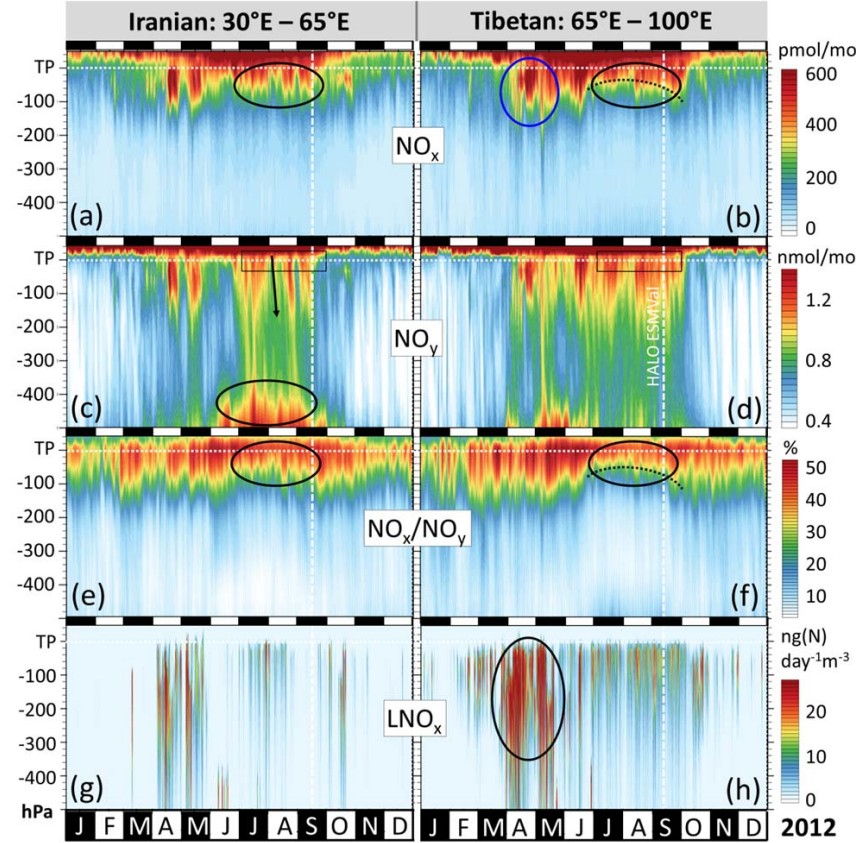

**Figure 3. As Fig. 2, but for different aspects of reactive nitrogen. See Appendix A for details concerning lightning NO$_x$ (LNO$_x$), and the supplementary material for a zoom into the regions indicated by black rectangles in panels c and d.**





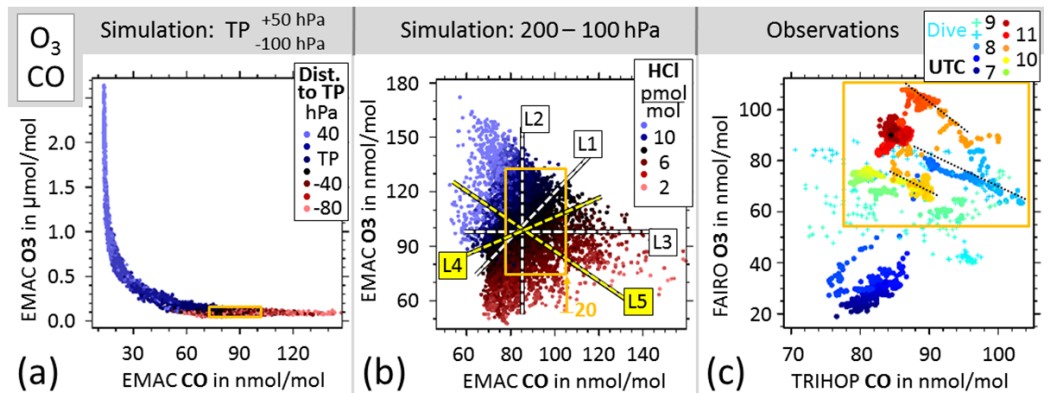

**Figure 4. Tracer-tracer relations (CO vs. O₃) as simulated by EMAC for the entire September 2012 in the ASMA region, and as observed by HALO during the HALO ESMVal campaign on 18 September 2012. (a) Simulated samples from the region 15°N – 35°N, 30°E - 100°E. Colour coding corresponds to the pressure distance to the tropopause, from 100 hPa below to 50 hPa above.**

10 **(b) Simulated tracer mixing ratios from the same region, but limited to tropospheric cells in the pressure altitude range 200 – 100 hPa. Colour coding indicates corresponding mixing ratios of HCl. See text for details of hypothetic lines L1 - L5. (c) Observed tracer mixing ratios of the HALO flight from Male to Larnaca (POI2 to POI6). Colours correspond to the UTC time of measurement, also indicating spatial proximity. The orange boxes show the ranges covered by the measurements from within the ASMA.**



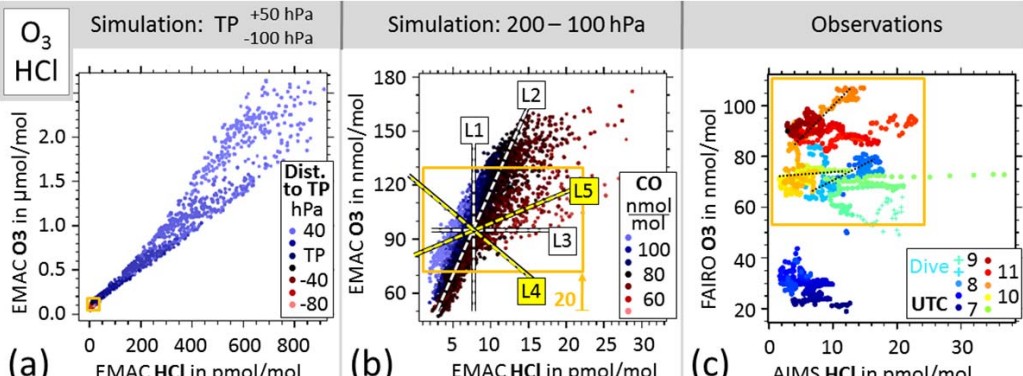

**Figure 5. As Fig. 4, but for HCl vs. O₃.**



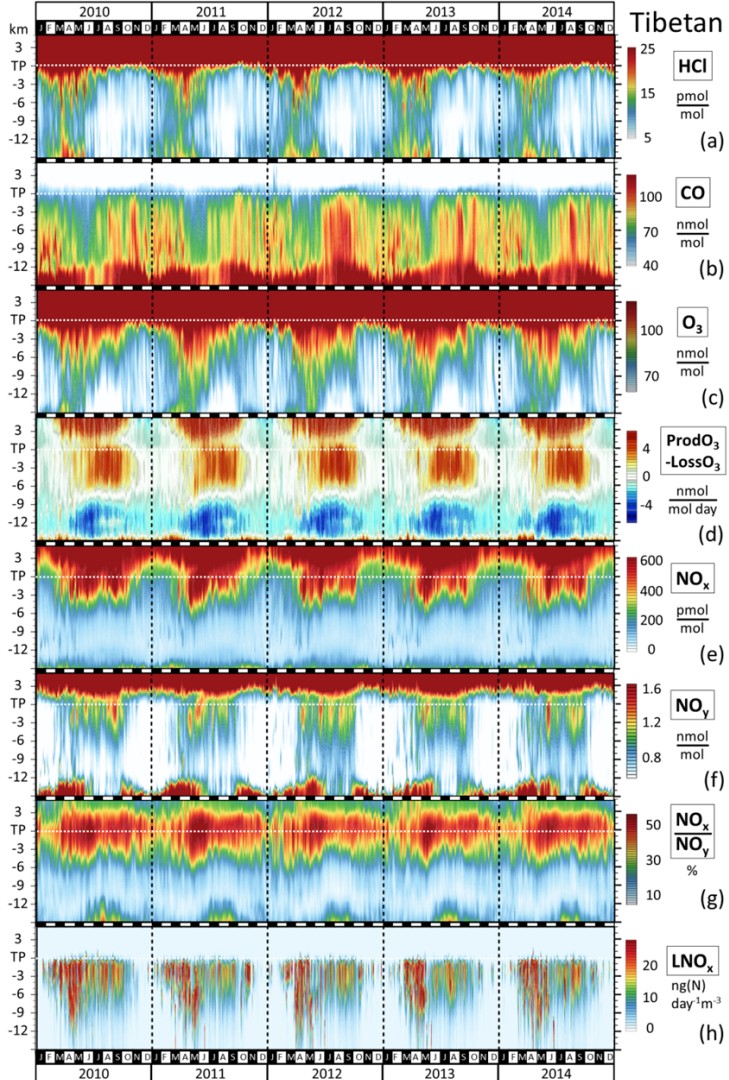

**Figure 6.** Evolution of simulated trace gas profiles and related diagnostics for the years 2010 - 2014 in the Tibetan ASMA region (65° - 100°E, see Fig. 1). Vertical coordinates are given as distance to the tropopause ("TP"), whose altitude depends on time and location. All values are grid-cell dry air mass weighted averages from 15°N to 35°N. The column for the year 2012 is identical to the corresponding panels in Figs. S4, S5. The corresponding figure for the Iranian region is shown in the supplementary material (Fig. S18).



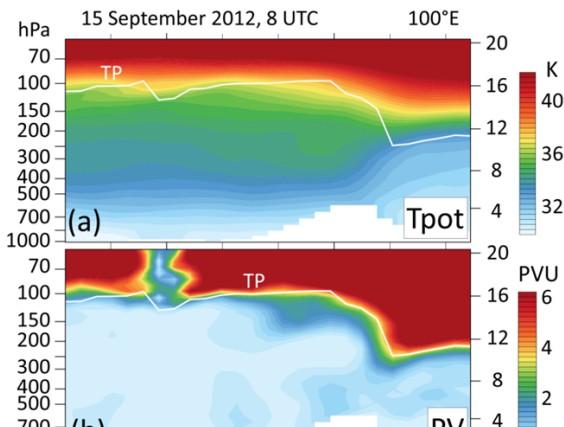

**Figure 7. EMAC simulated potential temperature and potential vorticity in curtains at 100°E, at the time when the air corresponding to POI3 was passing there at about 165 hPa. Note the steeply inclining TP over the Tibetan plateau, which marks the transition from the extratropics (dominated by baroclinic wave activity and downward stratospheric circulation) to the tropics (dominated by radiative-convective balance and upward stratospheric circulation). North of 30°N the EMAC TP is defined by the 3.5 PVU isocontour. Isentropes intersect the inclining TP, thereby allowing cross-TP transport without leaving a tell-tale signature of increased Tpot in the corresponding air masses in the tropics. This includes the 350 – 370 K isentropes that were encountered during the HALO ESMVal campaign in the tropics (see also Figs. S11, S12).**




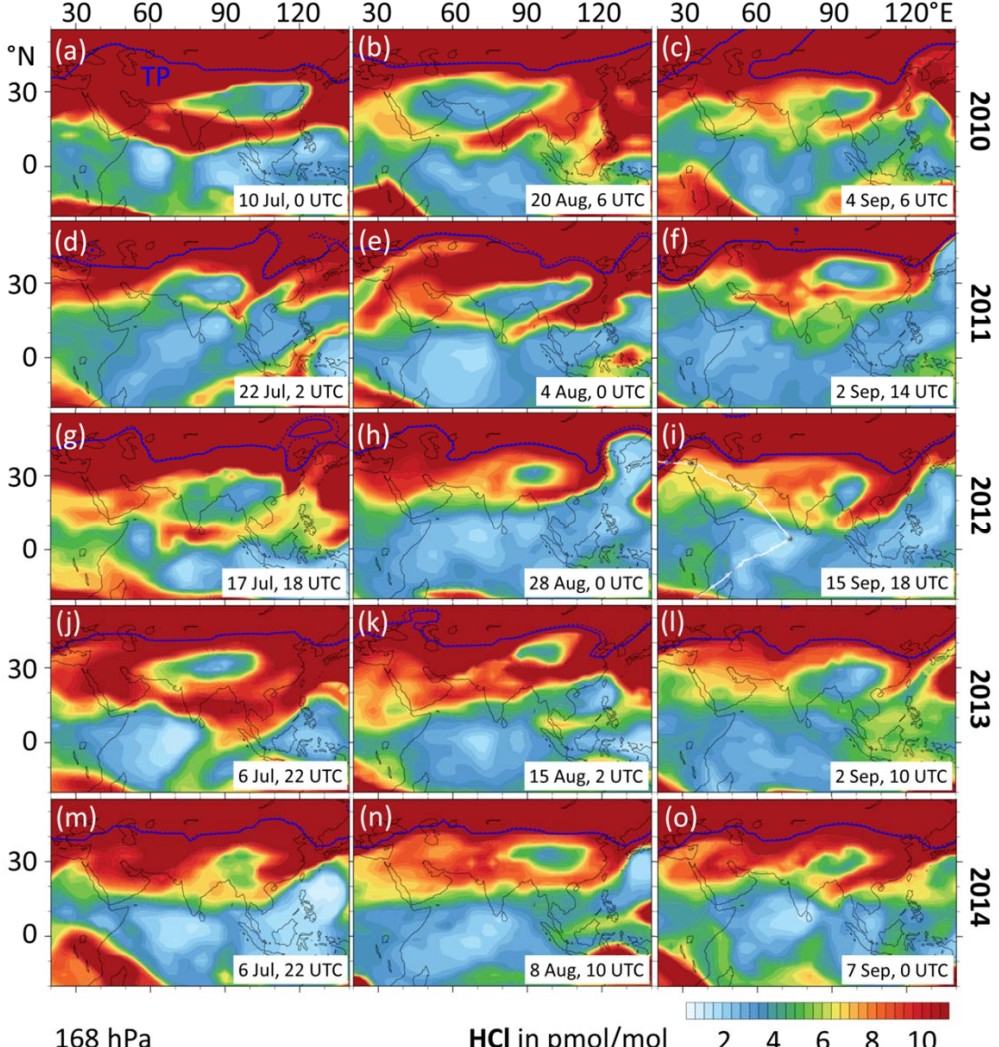

**Figure 8.** EMAC simulated HCl mixing ratios at 168 hPa in the ASMA region, complementing Fig. 10. The snapshots were selected to represent independent situations, where the southern ASMA fringe is marked by a filament of enhanced HCl. The filaments are often associated with a TP trough at the eastern ASMA flank. Enhanced HCl serves as a proxy for TL or stratospheric air.



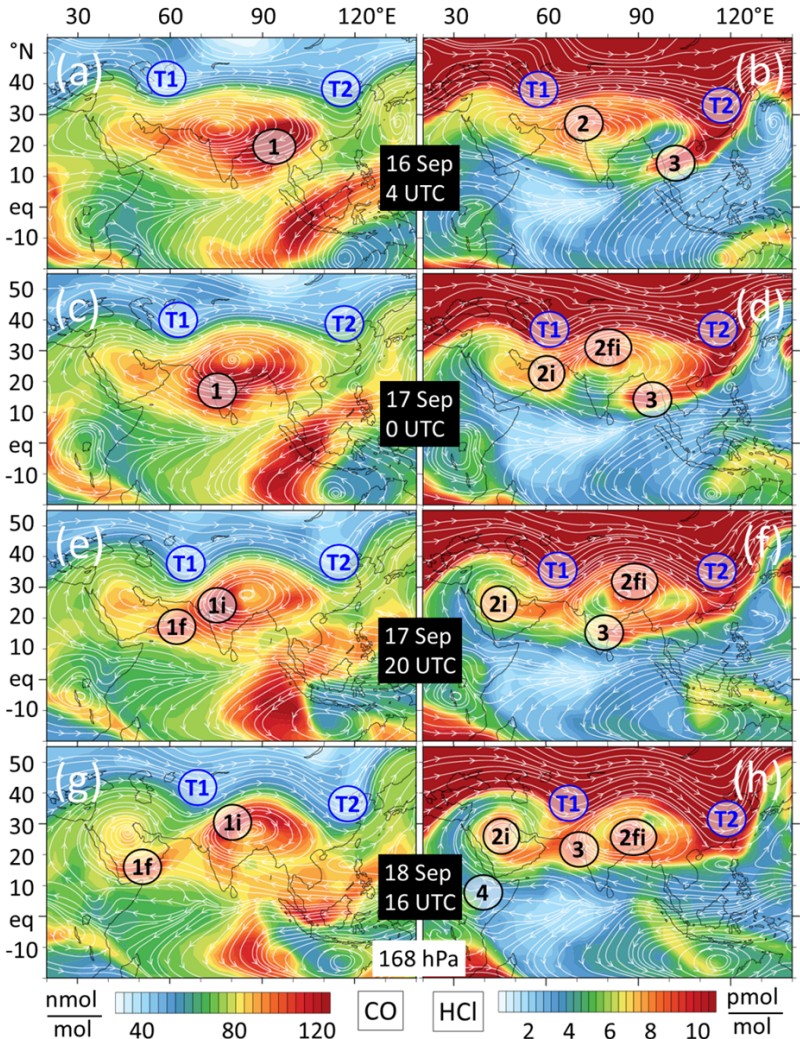

**Figure 9.** Sequence simulated tracer fields at 168 hPa, illustrating the stirring associated with the splitting-up event of the ASMA that occurred during the HALO ESMVal campaign in September 2012. It starts with a single large anticyclone, and ends with a Tibetan and an Iranian part, shortly after the HALO flight from Male to Larnaca had passed through. Streamlines represent instantaneous wind fields. Circled annotations mark some features discussed in the text.



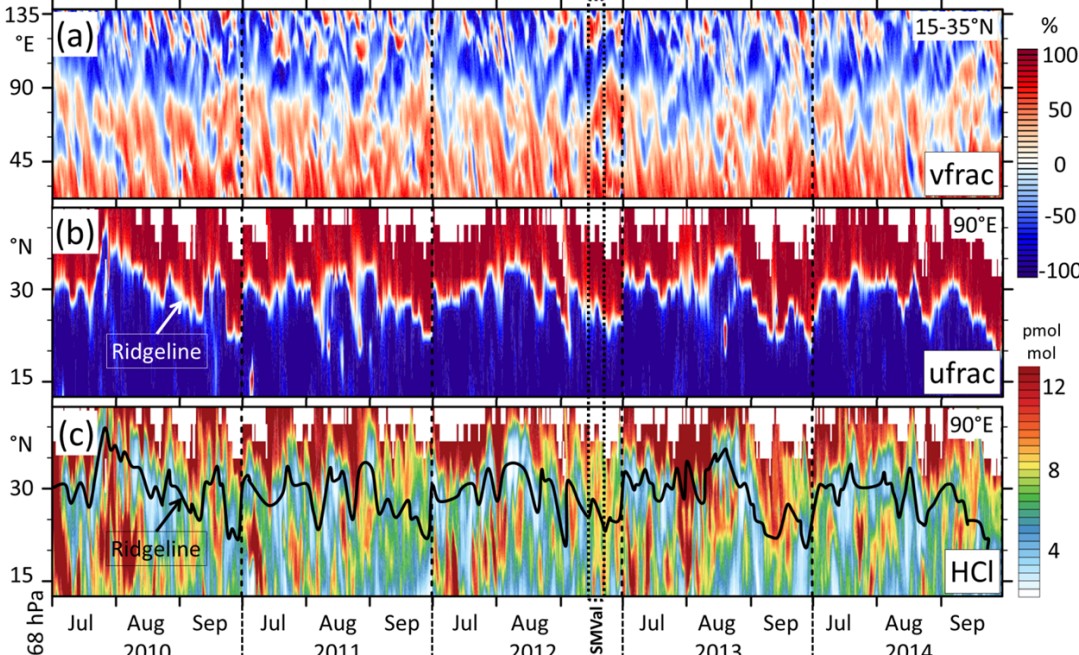

**Figure 10. Exploratory analyses of the frequency of occurrence of two aspects discussed for the HALO ESMVal campaign case, which are relevant for the (re-)distribution of trace gases in the ASMA region: (1) Splitting, eddy shedding/remerging of the ASMA; (2) Transport of TL air in the free troposphere along the southern ASMA fringe. All panels show EMAC simulation**

5     **results at 168 hPa, for the summer monsoon months in the ASMA region. The analyses are based on 10 h output steps. Grid cells not entirely within the troposphere are ignored. (a) Meridional wind fraction along a wide zonal transect, averaged with dry grid cell mass weighting at each longitude from 15°N to 35°N. Blue shades indicate southward and red shades northward winds. Each red-blue pair (from west to east) at a given time marks an anticyclone or a smaller eddy. (b) As panel (a), but for zonal wind fraction along a meridional transect at 90°E. Blue shades indicate eastward and red westward winds. (c) Time evolution of HCl**

10     **mixing ratios. At any given time, locally increased HCl south of the ridgeline is a proxy for air from the TL or the stratosphere.**





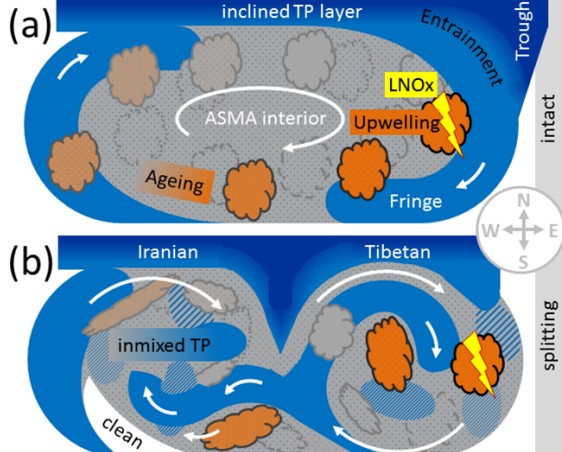

**Figure 11. Schematics of processes determining trace gas distributions in the ASMA at an UT pressure level: (a) One undisturbed anticyclone, encompassing the Tibetan and Iranian regions; (b) Splitting into an Iranian and a Tibetan part. See text for details.**





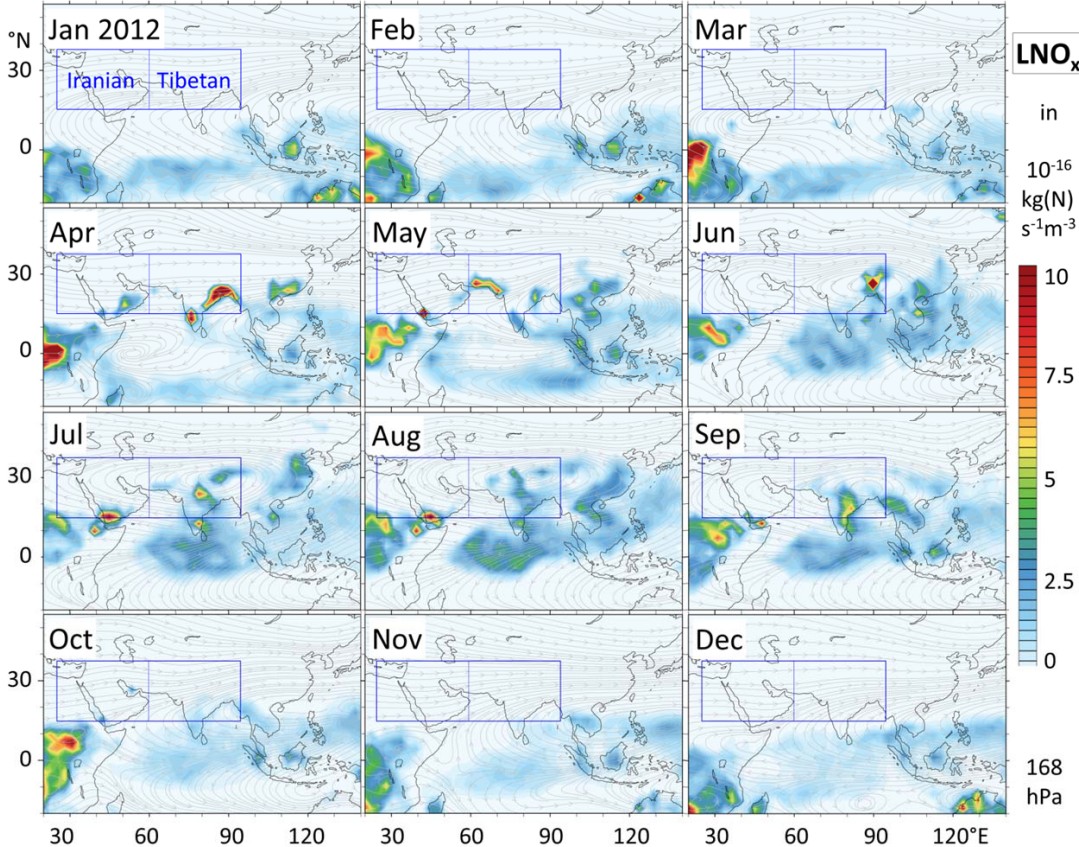

**Figure A1. Monthly mean lightning NO$_x$ emissions in 2012 at 168 hPa, based an EMAC simulation RC1SD-base-10a. Blue rectangles indicate the region(s) on that lateral averaging in Figs. 3 and A3 is based on. The ASMA circulation prints through in the monthly mean wind fields from June to September, as shown by streamlines (grey).**



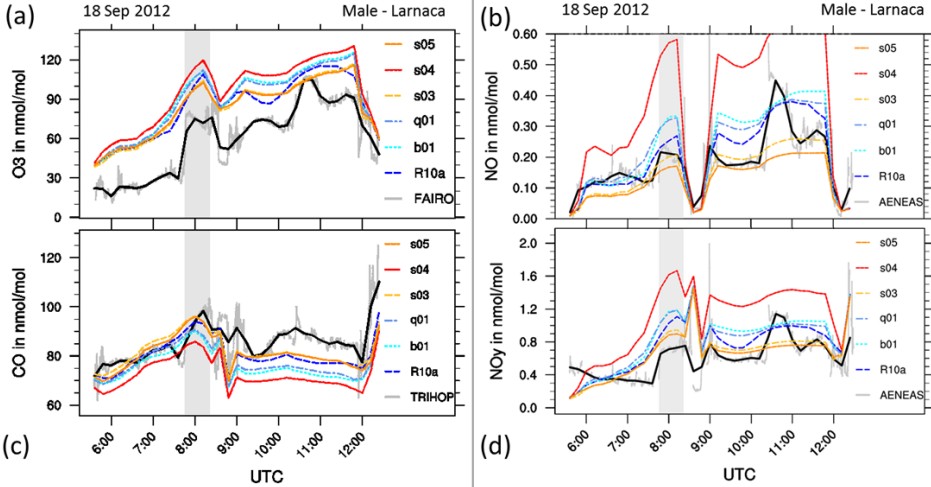

**Figure A2.** Mixing ratios of O₃, CO, NO and NOᵧ along the HALO flight track from Male to Larnaca, on 18 September 2012. Grey shading marks POI1. Grey line: in situ measurements in 10 s resolution, black: in situ averaged to 12 min simulation time steps, R10a: EMAC simulation RC1SD-base-10a. Sensitivity simulations are based on the almost identical RC1SD-base-10 simulation of (Jöckel et al., 2016), feature daily instead of monthly biomass burning emissions, and were performed in quasi chemistry transport model mode (Deckert et al., 2011) to facilitate isolating the effects of modified emissions.

b01: as R10a, but with different traffic and different biomass burning emissions;

q01: as b01, but QCTM;

s03: as q01, but halved LNO$_x$ emissions;

s04: as q01, but doubled LNO$_x$ emissions;

s05: as s03, but with a different vertical emission profile of LNO$_x$ (emission factors not decreased in the mid-troposphere, i.e. no C-shape)



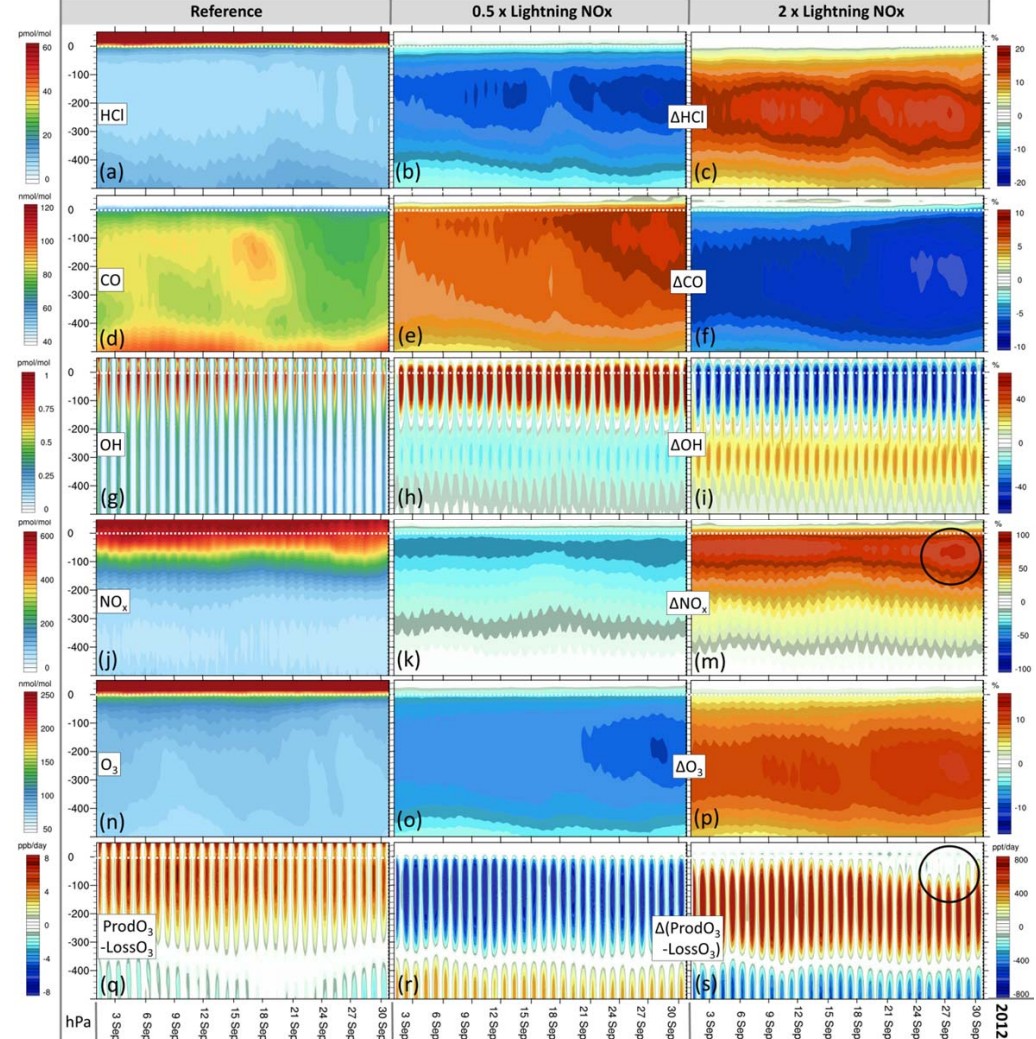

**Figure A3.** Evolution of simulated trace gas profiles and related diagnostics during September 2012 in the ASMA region (15° N – 35° N, 30° E – 100° E), and their sensitivity to LNOx emissions. The vertical axes cover the UTLS and middle troposphere, and their coordinates are given as pressure distance to the tropopause. Left column: QCTM reference simulation (q01). Middle column: s03 – q01, relative deviation of sensitivity simulation s03 wrt. q01 for trace gases, absolute deviation for net O₃ production. Right column: s04 – q01.