# Peer review of "Dynamics and composition of the Asian summer monsoon anticyclone"

_Atmospheric Chemistry and Physics, 2017_

## Referee Comment (RC1) · Anonymous Referee #2 · 27 Jul 2017

**General:**
This is very interesting and important paper which is worth to be published. The most important finding is the explanation of high ozone within the Asian summer monsoon anticyclone. The authors show that photochemical ozone production in the circulating air masses as well isentropic in-mixing from the stratosphere are key processes defining ozone. In this picture the anticyclone can be understood as a photochemically active and not well-isolated reactor. In this reactor there are two parts: "convectively driven" eastern (Tibetian) part and more "chemistry driven" western (Iranian) part. Although the paper is well-written, it suffers from an inadequate presentation (see major points). Because of this, the paper needs a major revision.

**Major points:**

[Figure]

1. As you show in many places, isentropic mixing between the stratospheric air (you call it TP layer) and the interior of the anticyclone is an important process in your chain of arguments. In Fig. 7 you show how such isentropes connecting the extratropical lower stratosphere intersect the tropopause (some almost perpendicular) and penetrate into the anticyclone itself. The mixing (stirring) happens on such isentropes and is almost a 2d process. So I do not understand, why you do not show the respective tracer distributions at such isentropes. I guess $\theta = 360$ K would be the right choice instead of using the 168hPa level (e.g. Fig. 8, Fig 9 or Fig. 10). I would recommend to show Fig. 7 much earlier in the text (e.g. as the second figure of your paper) and than use much more isentropic analysis. As you mentioned, such isentropes are tilded in pressure space, but transport occurs much more on such isentropes.

2. For me it is unreasonable to include 16 figures into the supplement! If your story exceeds something like 10-12 main figures and 2-5 figures of your appendix, you should divide the story into two parts or make your story shorter. The last point seems for me to be more your case. Your abstract roughly describes your main results (see also my general comments). So maybe, you can go through the text and remove everything what is not supportive for your main results (see also my minor points).

**Minor points:**

1. P 1/L 18-19
   "contrasted by...in autumn and winter" - ASM anticyclone does not exist in autumn and winter. Why we should talk about it.

2. P 1/L 20
   "is regularly entrained a the eastern flank" - This is the isentropic in-mixing mentioned in my major point and not correctly described in your paper

3. P 1/L 24
"by northerly" - I think "by southerly"

4. P 1/L 24
"Although..." - this sentence is not clear for me. I would remove it

5. P 2/L 11
I think that also "the eastward propagation of eddy shedding" is important now (Dethof et al., 1999; Vogel et al., 2014).

6. P 2/L 15
"the associated heat low" - do not understand what you mean

7. P 3/L7-34
I would recommend to focus the attention of the reader on ozone (observation very high in the core, but why, you will discuss it in the paper, also in the inter-annual context, etc). Instead of this you talk here too much about general aspects...

8. P 5/L1-6
For me this is the main motivation for the paper and it should roughly replace the part in P 3/L7-34 !

9. P 5/L14-16
"The extratropics are dominated..." - this sentence is unnecessary.

10. P 5/L24
...dominate the averages in the chosen regions.

11. P 5/L29-32
too much. I can only recommend to remove this material

12. Figure 1, caption
dynamical proxy of what.... I do not see any gray parts of the flight. "Panel a additionally shows..."????

13. P 7/L21-24
"slowly descending HCl..." - this feature is very strange. Typically, during the considered season (JJA) there is a strong diabatic upwelling in the UTLS region confined by the anticyclone. Maybe you should explain it with model or remove it...

14. P 8/L7
"...differ between the summer monsoon season and the rest of the year.." - I would say the strong difference is during AMJJA and not only during JJA

15. P 8
The main part of $NO_y$ in the stratosphere should be $HNO_3$, so I expect a much stronger correlation with HCl. Please comment.

16. P 8-9
Section 3.4 contains for me too much information. I would reduce it by considering only the ozone-relevant $NO_x$, $NO_y$ features.

17. P 10, L5
NOx, typo

18. P 10
Section 4 is a very important and novel part of the paper. It combines in situ observations (tracer-tracer correlations) with the model. It shows in a very nice way the interaction between the photochemical ozone production and stratospheric in-mixing. Because it does not use so much the observed $NO_x$, $NO_y$ features, it is the next motivation to shorten section 3.4.

19. P 12, L23
   "because HCl ...are decreased" - with the vertical mixing lines you argue that HCl should be constant. Maybe you should reformulate

20. P 13, L20-31 and P 14
   Here you show how important is the isentropic transport (mixing between the stratospheric and tropospheric air) on tilted isentropes. Here is also the origin for my major points.

21. On the following pages there are to many references to the supplement (see my major point) I can only recommend to shorten the following sections.

---

## Referee Comment (RC2) · Anonymous Referee #1 · 31 Aug 2017

Review of "Dynamics and composition in the Asian monsoon anticyclone" by K.D. Gottschaldt et al.

This paper is dedicated to the better characterization of the different processes (STE, convective uplift, dynamics, photochemistry...) that control the composition and specifically the O3 distribution in the Asian Summer Monsoon Anticyclone (ASMA). It is based on (i) in-situ observations taken on board the HALO research aircraft during the ES-MVal field experiment in September 2012 to document one particular case (ii) multi-annual simulations from the EMAC global chemistry climate model to document intra seasonal variabilities. The ASMA is a major feature of the Northern Hemispheric atmosphere during the monsoon season and its composition controls large parts of the tropospheric composition (such as over the Pacific and Middle East) during the mon-

soon season. The paper presents observations and model simulations that provide new insights about the contribution of different processes to the O3 distributions in the ASMA. It is well written and well organized and its argumentation is based on a robust methodology. I therefore recommend this paper for publication after a few important comments (one about the presentation of the results and one about LiNOx) and a number of rather minor comments are taken into account.

Major comments:

Presentation: - parts of the paper are too long and divert the reader from the main and strong elements brought by the paper. In particular, some rather general and introductory statements are given all along the manuscript (about the emissions, chemistry and dynamics...). It could be good if the authors try to shorten the paper and keep introductory elements to the introduction. Some examples of lengthy parts are mentioned in the detailed comments. - the paper is very often referencing to results from its accompanying paper which makes the reading and understanding somehow difficult. One example about O3 production regime is given below. - the same is true concerning the supplementary material which makes the paper a bit heavy to handle.

LiNOx and O3 production: Fig. 3h displays higher LiNOx production from EMAC in the Tibetan part of the ASMA during spring than during summer as mentioned P10L4-5. Nevertheless, the net O3 production is larger in summer than in spring down to 200 hPa below the tropopause (Fig. 2h). The authors explanation is that (i) in spring lower COV are uplifted by convection resulting in COV limitation and reduced O3 production (ii) LiNOx are produced locally in spring and not in summer. The latest argument also appears in the Annexe about LiNOx (p20L14-16).

Concerning (i) 1/ LiNOx production is linked to deep convection, especially in the models where both parametrization are coupled (in EMAC flashes are linked to convective updraught velocity as mentioned P20L10-11). Therefore more LiNOx should be associated with larger uplift of pollutants. Why EMAC displays more LiNOx with less uplifted

COV in spring ? 2/ Over South and East Asia the season of largest deep convection takes place in summer during the monsoon rather than in spring. Why are there more LiNOx in spring in EMAC ?

Concerning (ii): Looking at Fig. A1 displaying monthly LiNOx at 168 hPa, we see that they are localized over NW and NE India and Pakistan in May while in August they are more over SE India and Himalaya/Tibet. Nevertheless, the source is much stronger in spring. In both cases, the LiNOx emissions are very "patchy" and localized with also in June a single large emission spot over Bangladesh and in July the LiNOx spot localized over northern central India. Therefore, it is difficult to attribute a lower O3 production to more localized LiNOX emissions in spring. Why are the LiNOx emissions so "patchy" on a monthly scale? The averaging should smooth horizontally the distributions because convection does not always occur at the same place. It could be interesting to compare LIS/OTD distributions of lightnings to EMAC LiNOx distributions.

Finally, in Barret et al. (2016) the LiNOx are not shown but a sensitivity test shows that O3 and NOx produced by LiNOx are the highest during the monsoon season which seems rather logical for the reasons discussed above. This discrepancy between the EMAC and GEOS-Chem models concerning LiNOx should be discussed.

The east-west O3 net production gradient is logical as explained in the manuscript (P10L11-12) and in agreement with previous comparable evaluations by e.g. Liu et al. (JGR,2009) and Barret et al. (ACP,2016). Furthermore, the values of Fig. 2 seems in rather good agreement with those of the above mentioned papers for the monsoon season. A comparison and discussion of the EMAC O3 production with these previous studies could be interesting to strengthen and put the results in perspective.

Referring to the accompanying paper, it is mentioned that O3 production at 168 hPa is rather limited by CO than by NOx (P15L12-13). The exact sentence in the accompanying paper is "Net O3 production seems to depend more on CO (and related precursors) than on Nox". The use of "seems" shows that the authors are rather uncertain. I do not

really understand how is this possible because of the rather high CO concentrations within the ASMA (70-100 ppbv according to HALO and all references cited in the paper). This statement of a generally CO-limited regime in the ASMA at 168 hPa needs to be demonstrated and is not supported by the literature. For instance, according to Brune (IGAC Isuue 21, 2000), in the upper troposphere, the O3 regime is NOx limited for NOx concentrations lower than some hundreds pptv.

Tracer-tracer relationships:

This part is very interesting because the 3 tracers document different transport and chemical processes. HCl in particular which is rarely used is good to trace stratospheric air because O3 is photochemically produced in the troposphere.

The authors explain that "mixing line with negative CO/O3 slope dominate" in Fig 4c (corresponding to positive slopes in Fig. 5c). They correspond to mixing stratospheric air with photochemically processed tropospheric air. Nevertheless, in Fig. 5c, we also see some horizontal mixing lines (red and green). According to the discussion in p11 and 12 they correspond to mixing of fresh uplifted pollution (increasing CO, horizontal lines not so clear in Fig. 4c) and stratospheric air (increasing HCl in Fig 5c) with antagonist effects on O3. Another line with O3 decrease / HCl increase in Fig. 5c and O3 decrease / CO increase in Fig. 4c corresponding to mixing of fresh pollution in the UTLS can also be isolated. It is difficult to see whether these mixing lines correspond to important part of the sampled air masses but they could be mentioned.

Details: Part 3: the description of the different species at beginning of §3.1; 3.2, 3.3 and 3.4 (origin, chemistry etc.)  are too close to "textbook" descriptions and should be shorten for readability.  Part 5: this part tries to describe the different processes that control the ASMA composition one by one. It is interesting and well documented but rather lengthy and descriptive. For instance, the description of the evolution of the CO and HCL distributions P17L1-15 is very detailed and could be summarized. Fig2 and 3: the plots are shown in pressure coordinates that makes the region around the

tropopause very compact. Readability would be better in logP coordinates (altitude plots are provided in the supplement but it makes the reading uncomfortable and could be simply removed). p5l13: why choose PV = 3.5 for the tropopause in the extratropics ? Most studies choose 2 or 1.5. P5l27: Fig.1 is referenced after Fig 2 and 3. p9l5: "indicates that relatively..." P17L1-14: the dynamics are very detailed with the evolution of the air masses but is it really necessary to give so much details?

---

## Author Comment (AC1) · 12 Nov 2017

**Reply to Anonymous Referee #2**

We greatly appreciate those well-meaning comments, and thank the reviewer for his/her efforts to provide detailed suggestions to improve the paper.

Major comments:

*General:*
*This is very interesting and important paper which is worth to be published. The most important finding is the explanation of high ozone within the Asian summer monsoon anticyclone. The authors show that*
*photochemical ozone production in the circulating air masses as well isentropic in-mixing from the stratosphere are key processes defining ozone. In this picture the anticyclone can be understood as a photochemically active and not well-isolated reactor. In this reactor there are two parts: "convectively driven" eastern (Tibetian) part and more "chemistry driven" western (Iranian) part. Although the paper is well-written, it suffers from an inadequate presentation (see major points). Because of this, the paper*
*needs a major revision.*

*1. As you show in many places, isentropic mixing between the stratospheric air (you call it TP layer) and the interior of the anticyclone is an important process in your chain of arguments. In Fig. 7 you show how such isentropes connecting the extratropical lower stratosphere intersect the tropopause (some almost*
*perpendicular) and penetrate into the anticyclone itself. The mixing (stirring) happens on such isentropes and is almost a 2d process. So I do not understand, why you do not show the respective tracer distributions at such isentropes. I guess $\theta$ = 360 K would be the right choice instead of using the 168hPa level (e.g. Fig. 8, Fig 9 or Fig. 10). I would recommend to show Fig. 7 much earlier in the text (e.g. as the second figure of your paper) and than use much more isentropic analysis. As you mentioned, such*
*isentropes are tilded in pressure space, but transport occurs much more on such isentropes.*

The 168 hPa pressure level was chosen for analyzing the HALO ESMVal flight segment, which is the focus of the accompanying paper. Back-trajectories from the flight path showed that the ASMA circulation roughly stayed at that pressure altitude in that case.
However, we agree that isentropes are better than pressure levels for the current, more general study and will revise the manuscript accordingly.

*2. For me it is unreasonable to include 16 figures into the supplement! If your story exceeds something like 10-12 main figures and 2-5 figures of your appendix, you should divide the story into two parts or*
*make your story shorter. The last point seems for me to be more your case. Your abstract roughly describes your main results (see also my general comments). So maybe, you can go through the text and remove everything what is not supportive for your main results (see also my minor points).*

We agree that the main text should be shortened to better convey the main results and we thank the
reviewer for the corresponding detailed recommendations. Some figures have already become obsolete during the revision process. Some aspects (LiNOx, O3 photochemistry) need to be clarified, others become less important. However, the final shape of the paper will depend on the outcome of this discussion. Therefore we are cautious singling out individual paragraphs for shortening at this stage, but we are working on making the main text of the revised version more concise and shorter.
The approach to put quite a bit of information in the supplement is a compromise between getting a concise main text without compromising the reproducibility of our arguments. The revised main text will less heavily reference figures of the supplement, thus less feel to be part of the main story. Appendices are another means to keep the main text short, and contain self-contained discussions of aspects that might be of interest only to a minority of readers.

*1. P 1/L 18-19*

*"contrasted by...in autumn and winter" - ASM anticyclone does not exist in autumn and winter. Why we should talk about it.*

This note in the abstract reflects the approach we follow in the main text: Highlighting the specific features of the monsoon by comparing it to other seasons. For instance, LiNOx in summer is not higher than in spring, but net O3 production at maximum in the ASMA. This comparison shows that LiNOx alone is not responsible for enhanced O3 production in summer.

*2. P 1/L 20*

*"is regularly entrained a the eastern flank" - This is the isentropic in-mixing mentioned in my major point and not correctly described in your paper*

We will revise the text to make clear that the lateral entrainment of the ASMA is an adiabatic process.

*3. P 1/L 24*

*"by northerly" - I think "by southerly"*

Corrected. We mean "winds from the south".

*4. P 1/L 24*

*"Although..." - this sentence is not clear for me. I would remove it*

We removed the above sentence and now state at the beginning of the abstract that the measurements reflect the main processes acting throughout the monsoon season. It is one of the main objectives of the paper to put the HALO ESMVal measurements in the ASMA into perspective.

*5. P 2/L 11*

*I think that also "the eastward propagation of eddy shedding" is important now (Dethof et al.„ 1999; Vogel et al., 2014).*

Agree. We mentioned eastward eddy shedding only later in the paper, and have added the above recommendation to section 1.

*6. P 2/L 15*

*"the associated heat low" - do not understand what you mean*

Reformulated to: "The heat low associated with the hot desert conditions in summer …"

For details regarding large scale subsidence in the UT over that region please see Rodwell and Hoskins (1996), for the near surface conditions Lelieveld et al. (2009).

*I would recommend to focus the attention of the reader on ozone (observation very high in the core, but why, you will discuss it in the paper, also in the interannual context, etc). Instead of this you talk here too much about general aspects...*

We will revise this section to give it a better focus.

*For me this is the main motivation for the paper and it should roughly replace the part in P 3/L7-34 !*

We agree. Apart from stating the main motivation for the paper, the introduction shall also briefly introduce the topic, put the study in the context of the recent literature, and explain the structure of the paper.

*"The extratropics are dominated..." - this sentence is unnecessary.*

Removed.

*...dominate the averages in the chosen regions.*

Revised.

*too much. I can only recommend to remove this material*

The supplemental figures mentioned in this paragraph seem to be not essential indeed. We will double check with the revised text and shorten the supplement accordingly.

*12. Figure 1, caption*

*dynamical proxy of what.... I do not see any gray parts of the flight. "Panel a additionally shows..."????*

- Changed to: "… dynamical proxy to delimit the ASMA …"
- There is a grey section over the coast of Oman, which might be hard to see. Since the flight track has been discussed in the accompanying paper, we removed this information here.
- Changed to: "Panel (a) additionally …"

*"slowly descending HCl..." - this feature is very strange. Typically, during the considered season (JJA) there is a strong diabatic upwelling in the UTLS region confined by the anticyclone. Maybe you should explain it with model or remove it...*

This is explained in the introduction as follows: "In particular, the Rossby wave response to convection in the ASM region results in large-scale subsidence over the Arabian Peninsula (Rodwell and Hoskins, 1996), making it one of the warmest and driest regions on Earth. The associated heat low supports an anticyclone (Lelieveld et al., 2009), which intermittently merges with the ASMA."

*14. P 8/L7*
*"...differ between the summer monsoon season and the rest of the year.." – I would say the strong difference is during AMJJA and not only during JJA*

Changed to: "Simulated NO$_y$ profiles in the ASMA region from April to September differ to the rest of the year (Figs. 3cd), but the monsoon season is also distinct: …"

*15. P 8*
*The main part of NOy in the stratosphere should be HNO3, so I expect a much stronger correlation with HCl. Please comment.*

Apart from stratospheric contributions, NOy also contains LiNOx and uplifted pollutants. HCl is just related to stratospheric air.

*16. P 8-9*
*Section 3.4 contains for me too much information. I would reduce it by considering only the ozone-relevant NOx, NOy features.*

We consider the discussion of C- vs E-shaped NOy profiles interesting in itself, but agree that it is only a side aspect regarding ozone. Depending on other options for making the paper more concise, parts of this section might be moved from the main text.

*17. P 10, L5*
*NOx, typo*

Corrected

*18. P 10*
*Section 4 is a very important and novel part of the paper. It combines in situ observations (tracer-tracer correlations) with the model. It shows in a very nice way the interaction between the photochemical ozone production and stratospheric in-mixing. Because it does not use so much the observed NOx, NOy features, it is the next motivation to shorten section 3.4.*

Ok, thank you.

*19. P 12, L23*

*"because HCl ...are decreased" - with the vertical mixing lines you argue that HCl should be constant. Maybe you should reformulate*

Corrected to: "There are also a few almost vertical mixing lines in Fig. 5c, indicating case L3 described above."

*20. P 13, L20-31 and P 14*

*Here you show how important is the isentropic transport (mixing between the stratospheric and tropospheric air) on tilted isentropes. Here is also the origin for my major points.*

*21. On the following pages there are to many references to the supplement (see my major point) I can only recommend to shorten the following sections.*

Points 20 and 21 have already been addressed in the "Major comments" section. We will go through the paper and revise accordingly.

References

Lelieveld, J., Hoor, P., Jöckel, P., Pozzer, A., Hadjinicolaou, P., Cammas, J.-P., and Beirle, S.: Severe ozone air pollution in the Persian Gulf region, Atmos. Chem. Phys., 9, 1393-1406, 2009.

Rodwell, M. J., and Hoskins, B. A.: Monsoons and the dynamics of deserts, Q. J. R. Meteorol. Soc., 122, 1385-1404, 10.1002/qj.49712253408, 1996.

---

## Author Comment (AC2) · 12 Nov 2017

**Reply to Anonymous Referee #1**

The authors would like to thank the reviewer for those insightful comments, which greatly helped to improve the paper.

**5 Major comments:**

Presentation: - parts of the paper are too long and divert the reader from the main and strong elements brought by the paper. In particular, some rather general and introductory statements are given all along the manuscript (about the emissions, chemistry and dynamics...). It could be good if the authors try to shorten the paper and keep introductory elements to the introduction. Some examples of lengthy parts are mentioned in the detailed comments. - the paper is very often referencing to results from its accompanying paper which makes the reading and understanding somehow difficult. One example about O3 production regime is given below. - the same is true concerning the supplementary material which makes the paper a bit heavy to handle.

15

10

We agree that the presentation could be more focused and try to incorporate all corresponding recommendations. The final shape of the paper will depend on the outcome of this discussion, which might slightly redistribute the weighting between different aspects in the text. Therefore we are cautious singling out individual paragraphs for shortening at this stage, but we do aim to make the main text of

- 20 the revised version more concise and shorter. Each reference to the accompanying paper will be reconsidered to make the paper more stand-alone. Some references from the main sections can be moved to the introduction, some can be removed or reformulated to look less mandatory for the understanding of this paper, others can be substituted by existing or new analyses presented in the revised version (e.g. about the  $O_3$  production regime).
- 25 We also reconsider each reference to the supplementary material to make sure the main text is understandable without looking in the supplement. We prefer to keep references to individual supplemental figures in the main text, but now state in the introduction that the supplementary material contains only figures that illustrate side aspects, or show some additional details not compulsory for the understanding of the main text.

30

35

LiNOx and O3 production: Fig. 3h displays higher LiNOx production from EMAC in the Tibetan part of the ASMA during spring than during summer as mentioned P10L4-5. Nevertheless, the net O3 production is larger in summer than in spring down to 200 hPa below the tropopause (Fig. 2h). The authors explanation is that (i) in spring lower COV are uplifted by convection resulting in COV limitation and reduced O3 production (ii) LiNOx are produced locally in spring and not in summer. The latest argument also appears

in the Annexe about LiNOx (p20L14-16). Concerning (i) 1/ LiNOx production is linked to deep convection, especially in the models where both parametrization are coupled (in EMAC flashes are linked to convective updraught velocity as mentioned P20L10-11). Therefore more LiNOx should be associated with larger uplift of pollutants. Why EMAC

40 displays more LiNOx with less uplifted COV in spring?

> Figs. 2 and 3 in the discussion paper show profiles that have been averaged over relatively large lateral regions. Therefore spatial co-location of convection and increased CO mixing ratios in the lower troposphere is not guaranteed. Panels c, f, i, m of Figs. C1 and C2 reflect the spatial and temporal match

45 of deep convection and increased CO in different altitudes. The reviewer's comment mostly concerns the region marked "Tibetan" in Figs. C1 and C2. Convection is indeed stronger in spring (Figs. C1behk) than in

summer (Figs. C2behk), which is also supported by observations (Fig. C3). However, convection in April 2012 is very localized over the coastal regions of Western Bengal and Bangladesh. In contrast, during August 2012 convection is ubiquitous throughout the "Tibetan" region. It is most persistent at the southwestern flank of the Himalayas and over the Tibetan plateau. This coincides with the highest CO mixing

- 5 ratios, which are accumulated there by the prevalent south-westerly winds during the monsoon season. Consequently, more CO is transported through the troposphere in the "Tibetan" region during summer. However, the CO flux decreases towards higher altitudes (Figs. C2cfim), but profiles of the "Tibetan" region show almost constantly increased CO throughout the UT (Fig. 2d in the discussion paper). There is a different explanation for that: In the UT the ASMA is an –although leaky- transport barrier, allowing
- 10 some accumulation of the uplifted pollutants. There is no such transport barrier in spring. The manuscript will be revised to include the above reasoning.

**2/ Over South and East Asia the season of largest deep convection takes place in summer during the monsoon rather than in spring. Why are there more LiNOx in spring in EMAC?**

15

20

25

We compare EMAC-simulated lightning activity (intra-cloud + cloud-to-ground flash frequency) to the corresponding TRMM-LIS/OTD observations (Cecil, 2006) (Fig. C3). Convection is not explicitly resolved in the simulation, and the parameterizations for convection and lightning both introduce uncertainties to the simulation results for lightning. Uncertainties in the observations are due to a space- and time-dependent detection limit of 69% to 88%, and the application of a 3 month smoothing. Considering those uncertainties, the match between simulated and observed global distribution and the orders of magnitude of lightning activity is reasonable. In particular we note that also the observations over South Asia show stronger lightning activity during spring than during the monsoon season. The observed maximum of lightning activity over the coastal areas of Western Bengal and Bangladesh in April is well reproduced by the simulation. The above comparison will be made available with the revised manuscript.

- Concerning (ii): Looking at Fig. A1 displaying monthly LiNOx at 168 hPa, we see that they are localized over NW and NE India and Pakistan in May while in August they are more over SE India and Himalaya/Tibet. Nevertheless, the source is much stronger in spring. In both cases, the LiNOx emissions are very "patchy" and localized with also in June a single large emission spot over Bangladesh and in July the LiNOx spot localized over northern central India. Therefore, it is difficult to attribute a lower O3 production to more localized LiNOX emissions in spring.
- We agree that there is more lightning-produced NOx (LiNOx) in the "Tibetan" region in April/May compared to the monsoon season (Fig. 3h of the discussion paper). However, deep convection in EMAC is more evenly distributed throughout the "Tibetan" region in August than in April 2012 (Figs. C1, C2). This is reflected in the corresponding LiNOx distributions (Fig. C4dh). The LiNOx distribution is also more homogeneous in July and September (Figs. C4gi), compared to April June (Figs. C4def). The strong source of LiNOx in June is also rather limited in time (Fig. 3h).
- Photochemical O3 production (ProdO3) depends on a variety of parameters, e.g. ambient mixing ratios of  $H_2O$ , O3, CO and NOx (Ehhalt and Rohrer, 1994; Grooß et al., 1998; Jaeglé et al., 1998; Seinfeld and Pandis, 1998). We focus on NOx and CO in the following (Fig. C5). Prod3 non-linearly depends on ambient NOx mixing ratios: It increases proportional to NOx in the NOx-limited regime, but is almost independent
- 45 of NOx variations at higher NOx mixing ratios. A further increase of NOx even leads to decreasing ProdO3. Increasing CO has two effects: (i) It increases ProdO3; (ii) It shifts the point of maximum ProdO3 to higher NOx. Increasing H2O impacts ProdO3 qualitatively similar as increasing CO. Decreasing O3 leads to higher ProdO3, but NOx at the point of maximum ProdO3 is lowest for medium O3 mixing ratios. Just to give some approximate numbers (Ehhalt and Rohrer, 1994; Jaeglé et al., 1998; Grooß et al., 1998): For UT

conditions at northern mid latitudes the point of maximum  $O_3$  production may vary between 200 and 700 nmol/mol  $NO_x$ . Maximum net  $O_3$  production may vary by a factor of about 4, depending on ambient conditions.

We attribute the lower ProdO3 simulated for spring conditions mainly to the following two effects. 5 Firstly, locally very high NOx in spring does not help ProdO3 – or even pushes the system into the NOxsaturated regime. NOx close to maximum ProdO3 conditions throughout the region in summer leads to higher ProdO3 in the lateral average (Fig. 2h). Secondly, more CO in the UTLS in summer (Fig. 2d) increases ProdO3 and the maximum possible O3 production. We will clarify this in the manuscript accordingly. The above reasoning is also illustrated by an example, see the below discussion of Fig. C6.

10

Why are the LiNOx emissions so "patchy" on a monthly scale? The averaging should smooth horizontally the distributions because convection does not always occur at the same place. It could be interesting to compare LIS/OTD distributions of lightnings to EMAC LiNOx distributions.

- 15 The comparison is shown in Fig. C3 and discussed above. Smoothing applied to the observations certainly makes them less patchy. The parameterizations for convection (Tiedtke, 1989; Nordeng, 1994; Tost, 2006) and lightning (Grewe et al., 2001) used in our simulations have been tested in several studies (Tost et al., 2007; Grewe, 2009; Lopez, 2016) and appear to be state of the art. Simulated and observed NO along the HALO ESMVal flight track agree remarkably well within the ASMA region (Gottschaldt et al.,
- 20 2017). The same is true for comparisons of CARIBIC (www.caribic-atmospheric.com) measurements of NO and our simulation's output along the CARIBIC flight tracks in the ASMA region for the period May 2005 April 2014 (Fig. C7). The agreement is particularly noticeable for the monsoon season. There is no proof that EMAC is right for the right reasons, but at least those comparisons provide some confidence that the ESMVal monsoon case has been captured well by the simulation.
- 25 We are nevertheless aware that parameterizations for convection and LiNOx in global models are a notorious source of uncertainty. We do not consider an in-depth discussion of this aspect a focus of this paper, and therefore put most of the discussion concerning LiNOx in an appendix.

**Finally, in Barret et al. (2016) the LiNOx are not shown but a sensitivity test shows that O3 and NOx produced by LiNOx are the highest during the monsoon season which seems rather logical for the reasons discussed above. This discrepancy between the EMAC and GEOS-Chem models concerning LiNOx should be discussed.**

Our simulations also show that net  $O_3$  production is highest in the ASMA, as well as lightning being an important NOx source in the UT. We argue that LiNOx alone can not explain higher  $O_3$  production in the ASMA compared to spring. The main difference between the two seasons is the availability of CO (and related precursors) in the UT, resulting from different transport patterns and partial isolation of air in the ASMA circulation. More localized NOx emissions in spring, combined with the lack of confinement and stirring in the ASMA also contribute to less ProdO3 in spring. Despite higher LiNOx emissions in spring,

- 40 our simulation shows the highest net O3 production in the ASMA. EMAC sensitivity simulations also show a strong effect of LiNOx variations on ProdO3 (Figs. C6pq), so that aspect might not be too different to GEOS-Chem. The EMAC RC1SD-base-10a simulation is certainly not perfect regarding the temporal and spatial distribution of lightning, but at least the aspect of higher flash rates in South Asia during spring compared to summer is supported by observations (Fig. C3).
- 45 However, given the uncertainties related to LiNOx, a detailed comparison with other models is definitely warranted. Disentangling the various facets of ProdO3 beyond exemplary snapshots, and in different modelling systems would certainly contribute to a more robust understanding of the peculiar chemical conditions in the ASMA. Since this paper is considered to be too long already, we just add that as a

recommendation for future studies. This will also remind readers of the uncertainties associated with our results, derived from just one family of simulations.

- The east-west O3 net production gradient is logical as explained in the manuscript (P10L11-12) and in
  agreement with previous comparable evaluations by e.g. Liu et al. (JGR,2009) and Barret et al. (ACP,2016). Furthermore, the values of Fig. 2 seems in rather good agreement with those of the above mentioned papers for the monsoon season. A comparison and discussion of the EMAC O3 production with these previous studies could be interesting to strengthen and put the results in perspective.
- 10 The agreement of the values for net  $O_3$  production is indeed remarkable, given the uncertainties discussed above. Both studies have been cited already, but we will point out the agreement in the context of the different methodologies.
- Referring to the accompanying paper, it is mentioned that O3 production at 168 hPa is rather limited by
  CO than by NOx (P15L12-13). The exact sentence in the accompanying paper is "Net O3 production seems to depend more on CO (and related precursors) than on Nox". The use of "seems" shows that the authors are rather uncertain. I do not really understand how is this possible because of the rather high CO concentrations within the ASMA (70-100 ppbv according to HALO and all references cited in the paper). This statement of a generally CO-limited regime in the ASMA at 168 hPa needs to be demonstrated and is
- 20 not supported by the literature. For instance, according to Brune (IGAC Isuue 21, 2000), in the upper troposphere, the O3 regime is NOx limited for NOx concentrations lower than some hundreds pptv.

We illustrate the consequences of Fig. C5 (introduced above) by snapshots from a series of sensitivity simulations (Fig. C6). Going down from the tropopause in the ASMA,  $NO_x$  and  $O_3$  generally decrease,

25 while CO and H2O increase (Figs. 2 and 3, H2O not shown). Net O3 production (Fig. C6γ) in the UT is determined rather by ProdO3 (Fig. C6δ) than by LossO3 (Fig. C6ζ). Increasing CO increases ProdO3 and decreased CO results in decreased ProdO3 (compare Figs. C6ab with Figs. C6no). ProdO3 per NOx shows a strong gradient in the altitude of maximum ProdO3 (Fig. C6ε), indicating the transition from the NOxlimited to the NOx-saturated regime. The maximum corresponds to about 300 pmol/mol NOx (Fig. C6β).

- 30 Variations of NOx in the altitude region of increased NOx (above max. ProdO3) have a relatively little effect on ProdO3 (compare Figs. C6gh with Figs. C6pq). ProdO3 even decreases in the region of the largest NOx increase (Fig. C6m: 22°N, 16 km). CO sharply decreases in that altitude region (Fig. C6 $\alpha$ ), so the photochemical regime corresponds to the blue section ("3") in Fig. C5. More CO is available in the regions of maximum ProdO3, corresponding to the green section ("2") in Fig. C5. Towards lower
- 35 altitudes, NOx decreases and CO increases. Although NOx increases/decreases less than in higher altitudes, ProdO3 increases/decreases more (~12 km in Figs. C6ghpq). This is the NOx-limited regime ("1" in Fig. C5). ProdO3/NOx also varies (Figs. C6tu), indicative of the non-linear region (grey in Fig. C5). The non-linear dependence of ProdO3 on ambient trace gas mixing ratios leads to the simulated maximum within the opposite gradients of those trace gases in the UT ASMA. In principle, all above mentioned
- 40 operating modes of the chemical system ("1", "2", "3" in Fig. C5) could be in the  $NO_x$ -limited regime and still lead to a maximum of net  $O_3$  production in the UT. It's a multi-dimensional problem. CO (and others) determine the curve in the  $NO_x$ -vs-ProdO3 diagram, and  $NO_x$  determines the operating point on the curve. The formulation in the paper is indeed misleading in that respect and will be revised.

**45 Tracer-tracer relationships:**

This part is very interesting because the 3 tracers document different transport and chemical processes. HCl in particular which is rarely used is good to trace stratospheric air because O3 is photochemically produced in the troposphere. The authors explain that "mixing line with negative CO/O3 slope dominate" in Fig 4c (corresponding to positive slopes in Fig. 5c). They correspond to mixing stratospheric air with photochemically processed tropospheric air. Neverthelesse, in Fig. 5c, we also see some horizontal mixing lines (red and green). According to the discussion in p11 and 12 they correspond to mixing of fresh uplifted pollution (increasing CO, horizontal lines not so clear in Fig. 4c) and stratospheric air (increasing

- 5 HCl in Fig 5c) with antagonist effects on O3. Another line with O3 decrease / HCl increase in Fig. 5c and O3 decrease / CO increase in Fig. 4c corresponding to mixing of fresh pollution in the UTLS can also be isolated. It is difficult to see whether these mixing lines correspond to important part of the sampled air masses but they could be mentioned.
- Explanations for the different types of mixing lines are offered in the context of the hypothetical lines (L1-L5), shown in Figs. 4b and 5b. In the revised manuscript we refer more often to specific hypothetical lines when discussing Figs. 4c and 5c. The discussion of vertical lines in Fig. 5c has been corrected. A detailed quantification of different processes' contributions to individual measurements would require more sophisticated analyses along back-trajectories.
- 15

Details:

*Part 3: the description of the different species at beginning of §3.1; 3.2, 3.3 and 3.4 (origin, chemistry etc.) are too close to "textbook" descriptions and should be shorten for readability.*

20 We feel that the main characteristics of each tracer should be mentioned here, at least as far as needed to understand the tracers' behavior in Figs. 2 and 3. We also see the need for shortening and will remove side aspects from the main text.

Part 5: this part tries to describe the different processes that control the ASMA composition one by one. It
is interesting and well documented but rather lengthy and descriptive. For instance, the description of the evolution of the CO and HCL distributions P17L1-15 is very detailed and could be summarized.

P17L1-14: the dynamics are very detailed with the evolution of the air masses but is it really necessary to give so much details?

30

Will be revised accordingly.

*Fig2 and 3: the plots are shown in pressure coordinates that makes the region around the tropopause very compact. Readability would be better in logP coordinates*

35

Will be changed.

(altitude plots are provided in the supplement but it makes the reading uncomfortable and could be simply removed).

40

Ok.

*p5l13:* why choose PV = 3.5 for the tropopause in the extratropics? Most studies choose 2 or 1.5.

This is the standard tropopause definition of EMAC in the extratropics, as introduced by Jöckel et al. (2006).

5

P5l27: Fig.1 is referenced after Fig 2 and 3.

Fig. 1 is referenced first on page 4.

10 p9l5: "indicates that relatively: : :"

Reformulated.

---

## Author Response (AR1)

**Relevant changes**

The selection and order of figures have changed as follows:

| ACPD | Revised | ACPD | Revised | ACPD | Revised |
|---|---|---|---|---|---|
| **1** | **1** |  | S4cd | S9 | *B3abc* |
| **2abcdef** | **4abcdef** | **7ab** | **2ac** | S10 | *B3def* |
| **2gh** | **5ab** |  | **2bd** | S11 | S9 |
| **3ab** | **4gh** | **8** | S14 | S12, S13, S14 |  |
| **3cdef** | *B1abcd* | **9** | S16 | S15 | S17 |
| **3gh** | **5cd** | **10** | **7** | S16abce | S5abcd |
|  | **5efgh** | **11** | **8** | S16dh | S6ab |
| **4abc** | **3abc** | *A1* | *A1* |  | S6cd |
| **5abc** | **3def** | *A2* | A3 | S16fg | S7ab |
| **6abce** | S2abcd | *A3* | A4 |  | S7cd |
| **6dh** | S3ab | S1, S2, S3, S4, S5, S6 |  |  | **6** |
|  | S3cd | S7 | B2 |  | *A2* |
| **6fg** | S4ab | S8 | S8 |  | S1, S10, S11, S12, S13, S15 |

- Highlighted figures are new or have been modified compared to their respective ACPD versions. Most modifications are related to using different vertical coordinates. Other figures include additional panels or features.
- The paper has been restructured. The markup-version of the manuscript might be a bit misleading, because copy and paste of sections shows up as new or removed text. There is new text and there is removed text, but most snippets are largely unchanged.
- Supplement and accompanying paper are less heavily referenced. Readers should now be able to go through the main text without looking into the supplement.
- Discussion extended for: Photochemical O3 production, LiNOx, vertical coordinates
- Convection and uplift of CO: The discussion has been extended not only compared to the ACPD version, but the revised manuscript also contains an additional aspect that has been missing in the open discussion.
- More validation for lightning and nitrogen oxides
- Focus shifted towards processes in the ASMA
- Detailed discussion of NOy put in a new appendix. The strategy to shorten the main text is explained in the detailed responses to the referees.
- Shortened the text despite additional content, striving to make it more concise.

**Updated reply to Anonymous Referee #1**

Major comments:

5    *Presentation: - parts of the paper are too long and divert the reader from the main and strong elements brought by the paper. In particular, some rather general and introductory statements are given all along the manuscript (about the emissions, chemistry and dynamics...). It could be good if the authors try to shorten the paper and keep introductory elements to the introduction. Some examples of lengthy parts are mentioned in the detailed comments. - the paper is very often referencing to results from its*

10    *accompanying paper which makes the reading and understanding somehow difficult. One example about O3 production regime is given below. - the same is true concerning the supplementary material which makes the paper a bit heavy to handle.*

We thoroughly restructured the paper, slightly shifting the focus from putting the HALO ESMVal

15    measurements into context towards ASMA O3 and corresponding processes. Parts of the text became obsolete, while the discussion of aspects like photochemical O3 production, LiNOx, convective transport, vertical coordinates and validation was extended.

We strived to make the main text less dependent on references to the supplement or to the accompanying paper. Readers should now be able to go through the main text stand alone. There are

20    still many figures in the supplement, but they are there mainly for documentation and reproducibility. This is also stated in the introduction.

Despite additional contents, the main text (incl. figures) is about 20% shorter now, and hopefully more concise and focused.

25    *LiNOx and O3 production: Fig. 3h displays higher LiNOx production from EMAC in the Tibetan part of the ASMA during spring than during summer as mentioned P10L4-5. Nevertheless, the net O3 production is larger in summer than in spring down to 200 hPa below the tropopause (Fig. 2h). The authors explanation is that (i) in spring lower COV are uplifted by convection resulting in COV limitation and reduced O3 production (ii) LiNOx are produced locally in spring and not in summer. The latest argument also appears*

30    *in the Annexe about LiNOx (p20L14-16).*
   *Concerning (i) 1/ LiNOx production is linked to deep convection, especially in the models where both parametrization are coupled (in EMAC flashes are linked to convective updraught velocity as mentioned P20L10-11). Therefore more LiNOx should be associated with larger uplift of pollutants. Why EMAC displays more LiNOx with less uplifted COV in spring?*

We added a discussion of this very interesting question to the revised manuscript, section 6.2.

*2/ Over South and East Asia the season of largest deep convection takes place in summer during the monsoon rather than in spring. Why are there more LiNOx in spring in EMAC?*

We added a comparison between EMAC-simulated lightning activity and the corresponding TRMM-LIS/OTD observations to the revised manuscript (section 6.1 and Appendix A). Overall, the agreement is reasonable. In particular, also the observations show a maximum of lightning activity in spring.

The corresponding figure has been put in the supplement, because lightning is not a focus of this paper.

45    It is rather the resulting NOx background that interests here and a corresponding comparison to CARIBIC observations has been added to the revised manuscript (section 2, Appendix A).

*Concerning (ii): Looking at Fig. A1 displaying monthly LiNOx at 168 hPa, we see that they are localized over NW and NE India and Pakistan in May while in August they are more over SE India and Himalaya/Tibet. Nevertheless, the source is much stronger in spring. In both cases, the LiNOx emissions are very "patchy" and localized with also in June a single large emission spot over Bangladesh and in July the LiNOx spot localized over northern central India. Therefore, it is difficult to attribute a lower O3 production to more localized LiNOX emissions in spring.*

We extended the discussion on photochemical O3 production (section 6.3), which now addresses the above question. Essentially we attribute increased ProdO3 in summer to the combination of two effects: (i) The increased availability of CO; (ii) The decrease of ProdO3 in high-NOx air.

*Why are the LiNOx emissions so "patchy" on a monthly scale? The averaging should smooth horizontally the distributions because convection does not always occur at the same place. It could be interesting to compare LIS/OTD distributions of lightnings to EMAC LiNOx distributions.*

This has been added to the revised manuscript (section 6.1 and Appendix A).

*Finally, in Barret et al. (2016) the LiNOx are not shown but a sensitivity test shows that O3 and NOx produced by LiNOx are the highest during the monsoon season which seems rather logical for the reasons discussed above. This discrepancy between the EMAC and GEOS-Chem models concerning LiNOx should be discussed.*

A dedicated comparison would be needed to pinpoint and discuss discrepancies between different model systems. This didn't seem feasible in the context of this paper, which already required shortening. Therefore we added a corresponding statement to the draft (section 6.1) and for now resorted to showing that lightning and NOx are reasonably well represented in our simulation.

*The east-west O3 net production gradient is logical as explained in the manuscript (P10L11-12) and in agreement with previous comparable evaluations by e.g. Liu et al. (JGR,2009) and Barret et al. (ACP,2016). Furthermore, the values of Fig. 2 seems in rather good agreement with those of the above mentioned papers for the monsoon season. A comparison and discussion of the EMAC O3 production with these previous studies could be interesting to strengthen and put the results in perspective.*

This has been added to section 6.3.

*Referring to the accompanying paper, it is mentioned that O3 production at 168 hPa is rather limited by CO than by NOx (P15L12-13). The exact sentence in the accompanying paper is "Net O3 production seems to depend more on CO (and related precursors) than on Nox". The use of "seems" shows that the authors are rather uncertain. I do not really understand how is this possible because of the rather high CO concentrations within the ASMA (70-100 ppbv according to HALO and all references cited in the paper). This statement of a generally CO-limited regime in the ASMA at 168 hPa needs to be demonstrated and is not supported by the literature. For instance, according to Brune (IGAC Isuue 21, 2000), in the upper troposphere, the O3 regime is NOx limited for NOx concentrations lower than some hundreds pptv.*

We extended the discussion of photochemical O3 production (section 6.3). In particular we added a figure (Fig. 6) to resolve the mix-up between chemical regimes (e.g. NOx-limited, NOx-saturated) and operating modes of the chemical system.

*Tracer-tracer relationships:*
*This part is very interesting because the 3 tracers document different transport and chemical processes.*
*HCl in particular which is rarely used is good to trace stratospheric air because O3 is photochemically*
*produced in the troposphere. The authors explain that "mixing line with negative CO/O3 slope dominate"*
*in Fig 4c (corresponding to positive slopes in Fig. 5c). They correspond to mixing stratospheric air with*
*photochemically processed tropospheric air. Neverthelesse, in Fig. 5c, we also see some horizontal mixing*
*lines (red and green). According to the discussion in p11 and 12 they correspond to mixing of fresh*
*uplifted pollution (increasing CO, horizontal lines not so clear in Fig. 4c) and stratospheric air (increasing*
*HCl in Fig 5c) with antagonist effects on O3. Another line with O3 decrease / HCl increase in Fig. 5c and*
*O3 decrease / CO increase in Fig. 4c corresponding to mixing of fresh pollution in the UTLS can also be*
*isolated. It is difficult to see whether these mixing lines correspond to important part of the sampled air*
*masses but they could be mentioned.*

Explanations for the different types of mixing lines are offered in the context of the hypothetical lines (L1-L5), shown in Figs. 4b and 5b. In the revised manuscript we refer more often to specific hypothetical lines when discussing Figs. 4c and 5c. The discussion of vertical lines in Fig. 5c has been corrected.
A detailed quantification of different processes' contributions to individual measurements would require more sophisticated analyses along back-trajectories.

Details:

*Part 3: the description of the different species at beginning of §3.1; 3.2, 3.3 and 3.4 (origin, chemistry*
*etc.) are too close to "textbook" descriptions and should be shorten for readability.*

In the revised manuscript we only shortly motivate the selection of tracers in the context of the tracer-tracer correlations (section 4), which is now before the discussion of the annual evolution of tracer profiles (former section 3; now section 5).

*Part 5: this part tries to describe the different processes that control the ASMA composition one by one. It*
*is interesting and well documented but rather lengthy and descriptive. For instance, the description of the*
*evolution of the CO and HCL distributions P17L1-15 is very detailed and could be summarized.*
~
*P17L1-14: the dynamics are very detailed with the evolution of the air masses but is it really necessary to*
*give so much details?*

The corresponding figure has been moved to the supplement and the main text now only contains a summary of the splitting event (section 6.6).

*Fig2 and 3: the plots are shown in pressure coordinates that makes the region around the tropopause*
*very compact. Readability would be better in logP coordinates*

This is now Figs. 4 and 5, given in logP coordinates. A short discussion on the selection of vertical coordinate systems has also been added (end of section 3).

*(altitude plots are provided in the supplement but it makes the reading uncomfortable and could be simply removed).*

5  The altitude plots have been removed, but the extension of Figs. 4 and 5 to five years is still given in the supplement. Those supplemental figures are in isentropic coordinates, just for documenting that the general picture remains similar to logP coordinates.

*p5l13: why choose PV = 3.5 for the tropopause in the extratropics? Most studies choose 2 or 1.5.*

This is the standard tropopause definition of EMAC in the extratropics, as introduced by Jöckel et al. (2006).

*P5l27: Fig.1 is referenced after Fig 2 and 3.*

The figures have changed, and also their numbering. The numbering of figures is determined by the order of where they are mainly discussed, rather than by their first mentioning in the text. For example, a side aspect ("ridge line") of Fig. 7 is briefly mentioned in the context of defining the ASMA region in section 3., but Fig. 7 is not really discussed before section 6.4.

*p9l5: "indicates that relatively: : :"*

Reformulated (now p10l17).

**Updated reply to Anonymous Referee #2**

Major comments:

*General:*
*This is very interesting and important paper which is worth to be published. The most important finding is the explanation of high ozone within the Asian summer monsoon anticyclone. The authors show that photochemical ozone production in the circulating air masses as well isentropic in-mixing from the stratosphere are key processes defining ozone. In this picture the anticyclone can be understood as a photochemically active and not well-isolated reactor. In this reactor there are two parts: "convectively driven" eastern (Tibetian) part and more "chemistry driven" western (Iranian) part. Although the paper is well-written, it suffers from an inadequate presentation (see major points). Because of this, the paper needs a major revision.*

*1. As you show in many places, isentropic mixing between the stratospheric air (you call it TP layer) and the interior of the anticyclone is an important process in your chain of arguments. In Fig. 7 you show how such isentropes connecting the extratropical lower stratosphere intersect the tropopause (some almost perpendicular) and penetrate into the anticyclone itself. The mixing (stirring) happens on such isentropes and is almost a 2d process. So I do not understand, why you do not show the respective tracer distributions at such isentropes. I guess θ = 360 K would be the right choice instead of using the 168hPa level (e.g. Fig. 8, Fig 9 or Fig. 10). I would recommend to show Fig. 7 much earlier in the text (e.g. as the second figure of your paper) and than use much more isentropic analysis. As you mentioned, such isentropes are tilded in pressure space, but transport occurs much more on such isentropes.*

We added a short discussion on vertical coordinates at the end of section 3. Figure 7 has been moved to the beginning (now Fig. 2, see table on the first page of this document) and we revised the discussion accordingly throughout the paper. Several figures are now in isentropic coordinates. The 355 K level was chosen as a compromise between representing the HALO ESMVal measurements and little intersection with the TP near the equator.

*2. For me it is unreasonable to include 16 figures into the supplement! If your story exceeds something like 10-12 main figures and 2-5 figures of your appendix, you should divide the story into two parts or make your story shorter. The last point seems for me to be more your case. Your abstract roughly describes your main results (see also my general comments). So maybe, you can go through the text and remove everything what is not supportive for your main results (see also my minor points).*

Several figures have been removed, but new content also required new figures. In summary the main text is considerably shorter now. The focus of the paper has shifted towards O3 and related processes in the ASMA. There are still many figures in the appendices and the supplement. However, the main text is more stand-alone now. We note in the introduction that the supplement is mainly for documenting side aspects and throughout the text try not to encourage readers to visit the supplement.  However, we prefer to publish these supplemental figures with the paper for the reproducibility of our arguments.
The situation is less clear for the appendices. As a compromise between brevity and completeness, detailed discussions of LiNOx and NOy are put there, i.e. closer to the main text. Summaries of the findings in the main text should be sufficient for understanding our chain of arguments, but we feel that

some readers will be interested in the details. Since nitrogen oxides play an integral role in the ASMA, we would prefer to publish them with this rather than spawning a separate paper.

Minor Points:

*1. P 1/L 18-19*

*"contrasted by...in autumn and winter" - ASM anticyclone does not exist in autumn and winter. Why we should talk about it.*

This phrase was removed when shortening the abstract. However, throughout the paper we still highlight the specific features of the monsoon by comparing it to other seasons.

*2. P 1/L 20*

*"is regularly entrained a the eastern flank" - This is the isentropic in-mixing mentioned in my major point and not correctly described in your paper*

Starting from Fig. 2, the representation of isentropic in-mixing has been modified throughout the paper. We also note that the details of this process are a topic of ongoing research.

*3. P 1/L 24*

*"by northerly" - I think "by southerly"*

Corrected. We mean "winds from the south".

*4. P 1/L 24*

*"Although..." - this sentence is not clear for me. I would remove it*

We removed the above sentence and now state at the beginning of the abstract that the measurements reflect the main processes acting throughout the monsoon season. It is one of the main objectives of the paper to put the HALO ESMVal measurements in the ASMA into perspective.

*5. P 2/L 11*

*I think that also "the eastward propagation of eddy shedding" is important now (Dethof et al.„ 1999; Vogel et al., 2014).*

Agree. We mentioned eastward eddy shedding only later in the paper, and have added the above recommendation to section 1.

*6. P 2/L 15*

*"the associated heat low" - do not understand what you mean*

This has been reformulated in the revised draft. In short: Overall upwelling in the eastern part of the ASMA is accompanied by large scale subsidence in the western part. UT subsidence results in mostly

clear skies, heating up the landmass of the Arabian Peninsula. The hot air rises, generating a thermal low near the ground and an anticyclone in the outflow region in the mid troposphere.

*7. P 3/L7-34*

*I would recommend to focus the attention of the reader on ozone (observation very high in the core, but why, you will discuss it in the paper, also in the interannual context, etc). Instead of this you talk here too much about general aspects...*

The focus of the paper has been shifted towards O3.

*8. P 5/L1-6*
*For me this is the main motivation for the paper and it should roughly replace the part in P 3/L7-34 !*

Section 1 has been revised accordingly.

*9. P 5/L14-16*
*"The extratropics are dominated..." - this sentence is unnecessary.*

Removed.

*10. P 5/L24*
*...dominate the averages in the chosen regions.*

Revised.

*11. P 5/L29-32*
*too much. I can only recommend to remove this material*

Most of those supplemental figures have been removed (see table on the first page of this document).

*12. Figure 1, caption*
*dynamical proxy of what.... I do not see any gray parts of the flight. "Panel a additionally shows..."????*

- Changed to: "… dynamical proxy to delimit the ASMA …"
- There is a grey section over the coast of Oman, which might be hard to see. Since the flight track has been discussed in the accompanying paper, we removed this information here.
- Panels are now denoted with brackets, e.g. (a), (b), …

*13. P 7/L21-24*
*"slowly descending HCl..." - this feature is very strange. Typically, during the considered season (JJA) there is a strong diabatic upwelling in the UTLS region confined by the anticyclone. Maybe you should explain it with model or remove it...*

As noted above, there is overall upwelling in the eastern part of the ASMA, and large scale subsidence in the UT over the Arabian Peninsula. For instance, Nützel et al. (2016) show a corresponding figure.

*14. P 8/L7*

*"...differ between the summer monsoon season and the rest of the year.." – I would say the strong difference is during AMJJA and not only during JJA*

Changed to: "Simulated NO$_y$ profiles in the ASMA region from April to September differ to the rest of the year (Figs. 3cd), but the monsoon season is also distinct: …"

*15. P 8*

*The main part of NOy in the stratosphere should be HNO3, so I expect a much stronger correlation with HCl. Please comment.*

Apart from stratospheric contributions, NOy also contains LiNOx and uplifted pollutants. HCl is just related to stratospheric air.

*16. P 8-9*

*Section 3.4 contains for me too much information. I would reduce it by considering only the ozone-relevant NOx, NOy features.*

We consider the discussion of C- vs E-shaped NOy profiles interesting in itself, but agree that it is only a side aspect regarding ozone. The discussion of reactive nitrogen is now mostly in appendix B.

*17. P 10, L5*

*NOx, typo*

Corrected

*18. P 10*

*Section 4 is a very important and novel part of the paper. It combines in situ observations (tracer-tracer correlations) with the model. It shows in a very nice way the interaction between the photochemical ozone production and stratospheric in-mixing. Because it does not use so much the observed NOx, NOy features, it is the next motivation to shorten section 3.4.*

Ok, thank you.

*19. P 12, L23*

*"because HCl ...are decreased" - with the vertical mixing lines you argue that HCl should be constant. Maybe you should reformulate*

Corrected to: "There are also a few almost vertical mixing lines in Fig. 5c, indicating case L3 described above."

*20. P 13, L20-31 and P 14*

*Here you show how important is the isentropic transport (mixing between the stratospheric and tropospheric air) on tilted isentropes. Here is also the origin for my major points.*

*21. On the following pages there are to many references to the supplement (see my major point) I can only recommend to shorten the following sections.*

Points 20 and 21 have already been addressed in the "Major comments" section.

References

[revised manuscript text omitted]

Given the uncertainties of the parameterizations for convection and lightning in the model, smoothing and limited detection efficiencies in the observations, our simulated spatial-temporal distribution of lightning activity compares reasonably well to corresponding observations (supplement, Fig. S10). In particular we note that also the observations over South Asia show stronger lightning activity during spring than during the monsoon season. The observed maximum of lightning activity over the coastal areas of Western Bengal and Bangladesh in April also shows up in the simulation.

As noted in section 2, we compare simulated NO and $NO_x$ to the corresponding IAGOS-CARIBIC observations (Fig. A2). Commercial airliners do not fly as high as HALO and the tracks hardly reach the southern ASMA fringe, but the northern ASMA edge and the center of the monsoon region have been sampled multiple times. We evaluated all 345 IAGOS-CARIBIC flights between 19 May 2005 and 9 April 2014, considering the respective latest data versions as of 10 November 2017. In total 86 flights between Frankfurt (IATA code: FRA) and Chennai (MAA) or Guangzhou (CAN) or Bangkok (BKK) transected the ASMA region. 32 of these flights provide NO data there, and 66 flights provide $NO_x$. Neglecting data below 300 hPa and subsampling to the time resolution of the simulation yields the numbers of comparable data that are given in Figs. A2bd. Given the above uncertainties related to the representation of $LiNO_x$ in the simulation, NO matches the corresponding IAGOS-CARIBIC observations surprisingly well (Fig. A2b). This holds also for the more robust (more data) global comparison (Fig. A2a). Increased $LiNO_x$ emissions in the ASMA region in spring are also consistent to IAGOS-CARIBIC, and the simulation might even slightly underestimate those emissions (Fig. A2b: MAM).

In order to link the $LiNO_x$ emissions to the $NO_x$ burden in the ASMA region, a suite of EMAC sensitivity simulations with modified emission factors was conducted (Figs. A3, A4). All EMAC analyses in the main text are based on

simulation RC1SD-base-10a (Jöckel et al., 2016), which is given in Fig. A3 just for comparison. The other simulations discussed in the context of Figs. A3 and A4 here are derived from EMAC simulation RC1SD-base-10, which differs in road traffic emissions and optical properties of stratospheric aerosol (Jöckel et al., 2016) from RC1SD-base-10a. Total LiNO$_x$ emissions in RC1SD-base-10 are 4.6 Tg(N) in 2012 (Jöckel et al., 2016), which is in the realistic range of 2 – 8

5 Tg(N) yr$^{-1}$ (Schumann and Huntrieser, 2007). RC1SD-base-10 and our base simulation for the LiNO$_x$ sensitivity analysis (b01) are both operated in chemistry-climate model (CCM) mode, i.e. including interactive chemistry with feedback on dynamics. Simulation b01 differs only in the usage of daily (Kaiser et al., 2012) instead of monthly biomass burning emissions and 5 h instead of 10 h output intervals. Feedbacks from chemistry on dynamics in all quasi chemistry-transport model mode (QCTM) (Deckert et al., 2011) simulations are based on identical trace gas time series  from b01.

10 The same dynamics incl. convection is simulated in all QCTM simulations. Differences between a QCTM reference simulation (q01) and sensitivity simulations (s*) are thus exclusively due to chemical perturbations. All QCTM simulations cover June – September 2012, but the first 3 months were discarded for spin-up.

Figure A3 shows that RC1SD-base-10a captures observed O$_3$, CO, NO and NO$_y$ along the HALO ESMVal flight path slightly better than b01 and q01. We are yet confident that the overall agreement is good enough for the analysis of chemical

15 perturbations. For the QCTM sensitivity analyses it is more important to note that differences between b01 and q01 are negligible.

Figure A4k shows that halving LiNO$_x$ emission factors results in almost halved NO$_x$ in the uppermost troposphere. Doubling of LiNO$_x$ emissions leads to almost doubled NO$_x$ just below the tropopause (Fig. A4m). The biggest relative sensitivity in Fig. A4km almost coincides with the altitude range of the largest NO$_x$ mixing ratios just below the

20 tropopause (Fig. A4j). Thus, in our simulations LiNO$_x$ clearly dominates the NO$_x$ budget from the tropopause to 100 hPa below it. The impact of LiNOx fades out at lower altitude, and almost vanishes at 400 hPa below the tropopause. This is consistent with the profiles of LiNO$_x$ emissions in September 2012, which mainly occur in the Tibetan part of the ASMA (Fig. 5d).

Modifications of NO$_x$ print through on other O$_3$ precursors mainly via changes to the atmospheric oxidizing capacity (OH:

25 Figs. A4ghi). In response to halved LiNO$_x$, OH decreases 200 hPa below the tropopause and lower, and increases above (Fig. A4h). The effects are reversed for doubled LiNO$_x$ (Fig. A4i). The largest relative effects coincide with largest absolute OH mixing ratios.

CO decreases throughout the shown altitude range for halved LiNO$_x$ (Fig. A4e). Without major production terms in the UT, modifications to CO mixing ratios are dominated by the loss reaction CO + OH → H + CO$_2$. The rate coefficient of

30 this reaction is proportional to pressure, and otherwise depends only on constants ( supplement to Jöckel et al. (2016)). Laterally averaged CO mixing ratios vary little from 50 to 400 hPa below the tropopause (Fig. A4d), but are affected by decreased and increased OH (Figs. A4fg). Decreased OH in the lower half of the domain dominates the overall CO response. CO rises through this region with higher pressures before reaching the UT in the Tibetan part of the ASMA (Fig. 4), which obviously outweighs the CO response to increased OH in the UT of Tibetan and Iranian part combined. Increased

OH 200 – 500 hPa below the tropopause consequently leads to an overall decrease of CO in response to doubled ₓLiNOₓ (Fig. A3f). The O₃ precursors NOₓ and CO display opposite trends in response to ΔₓLiNOₓ. Curiously, HCl shows the opposite response to modified ₓLiNOₓ (Figs. A3abc). There is no chemical production of HCl in the UT, and the only loss term in the simulations is HCl + OH → Cl + H₂O. The rate coefficient of this reaction is

5   1.7E-12 * EXP(-230/Temperature), see supplement to Jöckel et al. (2016). However, the tropospheric response of HCl to ΔOH is dominated rather by the vertical profile of HCl mixing ratios than by lower temperatures towards the tropopause. Almost all HCl in the UT is of stratospheric origin, and HCl mixing ratios steeply increase across the tropopause. Thus the UT response of HCl is dominated by ΔOH near the tropopause: increased OH for halved ₓLiNOₓ increases HCl losses, and vice versa for doubled ₓLiNOₓ.

10  The response of UT net O₃ production to ΔₓLiNOₓ (Figs. A3qrs) has mostly the same sign as ΔNOₓ. As noted already in the context of Fig. 2, opposite gradients of O₃ precursors NOₓ and CO in the UT lead to a broad altitude range of enhanced net O₃ production in the ASMA, centred about 100 hPa below the tropopause. O₃ production is limited by NOₓ in lower altitudes and by CO (and other volatile organic compounds) towards the tropopause. NOₓ and CO display opposite trends in response to ΔₓLiNOₓ, but relative changes to NOₓ are larger and dominate the overall response of net O₃

15  production. We note, however, that the largest increase of NOₓ at the end of September (circled in Fig. A3m) decreases net O₃ production to zero or even net loss (circled in Fig. A3s), indicative of the NOₓ-limited photochemical regime.  O₃ mixing ratios respond to ΔₓLiNOₓ essentially like net O₃ production in the UT (Figs. A3nop). The altitude of

20  maximum relative ΔO₃ is slightly lower than the altitude of maximum absolute changes to net O₃ production. We attribute this effect to upwards increasing absolute O₃ mixing ratios.

25  ~~("end-members"). If the ratios of the end members remain constant over time, the slope of the mixing line is conserved, as long as the mixing process continues. If the mixing processes stops, the mixing lines converge to a single point in the tracer-tracer diagram. If the reservoir of one end-member is bigger than the other, points in the tracer-tracer diagram will be close to the dominating end-member. However, the relative size of the reservoirs does not affect the slope of mixing lines, thus allowing detection of even small entrainments. Slopes change in case of mixing~~

30

Appendix B: Reactive nitrogen in the ASMA

5  Nitrogen oxides are key parameters in atmospheric chemistry, partly controlling the $O_3$ production in the troposphere and lower stratosphere. In the UTLS, enhanced $NO_y$ originates both from tropospheric and from stratospheric sources. In the lower troposphere odd nitrogen species are co-emitted with carbon monoxide in combustion processes, resulting in a strong correlation between both species. $NO_y$ is enhanced in the stratosphere (mainly $HNO_3$), but it also comprises species with tropospheric sources ($LiNO_x$). Thus it is not a viable tracer for stratospheric air on its own.

10  The simulation matches IAGOS-CARIBIC $NO_y$ almost perfectly on the global scale (Fig. A2c), and only moderately overestimates it during summer in the ASMA region (Fig. A2d: JJA).

[revised manuscript text omitted]

**Figure A2.** Comparison of IAGOS-CARIBIC (black) measurements in the altitude range between 300 hPa and the TP ("Tr") with corresponding results of the EMAC RC1SD-base-10a simulation (red) for (a) NO globally, (b) NO in the ASMA region (15-35°N, 30-100°E), (c) $NO_y$ globally, (d) $NO_y$ in the ASMA region. All data stem from the period May 2005 to April 2014. The simulation was sampled along the IAGOS-CARIBIC flight tracks with a resolution of 12 min, and IAGOS-CARIBIC observations were subsequently interpolated (interval mean) to a resolution of 12 min. Numbers n below the plots show the number of the remaining data pairs (after interpolation and filtering) available for the respective seasons. Dots represent mean values, whiskers indicate standard deviation, min & max values. Rectangles represent the median, and whiskers the percentiles 5, 25, 75, 95.

[Figure]

Figure A32. Mixing ratios of O₃, CO, NO and NOy along the HALO flight track from Male to Larnaca, on 18 September 2012. Grey shading marks the first flight section in ASMA air POH. Grey line: in situ measurements in 10 s resolution, black: in situ averaged to 12 min simulation time steps, R10a: EMAC simulation RC1SD-base-10a. Sensitivity simulations are based on the almost identical RC1SD-base-10 simulation of (Jöckel et al., 2016), feature daily instead of monthly biomass burning emissions, and were performed in quasi chemistry transport model mode (Deckert et al., 2011) to facilitate isolating the effects of modified emissions.

b01: as R10a, but with different traffic and different biomass burning emissions;

q01: as b01, but QCTM;

s03: as q01, but halved LNOₓLiNOₓ emissions;

s04: as q01, but doubled LNOₓLiNOₓ emissions;

s05: as s03, but with a different vertical emission profile of LNOₓLiNOₓ (emission factors not decreased in the mid-troposphere, i.e. no C-shape)

[Figure]

**Figure A43.** Evolution of simulated trace gas profiles and related diagnostics during September 2012 in the ASMA region (15° N – 35° N, 30° E – 100° E), and their sensitivity to LiNOₓ emissions. The vertical axes cover the UTLS and middle troposphere, and their coordinates are given as pressure distance to the tropopause. Left column: QCTM reference simulation (q01). Middle column: s03 – q01, relative deviation of sensitivity simulation s03 wrt. q01 for trace gases, absolute deviation for net O₃ production. Right column: s04 – q01.

[Figure]

**Figure B1, As Fig. 4, but focussing on reactive nitrogen.**

[Figure]

**Figure B2. Simulated profiles of NO$_y$ as simulated for 15 August 2012. These are examples of a C-shaped profile in the Iranian region (a) and an E-shaped profile in the Tibetan region (b).**

[Figure]

**Figure B3. As Fig. 3, but focusing on reactive nitrogen. Panels (c) and (f) show NO instead of NO$_x$, because only NO was measured. At daytime, i.e. at the time of the measurements, NO is good proxy for NO$_x$.**

| Page 21: [1] Formatted | Gottschaldt, Klaus-Dirk | 21/12/2017 11:01:00 |

Not Highlight

| Page 21: [1] Formatted | Gottschaldt, Klaus-Dirk | 21/12/2017 11:01:00 |

Not Highlight

| Page 21: [1] Formatted | Gottschaldt, Klaus-Dirk | 21/12/2017 11:01:00 |

Not Highlight

| Page 21: [1] Formatted | Gottschaldt, Klaus-Dirk | 21/12/2017 11:01:00 |

Not Highlight

| Page 21: [1] Formatted | Gottschaldt, Klaus-Dirk | 21/12/2017 11:01:00 |

Not Highlight

| Page 21: [1] Formatted | Gottschaldt, Klaus-Dirk | 21/12/2017 11:01:00 |

Not Highlight

| Page 21: [1] Formatted | Gottschaldt, Klaus-Dirk | 21/12/2017 11:01:00 |

Not Highlight

| Page 21: [1] Formatted | Gottschaldt, Klaus-Dirk | 21/12/2017 11:01:00 |

Not Highlight

| Page 21: [1] Formatted | Gottschaldt, Klaus-Dirk | 21/12/2017 11:01:00 |

Not Highlight

| Page 21: [1] Formatted | Gottschaldt, Klaus-Dirk | 21/12/2017 11:01:00 |

Not Highlight

| Page 21: [1] Formatted | Gottschaldt, Klaus-Dirk | 21/12/2017 11:01:00 |

Not Highlight

| Page 21: [2] Formatted | Gottschaldt, Klaus-Dirk | 21/12/2017 11:01:00 |

Font color: Auto

| Page 21: [2] Formatted | Gottschaldt, Klaus-Dirk | 21/12/2017 11:01:00 |

Font color: Auto

| Page 21: [2] Formatted | Gottschaldt, Klaus-Dirk | 21/12/2017 11:01:00 |

Font color: Auto

| Page 21: [2] Formatted | Gottschaldt, Klaus-Dirk | 21/12/2017 11:01:00 |

Font color: Auto

| Page 21: [2] Formatted | Gottschaldt, Klaus-Dirk | 21/12/2017 11:01:00 |

Font color: Auto

| Page 21: [3] Formatted | Gottschaldt, Klaus-Dirk | 21/12/2017 11:01:00 |

Font color: Auto

| Page 21: [3] Formatted | Gottschaldt, Klaus-Dirk | 21/12/2017 11:01:00 |

Font color: Auto

| Page 21: [3] Formatted | Gottschaldt, Klaus-Dirk | 21/12/2017 11:01:00 |

Font color: Auto

| Page 21: [3] Formatted | Gottschaldt, Klaus-Dirk | 21/12/2017 11:01:00 |

Font color: Auto

| Page 21: [3] Formatted | Gottschaldt, Klaus-Dirk | 21/12/2017 11:01:00 |

Font color: Auto

| Page 21: [3] Formatted | Gottschaldt, Klaus-Dirk | 21/12/2017 11:01:00 |

Font color: Auto

| Page 21: [3] Formatted | Gottschaldt, Klaus-Dirk | 21/12/2017 11:01:00 |

Font color: Auto

| Page 21: [3] Formatted | Gottschaldt, Klaus-Dirk | 21/12/2017 11:01:00 |

Font color: Auto

| Page 21: [4] Formatted | Gottschaldt, Klaus-Dirk | 21/12/2017 11:01:00 |

Font color: Auto

| Page 21: [4] Formatted | Gottschaldt, Klaus-Dirk | 21/12/2017 11:01:00 |

Font color: Auto

| Page 21: [4] Formatted | Gottschaldt, Klaus-Dirk | 21/12/2017 11:01:00 |

Font color: Auto

| Page 21: [4] Formatted | Gottschaldt, Klaus-Dirk | 21/12/2017 11:01:00 |

Font color: Auto

| Page 21: [4] Formatted | Gottschaldt, Klaus-Dirk | 21/12/2017 11:01:00 |

Font color: Auto

| Page 21: [4] Formatted | Gottschaldt, Klaus-Dirk | 21/12/2017 11:01:00 |

Font color: Auto

| Page 21: [4] Formatted | Gottschaldt, Klaus-Dirk | 21/12/2017 11:01:00 |

Font color: Auto

| Page 21: [4] Formatted | Gottschaldt, Klaus-Dirk | 21/12/2017 11:01:00 |

Font color: Auto

| Page 21: [4] Formatted | Gottschaldt, Klaus-Dirk | 21/12/2017 11:01:00 |

Font color: Auto

| Page 21: [4] Formatted | Gottschaldt, Klaus-Dirk | 21/12/2017 11:01:00 |

Font color: Auto

| Page 21: [4] Formatted | Gottschaldt, Klaus-Dirk | 21/12/2017 11:01:00 |

Font color: Auto

| Page 21: [4] Formatted | Gottschaldt, Klaus-Dirk | 21/12/2017 11:01:00 |

Font color: Auto

| Page 21: [4] Formatted | Gottschaldt, Klaus-Dirk | 21/12/2017 11:01:00 |
|---|---|---|

Font color: Auto

| Page 21: [4] Formatted | Gottschaldt, Klaus-Dirk | 21/12/2017 11:01:00 |
|---|---|---|

Font color: Auto

| Page 21: [4] Formatted | Gottschaldt, Klaus-Dirk | 21/12/2017 11:01:00 |
|---|---|---|

Font color: Auto

| Page 21: [4] Formatted | Gottschaldt, Klaus-Dirk | 21/12/2017 11:01:00 |
|---|---|---|

Font color: Auto

| Page 21: [4] Formatted | Gottschaldt, Klaus-Dirk | 21/12/2017 11:01:00 |
|---|---|---|

Font color: Auto

| Page 21: [4] Formatted | Gottschaldt, Klaus-Dirk | 21/12/2017 11:01:00 |
|---|---|---|

Font color: Auto

| Page 21: [4] Formatted | Gottschaldt, Klaus-Dirk | 21/12/2017 11:01:00 |
|---|---|---|

Font color: Auto

| Page 21: [4] Formatted | Gottschaldt, Klaus-Dirk | 21/12/2017 11:01:00 |
|---|---|---|

Font color: Auto

| Page 21: [5] Formatted | Gottschaldt, Klaus-Dirk | 01/12/2017 16:23:00 |
|---|---|---|

Font color: Auto

| Page 21: [5] Formatted | Gottschaldt, Klaus-Dirk | 01/12/2017 16:23:00 |
|---|---|---|

Font color: Auto

[Figure]

[Figure]

**Figure** #+**S1. Enhanced O$_3$ was found in the ASMA during HALO ESMVal and reproduced by EMAC, while Santee et al. (2017) report decreased O$_3$. Here we show EMAC simulated O$_3$ at different isentropic surfaces. Monthly averages are calculated for the same period as in Santee et al. (2017) and agree well with their figures (370 K, 390 K). However, at the level corresponding to the HALO flight altitude (360 K), O$_3$ is enhanced in the ASMA in September (Panel i) and the observed sudden increase at the southern ASMA edge over Oman is also reproduced (arrow). {Livesey, 2013 #149}** Trace gas mixing ratios and net O$_3$ production as simulated by EMAC at 168 hPa for the monsoon months of 2012. The Iranian and Tibetan domains correspond to the regions used for lateral averaging throughout the paper (e.g. Figs. 2 - 5, S4 – S10). The Iranian region was traversed by the HALO ESMVal campaign during a flight from Male (Maledives) to Larnaca (Cyprus) on 18 September 2012. HALO was flying in the upper troposphere where the flight track is colored white and dived to the lower troposphere where it is grey. Beads show the HALO positions at full UTC hours.

[Figure]

[Figure]

**Figure S2.** Evolution of simulated trace gas profiles and related diagnostics for the years 2010 - 2014 in the Tibetan ASMA region (15°N to 35°N , 65° - 100°E). Vertical coordinates are given as distance to the tropopause ("TP"), whose altitude depends on time and location. All values are grid-cell dry air mass weighted averages. The column for 2012 corresponds to Fig. #+3, showing that the trace gas evolution in 2012 was largely similar to other years.

[Figure]

**Figure S3. As Fig. #+S2, but putting Fig. #+4 in a multi-annual perspective. Regarding the shown parameters, 2012 was a normal year.**

32 As Fig. S1, but for different tracers.

[Figure]

[Figure]

Figure #+S4. Evolution of simulated trace gas profiles and related diagnostics throughout 2012 in the ASMA region. Figure S4 is identical to Fig. 3, except that vertical coordinates are given as distance to the tropopause ("TP") here. Compared to the pressure-based vertical axis' used in Fig. 3, the tropopause region is resolved better here — at the expense of the middle troposphere. All values are grid-cell dry air mass weighted averages from 15°N to 35°N, respectively for the western (30° - 65°E) and eastern (65° - 100°E) parts of the ASMA (see Figs. 1, S1, S2, S3). The features marked by circles and arrows are the same as in Fig. 2 and discussed in the text. As Fig. #+S2, but putting Fig. #+A5 in a multi-annual perspective (Panels a and b). There is no noteworthy anomaly for 2012. Panels (c) and (d) show simulated photochemical $O_3$ production and destruction separately. Loss$O_3$ is negligible at and above 355 K, thus the photochemical regime (NO$_x$-limited or NO$_x$-saturated) is determined mainly by Prod$O_3$.

[Figure]

[Figure]

48 **Figure** #+S5. ~As Fig. #+S2, but for the Iranian ASMA region. Of the parameters shown, CO is the most variable one
49 in the UTLS.

51    As Fig. S4, but for different tracers. The individual NO$_y$-profiles shown in Fig. S7 are indicated in panels c and d.

[Figure]

[Figure]

55 **Figure #+S6, As Fig. #+S3, but for the Iranian region.**
56
57

61 **Figure** S7**. As Fig. #+S4, but for the Iranian region.**

63    Simulated profiles of NOy as simulated for 15 August 2012. These are examples of a C-shaped profile in the Iranian
64    region (a) and an E-shaped profile in the Tibetan region (b).

[Figure]

Figure #+S8. Simulated profiles of net $O_3$ production in the Tibetan region. Auxiliary red lines indicate the mean and maximum net $O_3$ production in the UT, which are both higher in summer than in spring.

70  See also Fig. S4h for the evolution of these profiles throughout 2012.

**Comment [KG1]:** revise referencing

72
73 **Figure #+S9. Measured data from the HALO ESMVal flight from Male to Larnaca. In contrast to Figs. #+6cf, #+A7cf**
74 **the samples here are color coded by their potential temperatures. During the dive (HALO descended from the upper**
75 **to the lower troposphere and back to obtain profiles; indicated by crosses) potential temperatures were lower than**
76 **shown, but the color scale is cut off at 340 K.**

[Figure]

78 **Figure S9. As Fig. 3, but for NOₓ vs O₃. Panel (c) shows NO instead of NOₓ, because only NO was measured. At**
79 **daytime, i.e. at the time of the measurements, NO is good proxy for NOₓ. Simulated O₃ and NOₓ increase in the**
80 **stratosphere with a higher $O_3/NO_x$ ratio than in the troposphere (Fig. S9a). At NOₓ mixing ratios of more than 0.7**
81 **nmol/mol the corresponding O₃ mixing ratios would allow distinguishing stratospheric influence from tropospheric in**
82 **situ production, but the range covered by the HALO ESMVal measurements is just at the intersection of stratospheric**

[Figure]

[Figure]

[Figure]

94 **Figure #+S10.** EMAC-simulated monthly mean lightning activity (intra-cloud + cloud-to-ground flash frequency)
95 compared to the corresponding TRMM-LIS/OTD observations (Cecil, 2006)(Cecil, 2006). Data coverage and color
96 scale are determined by the observations. Simulated lightning appears to be more localized, and thus exceeds the scale
97 more often.
98 Convection is not explicitly resolved in the simulation, and the parameterizations for convection and lightning both
99 introduce uncertainties to the simulation results for lightning. Uncertainties in the observations are due to a space-
100 and time-dependent detection limit of 69% to 88%, and the application of a 3 month smoothing. Considering those
101 uncertainties, the match between simulated and observed global distribution and frequency of lightning activity is
102 reasonable.

[Figure]

105 **Figure #+S11. EMAC-simulated three-month mean curtains of CO mixing ratios: (a) Covering Iranian and Tibetan**

106 **parts, meridional mean; (b) Tibetan part, zonal mean. On average, the hotspot of ascending CO in our simulation is**

107 **located at about 29°N, 80°E, corresponding to the south-western flank of the Himalayas.**

As Fig. S9, but for $NO_x$ vs $NO_y$. There are three distinct regions in Fig. S10a: a blueish stratospheric branch, a dark TL branch, and a reddish UT region. As a consequence of the local $NO_y$ minimum directly above the tropopause (Figs. 2d, S5, S7), the most decreased $NO_x$ mixing ratios in Fig. S10a also show up in samples taken from near the tropopause. Measured NO and $NO_y$ values in the ASMA filament are well correlated (Fig. S10c), consistent with almost constant NOx/NOy ratios in the UT (Figs 2ef). Furthermore, the simulation shows much more scatter in $NO_x$-vs-$O_3$ space than the observations. The narrow, linear distribution of the ASMA measurements in Figs. S9c and S10c indicates that all parts of the transected filament had similar sources of reactive nitrogen. This is consistent with Appendix A, where lightning is found to be the dominating source of reactive nitrogen in the ASMA. In the accompanying study is also shown that the filament had seen convection at the eastern ASMA flank three to five days before the measurements. Thus the gradients of NO and $NO_y$ in Fig. S9c and Fig. S10c can be explained by different amounts of lightning $NO_x$ of approximately the same age.

[Figure]

123

124

125

126 **Figure** +**S12. EMAC-simulated monthly mean distributions of CO (left) and deep convective mass flux (middle)**

127 **in different pressure altitudes during spring (April 2012). The right column shows the deep convective mass flux of**

128 **CO based on individual output steps. Blue rectangles mark the outline of the regions used to produce Figs. #+3 and**

129   #+4.  Panels a, b, c, d show measured data from the HALO ESMVal flight from Male to Larnaca as Figs. 4c, 5c, S9c,
130   S10c, respectively. The only difference is that the samples here are color coded by their potential temperatures.
131   During the dive (POI4: crosses) potential temperatures were lower than shown, but the color scale is cut off at 340 K.
132   See also Figs. S12 and S13 for simulated potential temperatures in the ASMA region.
133

134

[Figure]

135

136 **Figure S12#+S13. As Fig. #+S12, but for August 2012, i.e. during the monsoon season.**
137 **EMAC simulated potential vorticity at the 360 K isentropic level (panels a, b) and the 168 hPa pressure level (panels c,**
138 **d). The black circle in panel c shows the entrainment of TL air, which occurred when the air mass corresponding to**
139 **POI3 was passing the eastern ASMA flank. Panels b, d, f show simulated snapshots 2 hours before POI3, with the**
140 **region of POI3 marked in panel d. The corresponding potential temperatures are shown in panels e and f. Note that**
141 **TL layer entrainments are visible in PV, but are hardly detected by Tpot.**
142

143
144

**Figure S13. EMAC simulated potential temperature and potential vorticity in curtains at 100°E, at the time when the air corresponding to POI3 was passing there at about 165 hPa. Note the steeply inclining TP over the Tibetan plateau, which marks the transition from the extratropics (dominated by baroclinic wave activity and downward stratospheric circulation) to the tropics (dominated by radiative-convective balance and upward stratospheric circulation). North of 30°N the EMAC TP is defined by the 3.5 PVU isocontour. Isentropes intersect the inclining TP, thereby allowing cross-TP transport without leaving a tell-tale signature of increased Tpot in the corresponding air masses in the tropics. This includes the 350 — 370 K isentropes, that were encountered during the HALO ESMVal campaign in the tropics (see also Figs. S11, S12).**

[Figure]

154    355 K

155

156    **Figure** **4. EMAC simulated HCl mixing ratios at 168 hPa in the ASMA region, complementing Fig. #+7. The**

157    **snapshots were selected to represent independent situations, where the southern ASMA fringe is marked by a**

[Figure]

162
163    **Sequence of O₃ mixing ratios in the ASMA region and streamlines as simulated by EMAC for a layer 20 hPa below**
164    **the tropopause. This layer was chosen to illustrate O₃ variations and transport just below the TP. EMAC tropopause**
165    **height varies spatially and temporally, as it is diagnosed each time step, according to the WMO definition between**
166    **30°S and 30°N, and by PV = 3.5 PVU otherwise. Note that the layer may not, or may only partially show altitudes that**
167    **contributed to the HALO measurements. The streamlines are based on instantaneous wind fields and thus not**
168    **identical to backward trajectories. Grey arrows indicate a pocket of increased O₃, which originated in the tropopause**
169    **folding region over the Eastern Mediterranean. It is picked up by the ASMA circulation and transported along the**
170    **northern ASMA flank. The pocket passes the eastern ASMA flank before the time (last panel), when the air mass to**
171    **be encountered by HALO arrived there (region indicated by black circles). However, a part of the increased O₃ patch**
172    **might have been entrained in the divergent flow there, diverted away from the TP and carried along the southern**
173    **ASMA flank back east.**

[Figure]

174

175 **Figure #+S15. Selection of parameters related to photochemical O$_3$ production in a meridional curtain through the**

176 **Tibetan part of the ASMA in a snapshot taken mid August 2012. The left column shows the results of the EMAC**

177 **QCTM simulation that has been introduced in appendix A. The other columns show the difference of that reference to**

178 **sensitivity simulations, which feature identical dynamics but differ in biomass burning (BB) and lightning NO$_x$**

179 **(LiNOx) emissions. The black lines represent the 13 nmol mol$^{-1}$ day$^{-1}$ isocontour of net O$_3$ production (taken from**

180 **panel γ), the grey line denotes the tropopause.**

181

Comment [KG2]: JP3: I thought the QCTM reference was also with daily BB emissions??? But maybe I do not remember correctly.

KG: This refers to the difference between the QCTM reference simulation ("base") and the corresponding QCTM sensitivity simulations (here: no BB, 2 x BB). No ESCiMo simulation was analyzed for this figure.

[Figure]

**Figure #+S16. Sequence simulated tracer fields at 355 K, illustrating the stirring associated with the splitting-up event of the ASMA that occurred during the HALO ESMVal campaign in September 2012. Streamlines represent instantaneous wind fields, and arrows highlight the redistribution of selected air masses. CO mainly originates in the ASMA interior and HCl serves as a proxy to track the ASMA fringe.**

**The sequence starts with an elongated anticyclone on 16 September 2012. Then a tropopause trough (T) evolves from the west along the northern ASMA flank. The anticyclone succumbs to the perturbation and splits up into a Tibetan and an Iranian part, shortly after the HALO flight from Male to Larnaca had passed through. A part of the increased CO interior region is entrained by the outer streamlines of the Iranian part, while the rest of the patch is diverted into the interior of the Tibetan anticyclone (black arrows in the left panels). The evolution of freshly entrained HCl (black arrows) and an older patch (red arrows) are shown in the right panels. We also note entrainment of tropospheric air by southerly winds at the western flank (white arrows).**

**Figure S15. Evolution of meridional wind fraction, as averaged over 10°N – 40°N and the respective altitude range. The results are based on 10-hourly output of EMAC simulation RC1SD-base-10a for September 2012 in the ASMA region. Weighting considers the dry air mass in the cell, and completely or partly stratospheric cells are ignored. At a given time, each red-blue-pair (from left to right) represents an anticyclone, because positive values indicate overall northward meridional wind fractions. Blue shades represent mainly southward wind fractions.**

[Figure]

**Figure S16. EMAC simulated HCl mixing ratios at 168 hPa in the ASMA region, complementing Fig. 6. The snapshots were selected to represent independent situations, where the southern ASMA fringe is marked by a filament of enhanced HCl. The filaments are often associated with a TP trough at the eastern ASMA flank. Enhanced HCl serves as a proxy for TL or stratospheric air.**

[Figure]

**Figure #+S177.** EMAC simulated O$_3$ mixing ratios and streamlines at 168 hPa in the ASMA region. The snapshots are 20 hours apart and cover the period from 1 August to 18 September 2012. Larger fractions of the O$_3$-rich fringe were entrained during splitting up events. The sequence of snapshots (Fig. #+S17) covers almost half a monsoon season and episodic O$_3$-poor upwellings over the Tibetan plateau are smaller and shorter lived than O$_3$-rich regions at 168 hPa. This is consistent to the long-term average at a corresponding isentropic level (Fig. #+S1i).

[Figure]

Figure S18. Evolution of simulated trace gas profiles and related diagnostics for the years 2010 - 2014 in the Iranian ASMA region. The column for the year 2012 is identical to the corresponding panels in Figs. S4, S5.

[Figure]

Figure S19. Evolution of simulated trace gas profiles and related diagnostics for the years 2010 - 2014 in the Tibetan ASMA region. The column for the year 2012 is identical to the corresponding panels in Figs. S4, S5.

---

## Author Response (AR2)

We are pleased to inform you that the Co-Editor report for the following manuscript is now available:

Journal: ACP
Title: Dynamics and composition of the Asian summer monsoon anticyclone
Author(s): Klaus-Dirk Gottschaldt et al.
MS No.: acp-2017-420
MS Type: Research article
Iteration: Minor Revision
Special Issue: The Modular Earth Submodel System (MESSy) (ACP/GMD inter-journal SI)

The Co-Editor has decided that minor revisions are necessary before the manuscript can be accepted. Please find the Co-Editor Report at https://editor.copernicus.org/ACP/ms_records/acp-2017-420.

We kindly ask you to revise your manuscript accordingly and to upload the revised files, a point-by-point reply to the comments, and a marked-up manuscript version showing the changes made in your File Manager no later than 17 Feb 2018: https://editor.copernicus.org/ACP/file_manager/acp-2017-420. Please find all information on manuscript submission under https://www.atmospheric-chemistry-and-physics.net/for_authors/submit_your_manuscript.html.

Your revised manuscript will be reviewed by the Co-Editor and you will be informed about the outcome by separate email.

Besides adjustments requested by the Co-Editor or Referees, please check your manuscript carefully for typos, missing co-authors and their affiliations, terminology, updates of data in tables, or updates of variables in equations. All these have to be clarified with the Co-Editor and therefore have to be included before you submit your revised manuscript. Should your manuscript be finally accepted it will not be possible to include such rather substantial changes anymore when your manuscript is in final production (proofreading).
* * *
**Co-Editor Decision: Publish subject to minor revisions (review by editor)** (07 Feb 2018) by Marc von Hobe
Comments to the Author:
Generally, the structure of the main text is quite good. The storyline through the sections and subsections appears logical, the main findings are nicely summarized in Section 7, and some lengthy "side excursions" have been removed.

We thank the Co-Editor for his review and helpful comments. Our changes to the manuscript are highlighted in the markup version, with color coding according to Fig. 1. Please note that newly inserted references and modified figure numbers are not always highlighted (for technical reasons). Table 1 summarizes the renumbering of figures instead. Changes to citations are noted in our responses to your comments below.

However, the paper is still moving on the edge of being "information overload", especially with the large amount of cross referencing to the Appendices and Supplement. Some of this extra material is important and well justified, but in some cases, I'm not sure if it is really necessary and if the placement in an Appendix/Supplement is the correct choice. Please consider the following specific remarks and suggestions in that respect:

Appendix A: from the title of the Appendix, this is not different from Section 6.1, and some of the extra discussion of processing inside the ASMA could be blended into that Section in the main text. But much of the material, such as the comparison to CARIBIC and Satellite observations and the model sensitivity studies seem more appropriate in the supplement (some of the figures are

already there, e.g. Figure S10) to convince those readers who have doubts about the model lighting NOx.

Done.

Appendix B: I think this could be shortened a bit and added to the main text. Figures B1 and B3 can be added to Figures 4 and 3, and the discussion added to Sections 4, 5 and 6, maybe by adding subsections. I don't see that NOy is so much less important than the other material discussed in the main text, and I don't think it will disrupt the main text too much. You mention all tracer-tracer relations in the first sentence of Section 4.1 and state that some are shown in that Section and others in the Appendix. But a reasoning why the NOx relations are not discussed in the main text is not given.

Done.

Appendix C: does this "primer" contain any information that is not available in the literature? I don't think it is really needed and suggest referring to one or a few papers (or even books) where tracer relations are introduced in some detail.

We found no basic text-book description, just applications of the method in papers. Moved to the supplement and added two references.

With respect to the supplementary figures, some of them are referred to and discussed in the main text in just the same way as normal figures (e.g. page 11, line 26, page 15, line 20, and quite a few others), and I don't see any harm including some of them in the main part of the paper. Extra figures typically don't disrupt the story line, and there is not an official limit with respect to the number of Figures in a paper. I'm thinking that especially Figures S1, S11 and S16 would be nice to have in the main part of the paper: they are very interesting and highly relevant and do not just provide extra documentation and reproducibility.

We agree. Figures S1, S11 and S16 are now part of the main text.

For the multi-year information, data comparisons and model sensitivities, these are indeed appropriately placed in the supplement if you do want to show all of them. Maybe you can add a bit of discussion in the actual supplement, e.g. move some of the material from Appendix A into the supplement, so it becomes a little more independent and rounds up all "extra topics". In fact, you may want to consider giving the supplement a bit of structure, e.g. model comparison to satellite data, inter-annual variability, process sensitivities in the model. In that case, the referencing in the main text to this "extra material" becomes much clearer.

This is a good idea, the supplement is structured now.

Besides considering the suggested rearrangements into main text and supplements, the following technical corrections need to be made prior to final publication:

The in-text referencing of the figures is not in the correct order (e.g. Figure 2 should be referred to earlier than Figure 7). In particular, Figures discussed in detail later are referred to in quite random order in Section 3. Please either renumber your figures, or adjust the text so that the referring is in the correct order.

The order of figures was intended to reflect the storyline. Now they are ordered according to first referral in the text (Table 1).

On the same note, Section 5 discusses in detail what is seen in Figure 4, but the subsections 5.1 to 5.4 correspond to e/f, c/d, a/b and g/h respectively. I suggest changing the order of the gases

shown in the Figure according to the Section ordering, i.e. O3 on top, then CO, then HCl and then NOx.

Done.

Page 6, line 4: I don't understand the word "enhanced" here. If you refer to the vertical gradients in O3 itself, then please delete "Enhanced". If you mean something else, then please explain.

Reformulated.

Page 6, lines 13 and 16: please add some literature references that HCl is a good tracer for stratospheric O3 entrainments and that daytime NO is a good proxy for NOx.

HCl: The reference to Marcy et al. (2004) was intended for the whole sentence, but placing it at the end might indeed be misleading. We moved the reference to Marcy et al. to the first part of the sentence and added the reference to Park et al (2008), who used HCl as stratospheric tracer in the ASMA.

NOx: We restricted the statement to the UT, where daytime NO/NO2 is about 12 (Seinfeld & Pandis, 2006).

Page 9, line 20: the circled area looks more like 300 to 500 hPa below the TP rather than 200 to 300 hPa. Please check this and correct or clarify!

Corrected.

Page 13, line 14: should be ("2" in Fig. 6) instead of ("2" in Fig. 4)

Corrected.

Caption Figure 3: it should be (b,e) and (c/f) in lines 4 and 6

Corrected.

Caption Figure 4: please remove the #+ in front of the Figure numbering

Done, also in the footnote on page 12.

Caption Figure 6, line 9: between 200 and 700 ppb NOx seems incredibly high. Please check and correct this!

Corrected to pmol mol$^{-1}$. We also changed "mol/mol" to "mol mol$^{-1}$" throughout.

| old | new | old | new | old | new | old | new |
|---|---|---|---|---|---|---|---|
| 1 | 2 | A1 | 10 | S2 | S4 | S10 | S2 |
| 2 | 4 | A2 | S3 | S3 | S5 | S11 | 11 |
| 3 | 8 | A3 | S12 | S4 | S6 | S12 | S14 |
| 4abcdef 4gh | 5efcdab 6ab | A4 | S13 | S5 | S7 | S13 | S15 |
| 5 | 7 | B1abcd | 6cdef | S6 | S8 | S14 | S10 |
| 6 | 12 | B2 | S11 | S7 | S9 | S15 | S17 |
| 7 | 3 | B3 | 9 | S8 | S16 | S16 | 13 |
| 8 | 14 | S1 | 1 | S9 | S1 | S17 | S18 |

Table 1: Figure numbering; old = manuscript version 20171222, new = 20180221

[Figure]

Figure 1: Markup for changes in the manuscript.

[revised manuscript text omitted]

The discussion of the evolution of the $LiNO_x$ profile (Fig. 5) is supplemented by the corresponding lateral distribution of monthly mean $LiNO_x$ emission rates at the altitude of the HALO ESMVal measurements (Fig. A1). We note that there are strong, but localised emissions in the Iranian and Tibetan parts in spring (Fig. A1: Apr May). $LiNO_x$ emissions are distributed throughout the Tibetan region in summer (Fig. A1: Jul Sep). The simulated spatio temporal emission patterns are similar for 2013 and 2014 (not shown).

Given the uncertainties of the parameterizations for convection and lightning in the model, smoothing and limited detection efficiencies in the observations, our simulated spatial temporal distribution of lightning activity compares reasonably well to corresponding observations (supplement, Fig. S10). In particular we note that also the observations over South Asia show stronger lightning activity during spring than during the monsoon season. The observed maximum of lightning activity over the coastal areas of Western Bengal and Bangladesh in April also shows up in the simulation.

As noted in section 2, we compare simulated NO and $NO_y$ to the corresponding IAGOS CARIBIC observations (Fig. A2). Commercial airliners do not fly as high as HALO and the tracks hardly reach the southern ASMA fringe, but the northern ASMA edge and the center of the monsoon region have been sampled multiple times. We evaluated all 345 IAGOS CARIBIC flights between 19 May 2005 and 9 April 2014, considering the respective latest data versions as of 10 November 2017. In total 86 flights between Frankfurt (IATA code: FRA) and Chennai (MAA) or Guangzhou (CAN) or Bangkok (BKK) transected the ASMA region. 32 of these flights provide NO data there, and 66 flights provide $NO_y$. Neglecting data below 300 hPa and subsampling to the time resolution of the simulation yields the numbers of comparable data that are given in Figs. A2bd. Given the above uncertainties related to the representation of $LiNO_x$ in the simulation, NO matches the corresponding IAGOS CARIBIC observations surprisingly well (Fig. A2b). This holds also for the more robust (more data)

global comparison (Fig. A2a). Increased LiNO$_x$ emissions in the ASMA region in spring are also consistent to IAGOS-CARIBIC, and the simulation might even slightly underestimate those emissions (Fig. A2b: MAM).

In order to link the LiNO$_x$ emissions to the NO$_x$ burden in the ASMA region, a suite of EMAC sensitivity simulations with modified emission factors was conducted (Figs. A3, A4). All EMAC analyses in the main text are based on simulation RC1SD-base-10a (Jöckel et al., 2016), which is given in Fig. A3 just for comparison. The other simulations discussed in the context of Figs. A3 and A4 here are derived from EMAC simulation RC1SD-base-10, which differs in road traffic emissions and optical properties of stratospheric aerosol (Jöckel et al., 2016) from RC1SD-base-10a. Total LiNO$_x$ emissions in RC1SD-base-10 are 4.6 Tg(N) in 2012 (Jöckel et al., 2016), which is in the realistic range of 2 – 8 Tg(N) yr$^{-1}$ (Schumann and Huntrieser, 2007). RC1SD-base-10 and our base simulation for the LiNO$_x$ sensitivity analysis (b01) are both operated in chemistry climate model (CCM) mode, i.e. including interactive chemistry with feedback on dynamics. Simulation b01 differs only in the usage of daily (Kaiser et al., 2012) instead of monthly biomass burning emissions and 5 h instead of 10 h output intervals. Feedbacks from chemistry on dynamics in all quasi chemistry transport model mode (QCTM) (Deckert et al., 2011) simulations are based on identical trace gas time series from b01. The same dynamics incl. convection is simulated in all QCTM simulations. Differences between a QCTM reference simulation (q01) and sensitivity simulations (s*) are thus exclusively due to chemical perturbations. All QCTM simulations cover June – September 2012, but the first 3 months were discarded for spin up.

Figure A3 shows that RC1SD-base-10a captures observed O$_3$, CO, NO and NO$_y$ along the HALO ESMVal flight path slightly better than b01 and q01. We are yet confident that the overall agreement is good enough for the analysis of chemical perturbations. For the QCTM sensitivity analyses it is more important to note that differences between b01 and q01 are negligible.

Figure A4k shows that halving LiNO$_x$ emission factors results in almost halved NO$_x$ in the uppermost troposphere. Doubling of LiNO$_x$ emissions leads to almost doubled NO$_x$ just below the tropopause (Fig. A4m). The biggest relative sensitivity in Fig. A4km almost coincides with the altitude range of the largest NO$_x$ mixing ratios just below the tropopause (Fig. A4j). Thus, in our simulations LiNO$_x$ clearly dominates the NO$_x$ budget from the tropopause to 100 hPa below it. The impact of LiNOx fades out at lower altitude, and almost vanishes at 400 hPa below the tropopause. This is consistent with the profiles of LiNO$_x$ emissions in September 2012, which mainly occur in the Tibetan part of the ASMA (Fig. 5d).

Modifications of NO$_x$ print through on other O$_3$ precursors mainly via changes to the atmospheric oxidizing capacity (OH: Figs. A4ghi). In response to halved LiNO$_x$, OH decreases 200 hPa below the tropopause and lower, and increases above (Fig. A4h). The effects are reversed for doubled LiNO$_x$ (Fig. A4i). The largest relative effects coincide with largest absolute OH mixing ratios.

CO decreases throughout the shown altitude range for halved LiNO$_x$ (Fig. A4e). Without major production terms in the UT, modifications to CO mixing ratios are dominated by the loss reaction CO + OH → H + CO$_2$. The rate coefficient of this reaction is proportional to pressure, and otherwise depends only on constants (supplement to Jöckel et al. (2016)). Laterally averaged CO mixing ratios vary little from 50 to 400 hPa below the tropopause (Fig. A4d), but are affected by decreased and

increased OH (Figs. A4fg). Decreased OH in the lower half of the domain dominates the overall CO response. CO rises through this region with higher pressures before reaching the UT in the Tibetan part of the ASMA (Fig. 4), which obviously outweighs the CO response to increased OH in the UT of Tibetan and Iranian part combined. Increased OH 200 – 500 hPa below the tropopause consequently leads to an overall decrease of CO in response to doubled $LiNO_x$ (Fig. A4f). The $O_3$ precursors $NO_x$ and CO display opposite trends in response to $\Delta LiNO_x$.

Curiously, HCl shows the opposite response to modified $LiNO_x$ (Figs. A4abc). There is no chemical production of HCl in the UT, and the only loss term in the simulations is $HCl + OH \rightarrow Cl + H_2O$. The rate coefficient of this reaction is 1.7E-12 * EXP( 230/Temperature), see supplement to Jöckel et al. (2016). However, the tropospheric response of HCl to $\Delta OH$ is dominated rather by the vertical profile of HCl mixing ratios than by lower temperatures towards the tropopause. Almost all HCl in the UT is of stratospheric origin, and HCl mixing ratios steeply increase across the tropopause. Thus the UT response of HCl is dominated by $\Delta OH$ near the tropopause: increased OH for halved $LiNO_x$ increases HCl losses, and vice versa for doubled $LiNO_x$.

The response of UT net $O_3$ production to $\Delta LiNO_x$ (Figs. A4qrs) has mostly the same sign as $\Delta NO_x$. As noted already in the context of Fig. 4, opposite gradients of $O_3$ precursors $NO_x$ and CO in the UT lead to a broad altitude range of enhanced net $O_3$ production in the ASMA, centred about 100 hPa below the tropopause. $O_3$ production is limited by $NO_x$ in lower altitudes and by CO (and other volatile organic compounds) towards the tropopause. $NO_x$ and CO display opposite trends in response to $\Delta LiNO_x$, but relative changes to $NO_x$ are larger and dominate the overall response of net $O_3$ production. We note, however, that the largest increase of $NO_x$ at the end of September (circled in Fig. A4m) decreases net $O_3$ production to zero or even net loss (circled in Fig. A4s), indicative of the $NO_x$-limited photochemical regime.

$O_3$ mixing ratios respond to $\Delta LiNO_x$ essentially like net $O_3$ production in the UT (Figs. A4nop). The altitude of maximum relative $\Delta O_3$ is slightly lower than the altitude of maximum absolute changes to net $O_3$ production. We attribute this effect to upwards increasing absolute $O_3$ mixing ratios.

**Appendix B: Reactive nitrogen in the ASMA**

Nitrogen oxides are key parameters in atmospheric chemistry, partly controlling the $O_3$ production in the troposphere and lower stratosphere. In the UTLS, enhanced $NO_y$ originates both from tropospheric and from stratospheric sources. In the lower troposphere odd nitrogen species are co-emitted with carbon monoxide in combustion processes, resulting in a strong correlation between both species. $NO_y$ is enhanced in the stratosphere (mainly $HNO_3$), but it also comprises species with tropospheric sources ($LiNO_x$). Thus it is not a viable tracer for stratospheric air on its own.

The simulation matches IAGOS-CARIBIC $NO_y$ almost perfectly on the global scale (Fig. A2e), and only moderately overestimates it during summer in the ASMA region (Fig. A2d: JJA).

[revised manuscript text omitted]

Figure A2. Comparison of IAGOS-CARIBIC (black) measurements in the altitude range between 300 hPa and the TP ("Tr") with corresponding results of the EMAC RC1SD-base-10a simulation (red) for (a) NO globally, (b) NO in the ASMA region (15-35°N, 30-100°E), (c) NO$_y$ globally, (d) NO$_y$ in the ASMA region. All data stem from the period May 2005 to April 2014. The simulation was sampled along the IAGOS-CARIBIC flight tracks with a resolution of 12 min, and IAGOS-CARIBIC observations were subsequently interpolated (interval mean) to a resolution of 12 min. Numbers n below the plots show the number of the remaining data pairs (after interpolation and filtering) available for the respective seasons. Dots represent mean values, whiskers indicate standard deviation, min & max values. Rectangles represent the median, and whiskers the percentiles 5, 25, 75, 95.

[Figure]

Figure A3. Mixing ratios of O₃, CO, NO and NOy along the HALO flight track from Male to Larnaca, on 18 September 2012. Grey shading marks the first flight section in ASMA air. Grey line: in situ measurements in 10 s resolution, black: in situ averaged to 12 min simulation time steps, R10a: EMAC simulation RC1SD-base-10a. Sensitivity simulations are based on the almost identical RC1SD-base-10 simulation of (Jöckel et al., 2016), feature daily instead of monthly biomass burning emissions, and were performed in quasi chemistry transport model mode (Deckert et al., 2011) to facilitate isolating the effects of modified emissions. b01: as R10a, but with different traffic and different biomass burning emissions; q01: as b01, but QCTM; s03: as q01, but halved LiNO_x emissions; s04: as q01, but doubled LiNO_x emissions; s05: as s03, but with a different vertical emission profile of LiNO_x (emission factors not decreased in the mid-troposphere, i.e. no C-shape)

[Figure]

**Figure A4.** Evolution of simulated trace gas profiles and related diagnostics during September 2012 in the ASMA region (15° N – 35° N, 30° E – 100° E), and their sensitivity to LiNO$_x$ emissions. The vertical axes cover the UTLS and middle troposphere, and their coordinates are given as pressure distance to the tropopause. Left column: QCTM reference simulation (q01). Middle column: s03 – q01, relative deviation of sensitivity simulation s03 wrt. q01 for trace gases, absolute deviation for net O$_3$ production. Right column: s04 – q01.

[Figure]

Figure B1. As Fig. 4, but focussing on reactive nitrogen.

[Figure]

**Figure B2. Simulated profiles of NO$_y$ as simulated for 15 August 2012. These are examples of a C-shaped profile in the Iranian region (a) and an E-shaped profile in the Tibetan region (b).**

[Figure]

**Figure B3.** As Fig. 3, but focusing on reactive nitrogen. Panels (c) and (f) show NO instead of $NO_x$, because only NO was measured. At daytime, i.e. at the time of the measurements, NO is good proxy for $NO_x$.

**Contents**

**S1 Tracer-tracer diagrams**

**S1.1 Primer**

Tracer-tracer diagrams show mixing ratios of two species encountered simultaneously and are a standard method for analyzing mixing processes in the UTLS region (e.g. Zahn et al., 2000; Vogel et al., 2011). Sampling of two different air masses that are in the process of mixing is indicated by a mixing line in the tracer-tracer diagram. The slope of the mixing line provides additional clues about the origin of the original air parcels ("end-members"). If the ratios of the end members remain constant over time, the slope of the mixing line is conserved, as long as the mixing process continues. If the mixing processes stops, the mixing lines converge to a single point in the tracer-tracer diagram. If the reservoir of one end-member is bigger than the other, points in the tracer-tracer diagram will be close to the dominating end-member. However, the relative size of the reservoirs does not affect the slope of mixing lines, thus allowing detection of even small entrainments. Slopes change in case of mixing ratio changes over time (e.g. via in situ production or loss) of one or both reservoirs. Different effects may lead to similar tracer-tracer relations, resulting in ambiguity when trying to reconstruct end-members or disentangling mixing and chemical effects. Furthermore, mixing lines in general exist in a multi-dimensional tracer space, and thus lines in a tracer-tracer plot need to be considered as projections onto 2d space. They might also be the result of mixing between more than two reservoirs. Additional dimensions (tracers) need to be considered to reduce ambiguities.

**S1.2 Potential temperatures of the observations**

[Figure]

**Figure S1. Measured data from the HALO ESMVal flight from Male to Larnaca. In contrast to Figs. 8cf, 9cf the samples here are color coded by their potential temperatures. During the dive (HALO descended from the upper to the lower troposphere and back to obtain profiles; indicated by crosses) potential temperatures were lower than shown, but the color scale is cut off at 340 K.**

**S2 Observations and model evaluation**

**S2.1 Simulated versus observed lightning activity**

Convection is not explicitly resolved in the simulation, and the parameterizations for convection and lightning both introduce uncertainties to the simulation results for lightning. Uncertainties in the observations are due to a space- and time-dependent detection limit of 69% to 88%, and the application of a 3 month smoothing. Considering those uncertainties, the match between simulated and observed global distribution and frequency of lightning activity is reasonable (Fig. S2). In particular we note that also the observations over South Asia show stronger lightning activity during spring than during the monsoon season. The observed maximum of lightning activity over the coastal areas of Western Bengal and Bangladesh in April also shows up in the simulation.

[Figure]

[Figure]

**Figure S2. EMAC-simulated monthly mean lightning activity (intra-cloud + cloud-to-ground flash frequency) compared to the corresponding TRMM-LIS/OTD observations (Cecil, 2006). Data coverage and color scale are determined by the observations. Simulated lightning appears to be more localized, and thus exceeds the scale more often.**

~~Convection is not explicitly resolved in the simulation, and the parameterizations for convection and lightning both introduce uncertainties to the simulation results for lightning. Uncertainties in the observations are due to a space- and time-dependent detection limit of 69% to 88%, and the application of a 3 month smoothing. Considering those uncertainties, the match between simulated and observed global distribution and frequency of lightning activity is reasonable.~~

**S2.2 Comparison of simulated NO and NO$_y$ to IAGOS-CARIBIC observations**

As noted in section 2, we compare simulated NO and NO$_y$ to the corresponding IAGOS-CARIBIC observations (Fig. S3). Commercial airliners do not fly as high as HALO and the tracks hardly reach the southern ASMA fringe, but the northern ASMA edge and the center of the monsoon region have been sampled multiple times. We evaluated all 345 IAGOS-CARIBIC flights between 19 May 2005 and 9 April 2014, considering the respective latest data versions as of 10 November 2017. In total 86 flights between Frankfurt (IATA code: FRA) and Chennai (MAA) or Guangzhou (CAN) or Bangkok (BKK) transected the ASMA region. 32 of these flights provide NO data there, and 66 flights provide NO$_y$. Neglecting data below 300 hPa and subsampling to the time resolution of the simulation yields the numbers of comparable data that are given in Figs. S3bd. Given the above uncertainties related to the representation of LiNO$_x$ in the simulation, NO matches the corresponding IAGOS-CARIBIC observations surprisingly well (Fig. S3b). This holds also for the more robust (more data) global comparison (Fig. S3a). Increased LiNO$_x$ emissions in the ASMA region in spring are also consistent to IAGOS-CARIBIC, and the simulation might even slightly underestimate those emissions (Fig. S3b: MAM).

The simulation matches IAGOS-CARIBIC NO$_y$ almost perfectly on the global scale (Fig. S3c), and only moderately overestimates it during summer in the ASMA region (Fig. S3d: JJA).

[Figure]

**Figure S3.** Comparison of IAGOS-CARIBIC (black) measurements in the altitude range between 300 hPa and the TP ("Tr") with corresponding results of the EMAC RC1SD-base-10a simulation (red) for (a) NO globally, (b) NO in the ASMA region (15-35°N, 30-100°E), (c) NO$_y$ globally, (d) NO$_y$ in the ASMA region. All data stem from the period May 2005 to April 2014. The simulation was sampled along the IAGOS-CARIBIC flight tracks with a resolution of 12 min, and IAGOS-CARIBIC observations were subsequently interpolated (interval mean) to a resolution of 12 min. Numbers n below the plots show the number of the remaining data pairs (after interpolation and filtering) available for the respective seasons. Dots represent mean values, whiskers indicate standard deviation, min & max values. Rectangles represent the median, and whiskers the percentiles 5, 25, 75, 95.

**S3 Inter-annual variability**

**S3.1 Evolution of trace gas profiles in the Tibetan region 2010-2014**

[Figure]

**Figure S4.** **Evolution of simulated trace gas profiles and related diagnostics for the years 2010 - 2014 in the Tibetan ASMA region (15°N to 35°N , 65° - 100°E). Vertical coordinates are given as distance to the tropopause ("TP"), whose altitude depends on time and location. All values are grid-cell dry air mass weighted averages. The column for 2012 corresponds to Figs. 5, 6, 7, showing that the trace gas evolution in 2012 was largely similar to other years.**

[Figure]

**Figure S5.** **As** Fig. S4**, but putting** Fig. 7 **in a multi-annual perspective. Regarding the shown parameters, 2012 was a normal year.**

[Figure]

**Figure S6. As Fig. S4, but putting Fig. 6 in a multi-annual perspective (Panels a and b). There is no noteworthy anomaly for 2012. Panels (c) and (d) show simulated photochemical O₃ production and destruction separately. LossO₃ is negligible at and above 355 K, thus the photochemical regime (NOₓ-limited or NOₓ–saturated) is determined mainly by ProdO₃.**

**S3.2 Evolution of trace gas profiles in the Iranian region 2010-2014**

Figure S7. As Fig. S4, but for the Iranian ASMA region. Of the parameters shown, CO is the most variable one in the UTLS.

[Figure]

**Figure S8. As Fig. S5, but for the Iranian region.**

[Figure]

**Figure S9. As Fig. S6, but for the Iranian region.**

**S3.3 TL entrainment 2010-2014**

[Figure]

**Figure S10.** EMAC simulated HCl mixing ratios at 168 hPa in the ASMA region, complementing Fig. 3. The snapshots were selected to represent independent situations, where the southern ASMA fringe is marked by a filament of enhanced HCl. The filaments are often associated with a TP trough at the eastern ASMA flank. Enhanced HCl serves as a proxy for TL or stratospheric air.

**S4 Trace gas profiles**

**S4.1. Simulated NO$_y$ profiles**

NO$_y$-profiles with maxima in the lower troposphere, in the UT and in the lower stratosphere dominate the Tibetan part (Fig. S11b). E-shaped NO$_y$-profiles in northern mid-latitudes (Grewe et al., 2001) were in part attributed to aviation NO$_x$ emissions (Rogers et al., 2002), but aviation effects are much smaller in the tropics (Gottschaldt et al., 2013). Instead of aviation emissions, in situ production of lightning NO$_x$ in the prevalent thunderstorms of the monsoon season increases NO$_y$ in the UT over South Asia (Figs. 6d, 7d).

Photochemical production of HNO$_3$ and thus NO$_y$ mixing ratios also increase with altitude above the tropopause (Seinfeld and Pandis, 2006). However, NO$_y$ mixing ratios in the region between the tropopause and 15 hPa above the tropopause are often smaller than in the adjacent altitudes. There is little in situ production and not much transport from above or below. The maximum in the lower troposphere can be attributed to boundary layer pollution, but at about 400 hPa below the tropopause mostly non-solvable components (e.g. NO$_x$) are left.

Profiles in the Iranian ASMA part (Fig. 6c) have a different history of origins, and with just one minimum in the mid-troposphere are mostly C-shaped (Fig. S11a). During summer the Arabian Peninsula is dry. Deep convection (as indicated by lightning NO$_x$ emissions in Fig. 7c) is mainly localised in the south-western Yemen region (Fig. 10), i.e. at the edge of the region we defined for calculating profiles of the Iranian part of the ASMA. Washing out is negligible throughout most of the Iranian region, and therefore NO$_y$ can rise to about 400 hPa below the tropopause (circled in Fig. 6c).

[Figure]

[Figure]

**Figure S11.** Simulated profiles of $NO_y$ as simulated for 15 August 2012. These are examples of a C-shaped profile in the Iranian region (a) and an E-shaped profile in the Tibetan region (b).

**S5 Processes**

**S5.1 Lightning NO$_x$ sensitivity simulations**

[revised manuscript text omitted]
 loss (circled in Fig. S13s), indicative of the NO$_x$-limited photochemical regime.

O$_3$ mixing ratios respond to ΔLiNO$_x$ essentially like net O$_3$ production in the UT (Figs. S13nop). The altitude of maximum relative ΔO$_3$ is slightly lower than the altitude of maximum absolute changes to net O$_3$ production. We attribute this effect to upwards increasing absolute O$_3$ mixing ratios.

[Figure]

**Figure S13.** Evolution of simulated trace gas profiles and related diagnostics during September 2012 in the ASMA region (15° N – 35° N, 30° E – 100° E), and their sensitivity to LiNO$_x$ emissions. The vertical axes cover the UTLS and middle troposphere, and their coordinates are given as pressure distance to the tropopause. Left column: QCTM reference simulation (q01). Middle column: s03 – q01, relative deviation of sensitivity simulation s03 with respect to q01 for trace gases, absolute deviation for net O$_3$ production. Right column: s04 – q01.

**S5.2 Entrainment of lower tropospheric air**

[Figure]

**Figure S14. EMAC-simulated monthly mean distributions of CO (left) and deep convective mass flux (middle) in different pressure altitudes during spring (April 2012). The right column shows the deep convective mass flux of CO based on individual output steps. Blue rectangles mark the outline of the regions used to produce Figs. 5, 6, 7.**

[Figure]

**Figure S15. As Fig. S14, but for August 2012, i.e. during the monsoon season.**

**S5.3 Photochemical O₃ production**

[Figure]

**Figure S16.** **Simulated profiles of net O₃ production in the Tibetan region. Auxiliary red lines indicate the mean and maximum net O₃ production in the UT, which are both higher in summer than in spring.**

[Figure]

**Figure S17.** **Selection of parameters related to photochemical O₃ production in a meridional curtain through the Tibetan part of the ASMA in a snapshot taken mid August 2012. The left column shows the results of the EMAC QCTM simulation that has been introduced in appendix A. The other columns show the difference of that reference to sensitivity simulations, which feature identical dynamics but differ in biomass burning (BB) and lightning NOₓ (LiNOx) emissions. The black lines represent the 13 nmol mol⁻¹ day⁻¹ isocontour of net O₃ production (taken from panel γ), the grey line denotes the tropopause.**

**S5.4 Splitting-up and stirring**

[Figure]

**Figure S18. EMAC simulated O₃ mixing ratios and streamlines at 168 hPa in the ASMA region. The snapshots are 20 hours apart and cover the period from 1 August to 18 September 2012.**

**Larger fractions of the O₃-rich fringe were entrained during splitting up events. The sequence of snapshots (Fig. S18) covers almost half a monsoon season and episodic O₃-poor upwellings over the Tibetan plateau are smaller and shorter lived than O₃-rich regions at 168 hPa. This is consistent to the long-term average at a corresponding isentropic level (Fig. 1i).**

---

## Author Response (AR3)

**Co-Editor Decision: Publish as is** (15 Mar 2018) by Marc von Hobe
Comments to the Author:
Dear Authors,

I think that the manuscript has been further improved and the structure is much more helpful now. Thanks for putting this effort into it! I hereby accept your manuscript for publication in ACP.

Best regards,
Marc

We appreciate this and wish to thank the Co-Editor for his support throughout the review and discussion phase.

Changes to the manuscript:

1. Full first names for all authors
2. Modified figures according to the ACP guidelines (e.g. no spaces between start and end of ranges; space between degree sign and direction of coordinates)
3. Reduced number of font sizes in the figures

[revised manuscript text omitted]